# Assessing the Thermohaline Coherence of Mesoscale Eddies as described from In Situ Data

Yan Barabinot[1], Sabrina Speich[1], and Xavier Carton[2]

[1]Ecole Normale Supérieure, Laboratoire de Météorologie Dynamique (LMD), 24 rue Lhomond, Paris 75005, France
[2]Université de Bretagne Occidentale (UBO), Laboratoire d'Océanographie Physique et Spatiale (LOPS), IUEM, rue Dumont Durville, Plouzané 29280, France

**Correspondence:** Yan Barabinot (yan.barabinot@lmd.ipsl.fr)

**Abstract.** Mesoscale eddies are a ubiquitous feature of the global ocean. According to Lagrangian theory, these eddies often transport a distinct water mass within their cores, making them materially coherent. This study aims to determine if such a distinct water mass exists in eddy cores, thereby verifying their material coherence using in situ data, despite the lack of temporal continuity. We introduce the term "thermohaline coherence" to describe this approach. Identifying such a water mass would signal Lagrangian transport from the eddy formation region. We analyzed the water masses at the cores of various eddies sampled during eight research cruises, using high-resolution data (approximately 20 km horizontally and 10 m vertically). We revisited coherence definitions and checked data accuracy. Comparing the horizontal positions of these core anomalies with eddy surface signatures revealed that surface data alone are insufficient for characterizing the eddy material coherence. To calculate eddy volumes, we compare thermohaline anomalies with other criteria and we present two methods for extrapolating eddy volumes from a single hydrographic section. The results show that the outermost closed contour of the Brunt-Väisälä frequency anomaly at each depth provides a reliable approximation for the eddy boundary.

## 1 Introduction

Mesoscale eddies are ubiquitous energetic structures in the ocean and are one of the major sources of ocean variability (Stammer, 1997; Wunsch, 1999). They are thought to have a major influence on the propagation of hydrological properties by advecting them over long distances and timescales (McWilliams, 1985). The lifetime of such structures often exceeds several months and can reach several years (Laxenaire et al., 2018; Ioannou et al., 2022), a fact that highlights their resilience and their "coherence".

The word "coherence" was first introduced to describe specific structures in turbulent boundary layers. This terminology was used to imply that the near-wall region contains certain basic flow modules or structures that give rise to the apparently ordered development observed in the wall layer (Kline et al., 1967; Crow and Champagne, 1971; Roshko, 1976). Thus, "coherence" was initially a concept of persistence in time and of "order/disorder". Then, these structures were studied more and more, and other definitions appeared that included vorticity. Several papers have been published proposing that a "coherent" structure was characterized by an instantaneous component of large-scale vorticity that dominated the rest of the flow (Hussain and Zaman, 1980; Hussain, 1986; Zaman and Hussain, 1981). "Coherence" was thus a concept defined in space and time, but remained

qualitative. Later works applied the concept of "coherence" to geophysical fluid dynamics, especially for mesoscale eddies, and implicitly proposed that a "coherent eddy" was a temporally persistent vortex of radius larger than the Rossby deformation radius (Charney, 1971; Herring, 1980; Hua and Haidvogel, 1986; McWilliams, 1984, 1989).

With the advent of altimetry, oceanic eddies were often characterized by sea surface height anomalies organized as a set of concentric closed isolines. These isolines could be followed in time via a (mostly) continuous trajectory of their center (Chaigneau et al., 2009; Chelton et al., 2011; Pegliasco et al., 2016; Zhang et al., 2016). Since these studies investigated the persistence of the eddy flow in time, they were related to the concept of coherence defined by McWilliams (1984, 1989). Here, we refer to this concept of coherence as *Kinematic Coherence* (KC). However, KC is only qualitative as there is no quantitative criterion for determining when an eddy ceases to be persistent in time.

For a quantitative characterization of eddy coherence, oceanographers initially relied on flow stability criteria (e.g., Fjörtoft, 1950; Eliassen, 1951; Pedlosky, 1964; Bretherton, 1966; Hoskins, 1974; Carton and McWilliams, 1989; Ripa, 1991). However, recent studies have shown that even in the presence of moderate, localized instability, a vortex can remain kinematically coherent for long periods of time (de Marez et al., 2020). Conversely, long-lived vortices can become unstable, stretch, shed filaments, and break under the influence of ambient velocity shear (Carton, 2001; Carton et al., 2010b). Therefore, vortex stability is not equivalent to *kinematic coherence*.

Nor is KC equivalent to exact eddy invariance: indeed, an eddy can shed filaments or incorporate water masses into its core by fluid advection or entrainment. Lateral diffusion may transform or modify these water masses. These processes occur close to the maximum velocity location, where the strain is intense. Conversely, eddy cores are loci of stronger vorticity than strain. Consequently, Eulerian criteria for KC and for the determination of eddy shapes have been derived using these two quantities (Hunt et al., 1988; Okubo, 1970; Weiss, 1991; Chong et al., 1990; Tabor and Klapper, 1994).

*In situ* measurements have shown that mesoscale eddy cores contain different water masses from those of the surrounding environment. The core water masses are characteristic of the eddy formation region. Mesoscale eddies can transport these water masses over several thousand kilometers (Chelton et al., 2011; Dong and McWilliams, 2007; Zhang et al., 2014). To explain the persistence of the water mass properties of the eddy along the trajectory, Lagrangian approaches have been used to find coherence criteria. Flierl (1981) showed that when the tangential velocity of the vortex is higher than its translational velocity, fluid particles are trapped in the vortex core.

A new theory was then proposed by Haller (2000); Haller et al. (2015). First, Haller (2005) who criticized the KC theory for being reference frame-dependent and not objective, and imposed a vortex coherence criterion to be invariant under a reference frame change. To construct an objective Lagrangian definition of a mesoscale vortex, the Lagrangian Coherent Structures (LCS) framework was then proposed. In Haller's vision, a coherent vortex traps a mass of water in its core as it forms. This vortex ceases to be coherent when it loses its trapped water mass, that is when trajectories are no longer closed, although no publication has been able to quantitatively determine the point at which a vortex loses its trapped water mass. We refer to this definition as *Material Coherence* (MC). Objective Lagrangian criteria have been used by these authors to detect materially coherent vortices (Haller, 2015; Xia et al., 2022). The application of these criteria proves that ocean eddies identified by Eulerian perspectives leak material across their identified boundaries relatively quickly (Andrade-Canto et al., 2020; Serra and

Haller, 2017; Denes et al., 2022). This appears to be a major drawback of using Eulerian approaches to quantify material coherence and mass transport through eddies.

However, these criteria have mostly been applied using altimetry-derived geostrophic velocity fields, although some numerical simulations have attempted to do so (Beron-Vera et al., 2019); these 2D fields are not fully representative of the wide variety of oceanic eddies. In fact, eddy flow may be partially ageostrophic and not surface intensified. This is also true for eddies identified from satellite altimetry, as the observed sea surface dynamic height provides vertically integrated information about the local density field (e.g., Laxenaire et al., 2019, 2020). In addition, the along-track and gridded altimetry products are smoothed fields compared to the directly observed sea surface heights. Therefore, the derived surface geostrophic velocity is an approximation of the real velocity field (see e.g. Subirade et al. (2023)). Furthermore, MC theory is based on advection processes only and often does not consider the potential permeability of the eddy boundary due to diffusion processes or lateral intrusion (Joyce, 1977, 1984; Ruddick et al., 2010). Nevertheless, some criteria including diffusion can be found in the literature (Froyland et al., 2010; Froyland, 2013; Froyland and Padberg-Gehle, 2015). For instance, using data collected over several years in one single eddy, Armi et al. (1989) showed that Mediterranean Water Eddies (or meddies) can remain essentially coherent for 2 years before collapsing very rapidly due to thermohaline intrusions. In particular, MC theory ignores the fact that water masses at the edge of eddies can change their properties due to various types of instabilities. Finally, few long-lived MC eddies have been found compared to a larger number of KC eddies (Beron-Vera et al., 2013; Haller, 2015).

The MC definition of eddy coherence is rigorous: it describes how an eddy can trap and transport tracers over long distances. However, the MC view appears to be restrictive because it suggests that mesoscale eddies stop transporting water when the core loses its coherence, although an eddy can also advect a mass of water at its edge, creating a crown-like structure. Recent Lagrangian analyses found that only small coherent inner cores of ocean eddies exist for long periods of time (Abernathey and Haller, 2018; Wang et al., 2016), while others (Denes et al., 2022) found that ocean eddies may consist of coherent inner cores, and quasi/semi-coherent outer rings, thus challenging the notion that ocean eddies have precise boundaries. This has also been supported by observational evidence (Barabinot et al., 2024). The boundary of a materially coherent inner core is undistorted/unfilamented over a finite time window, such that diffusive mixing across the boundary is minimised (Haller, 2015). Future studies should further confront the Lagrangian and Eulerian visions of coherence, especially for eddy boundaries.

Recent studies have shown a difference of more than 30 % between the number of KC and MC eddies detected (Vortmeyer-Kley et al., 2019; Liu et al., 2019). The estimation of eddy mixing is highly dependent on the criterion used. The amount of tracer transported by mesoscale eddies appears to be larger using Eulerian criteria than Lagrangian criteria because the latter are more restrictive (see Figure 8 of Beron-Vera et al. (2013)). This lack of consensus has implications for estimating tracer transport (Dong et al., 2014; Wang et al., 2015; Xia et al., 2022) and hence ocean mixing.

It should be noted that the KC and MC visions do not appear to be incompatible. Altimetry and Argo floats show that almost all KC eddies are associated with a thermohaline anomaly in their core (Aguedjou et al., 2021). Thus, a kinematically coherent eddy can be a materially coherent eddy. The converse, MC implies KC, is also true, since the definition of MC requires an intense velocity field and kinematic coherence over a long period. While these two definitions are not exclusive, they are obviously not equivalent.

*In situ* data do not provide the temporal continuity necessary to apply standard MC criteria. However, they do show the vertical structure of eddies. Therefore, our purpose here is to define the coherence of eddies as the trapping of a distinct water mass in the eddy core, characteristic of the region of formation of that eddy. Indeed, our main idea is to emphasize how the Lagrangian coherence definition can be coupled to the uniqueness of ocean water masses. In fact, the latter represent distinct "fingerprints" within the ocean, characterized by specific combinations of temperature and salinity that are not randomly

distributed but rather result from precise regional conditions. Each water mass originates in a specific region where unique air-sea interactions imprint it with a characteristic temperature-salinity (T-S) signature. Once formed, these water masses are remarkably consistent in their properties as they are advected within the ocean, below the mixing layer, allowing them to be identified and tracked over great distances. Our assessment will also be based on the vertical structure of the potential vorticity (PV), by showing that a PV anomaly is trapped in this core. A compact PV anomaly is indeed associated with a local

recirculation of water masses. Furthermore, PV is mostly modified near the ocean surface (Marshall et al., 1999, 2012). PV can be considered as a tracer for the deeper part of surface eddies, or for subsurface eddies themselves.

      In this paper, using the water masses and PV approach, we focus on eddies that have been sampled with a good resolution of O(20 km) horizontally and O(10 m) vertically from oceanographic cruises in 7 different regions, providing a variety of structures and of trapped water masses. The first objective is to assess the number of materially coherent structures in the

collected dataset. This is to check if we retrieve the same fraction of eddies as observed by satellite altimetry and the "material coherence" approach. The approach is not new, but this is the first time that data from multiple cruises are used to assess the MC. This approach relies on the fact that the thermohaline properties of the eddy core are maintained throughout its lifetime. Indeed, with the advent of Argo floats, measuring thermohaline anomalies in eddy cores across different regions has become easier, and several examples of thermohaline anomalies maintained throughout the eddy lifecycle can be found (Aguedjou

et al., 2021; Armi et al., 1989; Laxenaire et al., 2019, 2020; Paillet et al., 2002). Therefore, by calculating the thermohaline anomalies on the isopycnals, the difference in thermohaline properties between the eddy cores and their surroundings can be highlighted and the material coherence can be assessed, even if it is only assessed at one point in time. An eddy is considered to be materially coherent when the maximum anomaly is reached at the eddy center on a 2D vertical section (region where the measured velocity tends to zero) and there is a marked difference in values between the enclosed and surrounding waters. We

propose to refer to this definition as *Thermohaline Coherence* (TC), which is a consequence of the *Material Coherence* (MC), but which can be assessed by *in situ* data.

      The second objective is to correlate the internal anomaly with its surface signature as revealed by satellite altimetry. This is done to test whether it is possible to assess the coherence of eddies from satellite data alone. The comparison of *in situ* data with satellite altimetry has already been done (see L'Hégaret et al. (2014); Carton et al. (2010a)) but we extend it to a larger amount

of data and in particular to the study of eddy coherence. The purpose here is to present cases where the use of satellite altimetry data could lead to some misinterpretations. Once eddies are identified as TC, the third objective is to find the best criterion to apply to 2D ship sections to compute their material volume. To this end, we propose two methods for extrapolating their transport volume from a single section sampling their properties at depth. We then compare several criteria to determine their boundaries: thermohaline anomalies, Ertel's Potential Vorticity ($EPV$), and relative vorticity. We also use a newly proposed

criterion based on $EPV$ (see Barabinot et al. (2024)). The goal here is to determine which of the criteria defined in the previous section (thermohaline anomalies, gradients, $EPV$) is most effective in detecting the coherent core. Although this approach of comparing criteria to determine eddy boundaries provides important information on heat and salt transport by eddies, it is rarely applied by studies that post-process cruise data. We refer the reader to the supplementary material for more details on these eddies.

The paper is organized as follows. Section 2 describes the set of *in situ* data used and the identification of eddies using ship-based or satellite altimetry data. Section 3 presents the diagnostics used to characterize the core and boundary of mesoscale eddies and relates them to MC definitions. In particular, a section is devoted to the relative errors in the data that affect the accuracy of the results. Then, assuming the circularity or ellipticity of a sampled eddy, two methods are proposed to reconstruct its 3D structure. In section 4, we propose two methods to extrapolate the eddy volume from a single ship section and in section 5, we discuss the thermohaline coherence of sampled eddies and present results on volume approximations.

## 2 Data collection and processing

### 2.1 Data collection: cruises

The data analyzed here were collected during 8 oceanographic cruises in 7 different regions: the EUREC4A-OA campaign along the northern coast of Brazil, which studied mesoscale eddies and the ocean-atmosphere coupling; the MARIA S. MERIAN MSM60 expedition, which was the first basin-wide section across the South Atlantic following the SAMBA/SAMOC line at 34°30'S; the Physindien 2011 experiment along the Omani coast (western Arabian Sea), which studied the eddy field in this area; the FS METEOR M124 expedition, which was the first of the two SACross2016 expeditions; the MSM74 cruise, which was dedicated to determining the intensity of southward water mass transport and transformation in the boundary current systems off the sea; the M160 measurements, which contributed to the understanding of the ocean eddies generated in the Canary Current system; and three cruises - KB 2017606, KB 2017618, HM 2016611 - whose main objective was to study eddy dynamics in the Lofoten Basin. The goal was to collect a relatively large number of eddies sampled in different regions at different times of their life cycle. To be able to derive our diagnostics from the data, the campaigns must not only have carried out hydrological measurements, but also velocity measurements over the same depth range. This requirement significantly reduces the number of potentially available cruises. Table 1 summarizes the basic information about the cruises. The instruments used are Conductivity Temperature and Depth (CTD), underway CTD (uCTD), (lowered and ship-mounted) Acoustic Doppler Current Profiler (lADCP or sADCP).

Here, we recall the measurement uncertainties for each instrument used. They will be important for estimating errors in the calculated diagnostics. For the CTD instrument, temperature and salinity are measured with uncertainties of $\pm0.002°$ C and $\pm0.005$ psu, respectively. For the uCTD instrument, the uncertainties are $\pm0.01°$ C and $\pm0.02$ psu for temperature and salinity measurements respectively. And for the ADCP instrument, the horizontal velocity is typically measured with an uncertainty of $\pm3$ cm.s$^{-1}$.

**Table 1.** Basic information about the cruises: date, main ocean basin where the campaign took place, sampling instruments used in this paper (it does not refer to every instrument used during the cruises). CTD for Conductivity Temperature Depth, uCTD for Underway Conductivity Temperature Depth, XBT for Expendable Bathythermograph, xCTD for Expendable Conductivity Temperature Depth, ADCP for Accoustic Doppler Current Profiler.

| Name | date | location | Instruments |
|---|---|---|---|
| EUREC4A-OA | 20/01/2022-20/02/2020 | North Brazil | CTD/uCTD/XBT/sADCP |
| MSM60 | 4/01/2017-1/02/2017 | SAMBA/SAMOC line $(34°30'S)$ | CTD/lADCP (38 kHz) |
| Physindien 2011 | 03/2011 | Red Sea, Persian Gulf | xCTD /ADCP (38 kHz) |
| M124 | 29/02/2016 - 18/03/2016 | South Atlantic | uCTD/XBT /lADCP (38 kHz) |
| MSM74 | 25/05/2018 - 26/06/2018 | Labrador Basin | CTD /sADCP (75kHz) |
| M160 | 23/09/2019 - 20/12/2019 | Canary | CTD / lADCP (75 kHz) |
| HM2016611 | 26/05/2016 - 15/06/2016 | Lofoten Basin | CTD /lADCP (38 kHz) |
| KB2017606 | 10/03/2017 - 23/03/2017 | Lofoten Basin | CTD /lADCP (38 kHz) |

## 2.2 Data processing

Oceanographic research cruises often collect data along vertical sections that include vertical profiles. Therefore, we define the resolution of a vertical section as the average of all distances between successive profiles along the same section. Since hydrological and velocity instruments do not sample the ocean with the same resolution, the two types of measurements are distinguished (see table 2). For example, the hydrological properties of the surface anticyclonic eddy from EUREC4A-OA (denoted $N°1$ in Table 2) were sampled using CTD/uCTD instruments with a resolution of 3.5 km horizontally and 1 m vertically, while its dynamical properties were measured using sADCP (75 kHz) instruments with a resolution greater than 1 km horizontally and 8 m vertically.

The raw data were calibrated and then interpolated. To limit noise, linear interpolations were performed in $x$ (horizontal) and $z$ (vertical) directions. We chose first-order polynomial functions to avoid creating artificial fields. The typical grid size of the interpolated data is 1 km horizontally and 1 m vertically. The data were then smoothed with a numerical low-pass filter of order 4 (scipy.signal.filt in Python). The choice of cut-offs is subjective and depends on the scales considered. Here we are considering mesoscale eddies, so we chose $L_x \geq 10$ km and $L_z \geq 10$ m for the horizontal and vertical length scales where possible to remove submesoscale processes that can blur eddy boundaries. In fact, the cut-off period must be longer than the sampling resolution of the calibrated data. The smoothing parameters are summarized in Table 2.

## 2.3 Eddy identification in *in situ* data acquired from research cruises

Since on density vertical sections the rotating flow mainly satisfies the geostrophic equilibrium with often a small cyclostrophic correction (Cushman-Roisin, 1994; Penven et al., 2014; Ioannou et al., 2019), eddies can be identified by observing vertical deviations of isopycnals; they are usually accompanied by changes in the sign of the velocity field orthogonal to the section.

**Table 2.** Cruise names, type, and resolution of the 25 mesoscale eddies studied. The resolution of the hydrographic data is denoted by $\Delta_H$, while the velocity data is denoted by $\Delta_V$. For each type of data, the horizontal and vertical resolutions are explained, as well as the cutoff of the low-pass filter used to smooth the data. Some eddies have the same horizontal resolution when sampled along the same transect. The variation in resolution for eddies on the same transect is negligible. AE = anticyclonic eddy, CE = cyclonic eddy, surf = surface eddy, sub = subsurface eddy.

| $N°$ | Cruise | Type | $\Delta_H x\ (L_x)$ [km] | $\Delta_H z\ (L_z)$ [m] | $\Delta_V x\ (L_x)$ [kHz] | $\Delta_V z\ (L_z)$ [m] |
|---|---|---|---|---|---|---|
| 1 | | AE KSurf/Tsub | 3.5 (10) | | | |
| 2 | EUREC4A-OA | AE KSub/TSub | 8.4 (10) | 0.5 (10) | 0.3 (10) | 8 (10) |
| 3 | | AE KSub/TSub | 13 (15) | | | |
| 4 | | CE KSurf/TSub | 26.3 (50) | | 26.3 (50) | |
| 5 | MSM60 | CE KSurf/TSub | 41.7 (50) | 1 (10) | 41.7 (50) | 8 (10) |
| 6 | | CE KSurf | 43 (50) | | 43 (50) | |
| 7 | Physindien 2011 | AE KSurf/TSub | 1.8 (10) | 0.1 (10) | 0.3 (10) | 8 (10) |
| 8 | | AE KSub/TSub | 1.7 (10) | | | |
| 9 | | CE KSurf/TSub | 25 (30) | | | |
| 10 | | AE KSurf/TSub | 23 (30) | | | |
| 11 | | AE KSurf/TSub | 23 (30) | | | |
| 12 | | AE KSub/TSub | 23 (30) | | | |
| 13 | M124 | AE KSub/TSub | 12 (30) | 0.5 (10) | 0.3 (10) | 32 (40) |
| 14 | | AE KSub/TSub | 21 (30) | | | |
| 15 | | AE KSub/TSub | 21 (30) | | | |
| 16 | | AE KSub/TSub | 20 (30) | | | |
| 17 | | AE KSurf/TSub | 35.7 (40) | | | |
| 18 | | CE KSurf/TSub | 33.5 (40) | | | |
| 19 | MSM74 | CE KSurf/TSub | 33.5 (40) | 1 (10) | 0.3 (10) | 8 (10) |
| 20 | | CE KSurf | 20.3 (30) | | | |
| 21 | | AE KSurf | 20.3 (30) | | | |
| 22 | M160 | CE KSurf | 15.1 (20) | 1 (10) | 0.3 (10) | 8 (10) |
| 23 | KB2017606 | AE KSub/TSub | 6.6 (10) | 1 (10) | 6.6 (10) | 8 (10) |
| 24 | | AE KSub/TSub | 5.3 (10) | | 5.3 (10) | |
| 25 | HM2016611 | AE KSub/TSub | 5.8 (10) | 1 (10) | 5.8 (10) | 8 (10) |

To analyze the true thermohaline anomalies in eddy cores, the ship must have passed close enough to the eddy center. In the following, we separate such sampled eddies from others. We call $R_{max}$ the radius of maximum velocity if the eddy is axisymmetric, and $e$ the distance between the eddy center and its orthogonal projection on the ship's track (see Figure 1). An eddy is considered well sampled if $e \leq R_{max}/2$. Obviously, eddies are not completely axisymmetric and we adjust the criterion

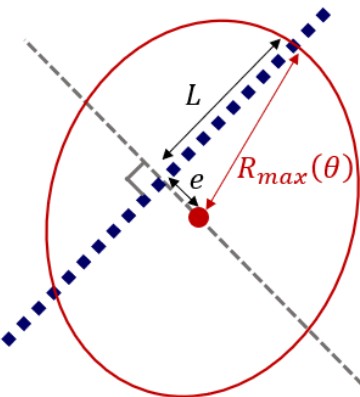

**Figure 1.** A schematic example of a well-sampled eddy at the sea surface: the red dot indicates the estimated center; the dark blue squares are locations of vertical profiles; the red circle is the radius of maximum tangential velocity. The dashed gray line is perpendicular to the ship track passing the eddy center.

for this case using $L$ as defined in Figure 1. Using the Pythagorean theorem, an eddy is well sampled if the following condition is satisfied: $e \leq L/\sqrt{3}$. Table 3 summarizes the basic properties of eddies and describes which eddies are well-sampled. In fact, this table underscores the difficulty of obtaining complete (all boundaries visible) well-sampled structures with *in situ* data. For a mesoscale eddy marked "B" in the table, the eddy radius cannot be calculated and dashes are used. Note that the radius $L$ has also been estimated for not well-sampled eddies.

The position of the eddy center is estimated using the routine from Nencioli et al. (2008) at the depth of the observed maximum velocity, assuming that the position of the center does not vary too much with depth. The routine constructs a rectangular area around the ship track with a given grid size. Then, for each grid point, the distance-weighted average of the tangential velocity is computed using each velocity vector measured along the transect. The center of the eddy is defined as the point where the mean tangential velocity is maximum. This routine is implemented at each geopotential level on a 2D grid

plane.

      Finally, we are able to locate every well-sampled eddy during the 8 cruises. In practice, however, some non-well-sampled eddies have sufficient characteristics to assess their thermohaline coherence. In total, 25 eddies with 17 anticyclonic eddies (hereafter AE) and 8 cyclonic eddies (hereafter CE) were sampled, including 19 well-sampled eddies (12 AE and 7 CE).

      Here we specify the determination of the eddy type. On the one hand, the cyclonic or anticyclonic aspect is derived from the

deviation of the isopycnals. On the other hand, the surface or subsurface intensification of the vortex depends on the variable used to characterize its vertical structure. Thus, two variables can be used: the location of the maximum velocity and the location of the maximum thermohaline anomalies (defined later by Eqs. (1) and (2)). A kinematic subsurface eddy (KSub) is defined as an eddy for which the maximum velocity is below $-70\,\mathrm{m}$ depth. Conversely, a kinematic surface eddy (KSurf) has its maximum velocity in the upper $-70\,\mathrm{m}$ depth. A thermohaline subsurface eddy (TSub) is an eddy for which the maximum

of the thermohaline anomalies on isopycnals (see separate section) is below $-70$ m depth. In contrast, thermohaline surface eddies (TSurf) have their maximum anomalies defined within this upper layer. In fact, ADCP data are only accurate after 2 or 3 bins of depth. Some cruises do not even provide data in the first $-50$ m. In addition, isopycnal levels must match between the section and the climatological mean, which is rarely satisfied near the surface due to near-surface variability. As a result, it is often impossible to calculate anomalies above $-70$ m. Therefore, the $-70$ m depth threshold has been chosen to have a unique value regardless of the variable being considered. In some cases, eddies are not thermohaline coherent and no maximum of anomalies can be found at the center of the eddy (see part 5.1). Therefore, only the velocity is used to evaluate the vertical structure. One can note that an eddy labeled KSurf is not necessarily TSurf.

## 2.4 Satellite altimetry data and the TOEddies algorithm

To compare the surface and subsurface signature of sampled eddies, we present satellite altimetry data and a detection algorithm based on Absolute Dynamical Topography (ADT) derived from these data.

Sampled eddies are identified and tracked in time by the TOEddies automatic detection algorithm (Laxenaire et al., 2018, 2019, 2020). This detection is applied to ad-hoc Near Real Time (NRT) ADT maps during the field experiments. These products are provided by Collecte Localisation Satellites (CLS) and have been generated using a Mean Dynamic Topography (MDT) with a higher resolution (1/8° instead of 1/4°) than the standard MDT product (Rio et al., 2011, 2014).

The TOEddies method is based on the algorithm proposed and developed by Chaigneau et al. (2009) and has already been used in studies analyzing different aspects of Atlantic Ocean dynamics, such as the origin and evolution of the Agulhas Current rings (Laxenaire et al., 2018, 2019, 2020), the role of mesoscale eddies in meridional transport over the zonal South Atlantic GO-SHIP section of the MSM60 cruise (Manta et al., 2021), in the EUREC4A-OA region (Subirade et al., 2023), and the effect of mesoscale eddies on the formation and transport of South Atlantic subtropical mode water (Chen et al., 2022).

Assuming that eddies are in geostrophic equilibrium, TOEddies identifies eddies as closed contours of the ADT that contain only a local extremum. As a result, at any given time, eddy streamlines should coincide with the closed isolines of the daily ADT maps. Thus, the ADT, and not the Sea Level Anomaly (SLA), represents the geostrophic stream function. In fact, the SLA is very sensitive to large Sea Surface Height (SSH) gradients associated with intense currents and quasi-stationary meanders or eddies that characterize the MDT (see example in Pegliasco et al. (2021)). TOEddies thus identifies the local ADT extrema (maxima and minima) and searches for the outermost closed ADT contour around each extremum. In addition to the outermost closed ADT contour, TOEddies also identifies the contour where the mean azimuthal velocity is maximum using geostrophic velocities derived from ADT maps.

## 3   Methods for eddy boundaries characterization

In this section, we describe four eddy boundary detection methods that have been widely applied to *in situ* data analysis. In particular, the use of T/S anomalies, gradients, and potential vorticity (PV) has been implemented extensively to develop

**Table 3.** Basic properties of mesoscale eddies: typical variation of isopycnal deviation, $H$; radius of maximum velocity on the vertical section ($L \neq R_{max}$ of Figure 1); maximum velocity ($V_m$) associated with $L$; apparent Rossby number $R_o = V_m/(f_0 L)$. Since mesoscale eddies are not axisymmetric, $V_m$ is taken as the maximum modulus of $V_o$, the velocity component orthogonal to the ship section. The "Well-sampled" column indicates whether the eddy is well-sampled (Yes) or not (No). The "Complete" column indicates whether the eddy has been completely sampled. The letters [C/B/H] mean [Complete/Boundary/Half]: "Complete" if the eddy structure is clearly visible on vertical sections, a "+" is added if vertical boundaries are visible, "Boundary" if only one boundary is visible, and "Half" if one boundary plus the center is visible. The center refers to the location where the velocity $V_o$ is zero. If only half of the vortex structure has been sampled, the Nencioli et al. (2008) routine cannot be applied, so we enter "-".

| $N°$ | Cruise | type | $H$[m] | $L$[km] | $V_m$[m.s$^{-1}$] | $R_o$ | Well-sampled | Complete [C/H/B] |
|---|---|---|---|---|---|---|---|---|
| 1 | | AE | 70 | 121 | 1.14 | 0.44 | Yes | C+ |
| 2 | EUREC4A-OA | AE | 220 | 71 | 0.96 | 0.61 | Yes | C+ |
| 3 | | AE | 115 | 111 | 0.83 | 0.32 | Yes | C+ |
| 4 | | CE | 375 | 85 | 0.6 | 0.11 | Yes | C+ |
| 5 | MSM60 | CE | 190 | 42 | 0.33 | 0.10 | Yes | C |
| 6 | | CE | 170 | 28 | 0.6 | 0.26 | Yes | C |
| 7 | Physindien 2011 | AE | 55 | 95 | 0.99 | 0.38 | Yes | C+ |
| 8 | | AE | 20 | 10 | 0.36 | 0.66 | Yes | C+ |
| 9 | | CE | 120 | 67 | 1.53 | 0.28 | Yes | C |
| 10 | | AE | 200 | 58 | 1.27 | 0.26 | Yes | H |
| 11 | | AE | 105 | 55 | 0.95 | 0.21 | Yes | C |
| 12 | | AE | - | - | - | - | - | B |
| 13 | M124 | AE | 130 | 54 | 0.75 | 0.19 | Yes | C |
| 14 | | AE | 40 | 34 | 0.32 | 0.13 | No | C |
| 15 | | AE | 30 | 52 | 0.32 | 0.08 | No | C |
| 16 | | AE | 150 | 61 | 0.73 | 0.16 | Yes | C |
| 17 | | AE | 180 | 28 | 0.23 | 0.06 | Yes | C |
| 18 | | CE | 100 | 35 | 0.17 | 0.04 | No | C |
| 19 | MSM74 | CE | 100 | 32 | 0.43 | 0.1 | Yes | C |
| 20 | | CE | 150 | 23 | 0.24 | 0.04 | Yes | C |
| 21 | | AE | 150 | 12 | 0.3 | 0.2 | Yes | C |
| 22 | M160 | CE | 50 | 49 | 0.46 | 0.09 | Yes | C |
| 23 | KB2017606 | AE | - | - | - | - | - | B |
| 24 | | AE | 500 | 15 | 0.78 | 0.34 | Yes | C+ |
| 25 | HM2016611 | AE | - | - | - | - | - | B |

diagnostics for eddies sampled during *in situ* experiments (Aguedjou et al., 2021; Paillet et al., 2002; Bosse et al., 2019; Carton et al., 2002). These methods have proven effective in improving our understanding of the dynamic properties of oceanic eddies.

## 3.1 Thermohaline anomalies on isopycnals surfaces

The ability of eddies to trap and transport water masses is the basis of the MC definition. Here, we evaluate this definition by computing temperature and salinity anomalies on isopycnals in eddy cores relative to a climatological average following the method of Laxenaire et al. (2019, 2020). The climatological average of temperature/salinity on geopotential levels is calculated using ARGO float profiles over 20 years in a small area around the sampled eddy. It is worth noting that the average is computed using profiles measured during the corresponding month in which the considered eddy has been sampled. The Coriolis dataselection.euro-argo.eu database is used. A square of side $0.5°$ is built around the eddy center estimate, so that the center is at the intersection of the diagonals. Taking $T^*$ and $S^*$ as two reference profiles in temperature and salinity (outside the eddies) and $T$ and $S$ as *in situ* profiles (inside the eddies), thermohaline anomalies on isopycnals are computed as follows:

$$\forall \sigma, \quad \Delta T(\sigma) \quad = \quad T(\sigma) - T^*(\sigma), \tag{1}$$

$$\forall \sigma, \quad \Delta S(\sigma) \quad = \quad S(\sigma) - S^*(\sigma), \tag{2}$$

where $\sigma$ is the potential density at atmospheric pressure. These anomalies are computed on isopycnal surfaces but interpolated to the geopotential levels to facilitate comparison with other criteria. As introduced earlier, we define a thermohaline subsurface eddy (TSub) as an eddy with an anomaly maximum location deeper than $-70$ m. Conversely, a thermohaline surface eddy (TSurf) exhibits an anomaly maximum above $-70$ m depth. These anomalies can separate two water masses that have the same potential density but differ in their thermohaline compositions. As a result, they are highly effective in delineating the TC core of an eddy. Taking into account the resolution of the instruments, the uncertainty in the thermal (or salinity) anomalies is approximately $\pm 0.01°$ C ($\pm 0.02$ psu) when uCTD data are considered, and $\pm 0.002°$ C ($\pm 0.005$ psu) when only CTD measurements are used.

These anomalies are highly dependent on the temperature or salinity gradient along the isopycnals. Therefore, we compare the maximum values of our anomalies with the standard deviation of the temperature and salinity fields in each region in the period 1991-2020 provided by the World Ocean Atlas 2023 (Locarnini et al., 2024; Reagan et al., 2024). The standard deviation of salinity and temperature for the month of each cruise is selected in a square with side $1°$ where the center of the eddy is located. Since these climatological standard deviations are based on Argo float profiles, eddy anomalies are often included in the construction of the climatological mean. Therefore, we consider our anomalies to be significant if their values are above the temperature and salinity standard deviations. Therefore, We define an eddy as TC if at least one of the two anomalies (temperature, salinity) is significant.

## 3.2 Gradients

Let $(x, z)$ be the vertical plane of the section, and using smoothed data, the derivatives of a quantity $a$ are approximated by a second-order Taylor expansion as follows $\partial_x a(x, z) \approx (a(x + \Delta x, z) - a(x - \Delta x, z))/(2\Delta x)$ (same for the variable $z$). For a given quantity $a(x, z)$, the norm of a gradient in a 2D slice is defined as follows

$$|\boldsymbol{\nabla} a| = \sqrt{(\partial_x a)^2 + (\partial_z a)^2}. \tag{3}$$

Since an eddy locally modifies isothermal or isohaline conditions with respect to the rest state, we expect this quantity to be useful for detecting eddy boundaries.

We also define the anomaly of the Brunt-Väisälä frequency as

$$N^2 = \frac{-g}{\sigma_0} \frac{\partial \sigma'}{\partial z}, \tag{4}$$

where $\sigma_0$ is a reference value averaged over each profile of the section, $g$ is the acceleration due to gravity and $\sigma'(x,z) =$
$\sigma(x,z) - \overline{\sigma}(z)$ the anomaly of the potential density computed on geopotential levels with respect to the climatological mean $\overline{\sigma}$. Since the eddy properties deviate from those of the background environment along the isopycnal surfaces, they are actually stratification anomalies. As such, the core appears as a region of low (or high) gradients for AE (or CE).

To calculate the relative vorticity, derivatives in two different horizontal directions are needed. For a single section from a research cruise this is not possible without further assumptions. An approximation of the relative vorticity is the "Poor Man's
Vorticity" (PMV) introduced by Halle and Pinkel (2003). It decomposes the measured velocities into a cross-track component $v_\perp$ and an along-track component $v_\parallel$. The relative vorticity is then approximated as $\zeta_z \approx 2\frac{\partial v_\perp}{\partial x}$. The factor 2 is added so that the PMV is equal to the actual $\zeta$ in an eddy core with solid body rotation. However, Rudnick (2001) and Shcherbina et al. (2013) used the along track derivative of the cross track velocities without the factor 2. Both approximations differ only in the way they estimate the cross-track derivative of the along track velocities. This method can be criticized and other approximations
can be found in the literature. In this article we arbitrarily choose the 2D approximation of Rudnick (2001):

$$\zeta_z \approx \frac{\partial v_\perp}{\partial x}. \tag{5}$$

Unless otherwise stated, the velocity field is always perpendicular to the section plane. Relative vorticity has been used extensively in studies based on analyses of satellite altimetry data or high resolution numerical models to locate eddies (Chouksey, 2023; Gula et al., 2016a, b). Some Lagrangian criteria are also based on this quantity and are therefore of interest (Haller, 2015;
Haller and Beron-Vera, 2013).

For these gradients, we refer the reader to Appendix B for details on uncertainties.

### 3.3 Ertel Potential Vorticity (EPV)

Here the 3D formula of $EPV$ (Ertel, 1942) is simplified and applied to *in situ* data. Under the Boussinesq approximation and hydrostatic equilibrium, the vertical momentum equation can be replaced by the hydrostatic approximation $\partial_z p = -\rho g$,
where $p$ is the pressure, $\rho$ the total density and $g$ the acceleration due to gravity. We also approximate $1/\sigma$ by $1/\sigma_0$. Therefore, following the method of L'Hegaret et al. (2016), the $EPV$ for a 2D vertical section has the following form

$$EPV = EPV_x + EPV_z = -\frac{\partial V_o}{\partial z}\frac{\partial b}{\partial x} + (\zeta_z + f_0)\frac{\partial b}{\partial z}, \tag{6}$$

where $b(x,z) = -g\sigma(x,z)/\sigma_0$ is the buoyancy, $V_o(x,z)$ is the velocity component orthogonal to the section plane, $f_0$ the Coriolis parameter and $\zeta_z(x,z)$ is as defined above. Note that this expression only gives a 2D approximation of the real $EPV$

with a baroclinic term $EPV_x$ and a term including the relative vorticity and stretching $EPV_z$. Therefore, the climatological $EPV$ average of the considered ocean region (hereafter $\overline{EPV}$) is

$$\overline{EPV} = f_0 \frac{d\bar{b}}{dz}, \tag{7}$$

where $\bar{b}(z)$ is the climatological reference profile of buoyancy in the area of the eddy. The *Ertel Potential Vorticity Anomaly* is then calculated on density surfaces (i.e. using density as the vertical coordinate) as follows

$$\Delta EPV(x,z) = EPV_x(x,z) + \Delta EPV_z(x,z), \tag{8}$$
$$\Delta EPV_z(\sigma) = EPV_z(\sigma) - \overline{EPV}(\sigma), \tag{9}$$

where $(x,z)$ are the coordinates on a ship section. As with thermohaline anomalies, this quantity is calculated on isopycnic surfaces and then represented on geopotential levels. This quantity has been widely used to define the materially coherent core of eddies and is therefore of interest (Zhang et al., 2014; Barabinot et al., 2024; Carton et al., 2010b).

Following the approach of Barabinot et al. (2024), we also define the ratio between the anomaly of the vertical component $\Delta EPV_z$ and the horizontal one $EPV_x$: $\Delta EPV_z/EPV_x$. In fact, it was shown that the eddy boundary is not locally defined and behaves like a frontal region subject to symmetric instabilities. These instabilities occur when the baroclinic term is not negligible compared to the vertical term (Hoskins, 1974; Hoskins and Bretherton, 1972). Consequently, a criterion of the type

$$\frac{\Delta EPV_z}{EPV_x} > \beta, \tag{10}$$

with $\beta \gg 1$ will detect the core water that is not in the turbulent frontal region. Symmetric instabilities can erode the core by changing the properties of the water parcels at the boundaries or by generating small scale turbulence (Thomas et al., 2016; D'asaro et al., 2011; Haine and Marshall, 1998; Goldsworth et al., 2021). This detected water is more stable and is subject to drift with the eddy without being altered by the environment.

### 3.4 Comparison between criteria

The goal here is to determine which of the criteria defined in the preceding section (thermohaline anomalies, gradients, EPV) is most effective in detecting the coherent core. Some criteria have already been studied by Barabinot et al. (2024). They showed that the eddy core is surrounded by a turbulent region subject to instabilities characterized by a value of $EPV_x/EPV_z$ close to 1. Consequently, the largest values of the ratio $\Delta EPV_z/EPV_x$ define the eddy core, which is less subject to instabilities and where the trapped water is less likely to be mixed and modified by the environment. By superimposing the thermal anomaly and the $\Delta EPV_z/EPV_x$ contours, we determine the materially coherent core, which should undergo little change in properties during the eddy drift. However, this criterion must be applied to the eddy core where the distinct water is retained.

To capture the true materially coherent core of an eddy, two criteria must be used. First, thermohaline anomalies on isopycnal surfaces must be computed to detect the region where the trapped water is located. The outermost closed contour is used to bound an approximate core. However, the boundary provided by thermohaline anomalies is only a line. But some water in its vicinity may cross it and escape the core due to instabilities. Therefore, the $\Delta EPV_z/EPV_x$ criterion is used within the first

region to remove the boundary region subject to instabilities. The last region is much more restrictive, but represents the stable confined water inside the core.

Ertel Potential Vorticity combines the stratification anomaly, the rotating flow, and the influence of the Earth's rotation. As a result, the boundaries determined by the thermohaline anomalies on isopycnals, the relative vorticity, and the buoyancy frequency drive those determined by Ertel Potential Vorticity.

In practice, it is difficult to apply the $\Delta EPV_z / EPV_x$ criterion to *in-situ* data because it requires high resolution data due to multiple spatial derivatives and is quite sensitive to noise. We now show that this criterion can be theoretically approximated by the buoyancy frequency.

In the region where $EPV_z / EPV_x \gg 1$, we have

$$EPV = (\zeta_z + f_0)\frac{\partial b}{\partial z}. \tag{11}$$

We then decompose the buoyancy field such that $b(x,z) = \overline{b}(z) + b'(x,z)$ where $\overline{b}$ is the climatological average and $b'$ the anomaly resulting from the eddy dynamics. Because $EPV_z / EPV_x \gg 1$, $\Delta EPV \approx \Delta EPV_z$. Following equation (9), on isopycnal surfaces, the anomaly of EPV is thus decomposed in three terms

$$\Delta EPV_z = f_0 \frac{\partial b'}{\partial z} + \zeta_z \frac{\partial b'}{\partial z} + \zeta_z \frac{d\overline{b}}{dz}. \tag{12}$$

Now, to analyze orders of magnitude, we have to keep in mind that the vertical scale for $b'$ will not be the same as $\overline{b}$. For $b'$, we take $H$ previously defined as $b'$, which is related to the isopycnals deviation. From Table 3, $H = 200$m. For $d\overline{b}/dz$, a typical order of magnitude in the ocean is $N_0^2 = 10^{-3}$ s$^{-1}$. Then, we use $\Delta b' = 1$ kg.m$^{-3}$ for a typical scale of $b'$ (Barabinot et al., 2024), $V$ for $V_o(x,z)$ and $R$ for the eddy radius. Dimensionless quantities are marked with a hat. By nondimensionalising $\Delta EPV_z$ with $f_0 \Delta b'/H$, we obtain

$$\widehat{EPV} = \frac{\partial \hat{b}}{\partial \hat{z}} + Ro\,\hat{\zeta}_z \frac{\partial \hat{b}}{\partial \hat{z}} + Ro\,N_0^2 \frac{H}{\Delta b'} \hat{\zeta}_z \frac{d\hat{\overline{b}}}{d\hat{z}}. \tag{13}$$

where $Ro = V/(f_0 R)$ is the Rossby number for an axisymmetric vortex. For mesoscale eddies, $Ro < 1$ and even $Ro = O(0.1)$. By construction, $\hat{\zeta}_z$ is of order one. So the second term is always smaller than the first one. Then $N_0^2 H/\Delta b' = 0.2$ which is smaller than one and finally, the third term is also dominated by the first one. Therefore, the buoyancy frequency anomaly defined in (4) is a good proxy for our criterion. We confirm what was found using *in situ data* from Meunier et al. (2021). Note that $f_0 \partial_z b'$ has already been considered as a PV anomaly by previous studies (Paillet, 1999; Paillet et al., 2002).

## 4  Methods to compute eddies volume

There are many methods in the literature to approximate and calculate mesoscale eddy volumes. This step is critical for estimating the tracer transported by these structures. For example, some altimetric studies have used cylinders to approximate eddy cores even when the true vertical structure is unknown (Fratantoni et al., 1995; Johns et al., 2003; Bueno et al., 2022). Lagrangian studies are also very powerful for estimating tracer transport using Lagrangian criteria (Hadjighasem et al., 2017).

However, as mentioned in the introduction, it is impossible to perform temporal studies with *in situ* data. In this section, we describe two reconstruction methods to estimate eddy volumes from a single ship section.

## 4.1 Geometric Considerations

Consider an eddy whose boundaries are defined by a criterion (a given isoline of temperature/salinity anomaly, EPV, gradients, etc., see Barabinot et al. (2024)). This eddy was sampled by a ship transect that does not necessarily cross the real eddy center, defined as the location of the zero velocity. Therefore, the difference between the exact eddy center and the center on the resulting 2D section will affect the reconstruction of the 3D structure and thus the volume.

To illustrate this fact, consider a perfect cylindrical vortex core with radius $R$ and height $H$. We assume that it is located at the ocean surface and that it has been sampled by a ship track as shown in Figure 1, so that $L$ appears as the eddy radius on the 2D vertical section. An estimate by a simple calculation of the eddy volume using this 2D vertical section gives a volume of $\pi L^2 H$, which has to be compared with the real volume of $\pi R^2 H$. Using the Pythagorean theorem, it can be shown that the relative error, expressed as a fraction of the exact volume, is $e^2/(2R^2)$, assuming $e \ll R$. The relative error is less than 5 % if $e \leq R/\sqrt{10} \approx 0.316R$. In this case, $e$ must be less than $31.6$ % of $R$ for this condition to be true. This condition is not really restrictive, and the reconstruction can be quite faithful.

If we now assume that the eddy is cone-shaped with a base of radius $R$ and height $H$, the relative error is different. Assuming that the eddy was sampled by a ship's cruise, as in Figure 1, the boundary of the eddy will appear as a hyperbola of maximum height $H_e$ on the 2D vertical section. Now the eddy will appear to be less deep than it is in reality. The relative error between the exact and reconstructed volumes will be $3e/R$. This result follows only from basic geometric considerations (see Figure 2). In this case, for the relative error to be less than 5 %, $e$ must be less than $1.7$ % of the eddy radius, which is very restrictive. Given the horizontal resolution of the data, and thus the uncertainty in the radius, the reconstruction method will be highly inaccurate.

Therefore, depending on the shape of the eddy, the distance between the ship track and the eddy center $e$ is a critical parameter and strongly influences the uncertainty of the volume approximations. To ensure an accurate estimation of volume, we have computed the values only for eddies with a very small value of e. This approach helps us minimize the potential uncertainty in the computed volumes. Our database includes only four eddies ($N°1, 2, 7$, and $24$) that have been sampled by a ship track crossing the eddy within a very small distance from its center (i.e., with $e < 3$km). These eddies are suitable for computing volumes due to their proximity to the center.

Different idealized volumes can be calculated analytically, and the same approach can be followed for subsurface eddies. As shown in previous studies, surface eddies appear to have shapes close to cylindrical or conical volumes (not necessarily with a circular basis), but some approximations exist for subsurface eddies. Some of them described eddies EPV anomaly as pancakes because the horizontal scale is much larger than the vertical one (Bars et al., 2011). In reality, however, an eddy has a more complex shape, depending on the criterion used to define its boundaries. It is not perfectly axisymmetric and its rotation axis is not perfectly vertical. More precisely, the shape is determined by the rotating flow and depends on the deformation that the vortex undergoes. It can be stretched and sheared by the mean background flow. It has been shown that the flow function

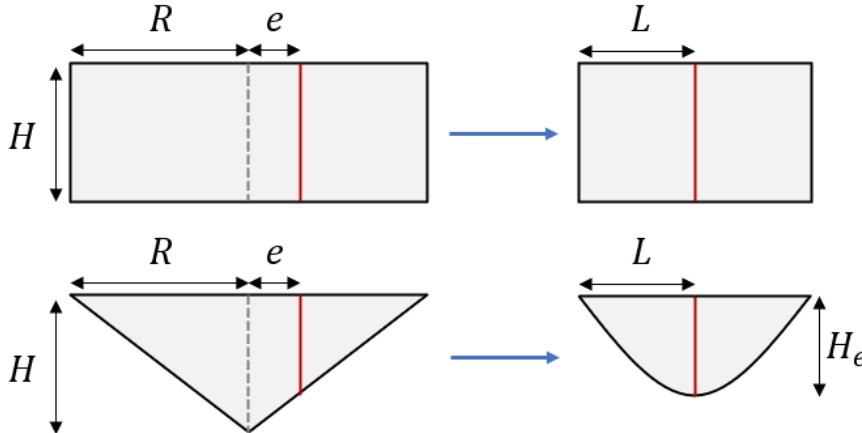

**Figure 2.** Simple approximation using a ship cross section: an eddy is a solid of revolution (cylindrical at the top, conical at the bottom). On the left is the real eddy core, bounded by a criterion. On the right, the reconstruction based on the ship section. The dashed gray line is the position of the eddy center, which does not vary, and the red line is the perfectly vertical section. For clarity, only a 2D view is shown, but each volume is axisymmetric.

of the rotating flow can be decomposed into azimuthal normal modes (Gent and McWilliams, 1986). Depending on the order of the modes, the flow pattern is modified. If the eddies are strongly disturbed, the decomposition of the flow function into normal modes may include high order terms. In most cases, however, three modes dominate: order 0, which corresponds to a purely circular eddy, order 1, which is the dipolar mode typical of self-propagating eddies, and order 2, which corresponds to an elliptical eddy (Carton, 2001; de Marez et al., 2020). In this context, we propose two approaches to approximate the volume

(associated with a criterion) of an eddy sampled by a ship section, assuming first mode 0 and then mode 2 are dominant. Both approaches use the $f-$plane approximation. Both reconstructions are thus performed in a Cartesian space, neglecting the local curvature of the sea surface.

### 4.2   Reconstruction using cylinders with a circular base

The methodology is illustrated in Figure 3. We now reconstruct the 3D structure of an eddy using the same approach as in

Figure 2, but we take into account its vertical tilt. The eddy remains perfectly circular at each geopotential level, its center being the one given by the ship's section. The total volume is the sum of the volumes of the elementary cylinders.

This method preserves the variation of the eddy radius with depth and the variation of the eddy rotation axis on the vertical. This reconstruction is also relatively straightforward. However, it assumes that the eddy is perfectly circular at each geopotential level, which is a strict hypothesis. Also, the center is that of the 2D ship section, and the calculation of the volume does not

depend on $e$, although we have shown that it has an influence. In summary, the approach consists of three steps. First, a criterion (the outermost closed contour of a given size) is chosen to delimit the materially coherent eddy core from its surroundings on

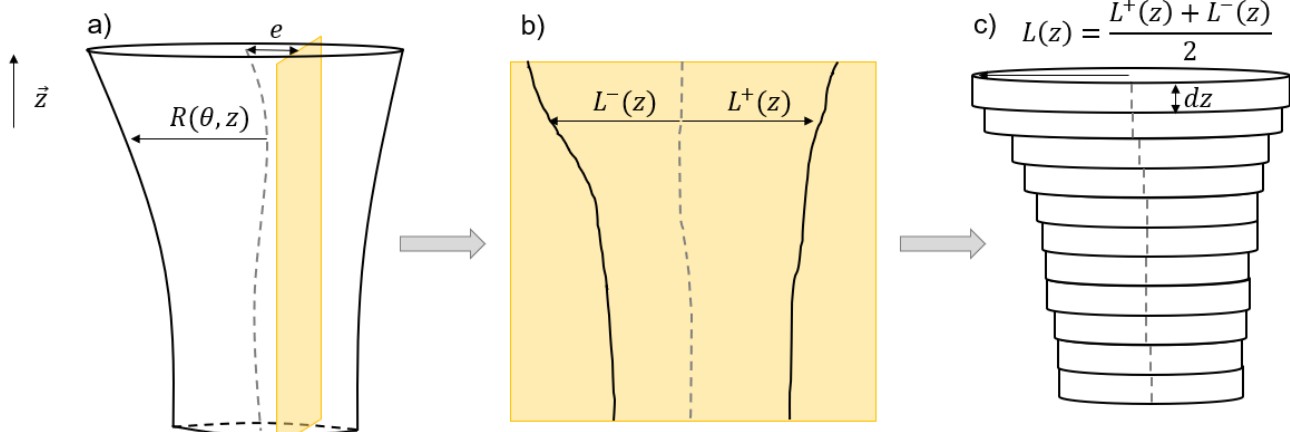

**Figure 3.** Methodology for reconstructing the 3D structure of an eddy from a single ship track. Here, a surface eddy was used, but the approach also works for a subsurface eddy. a) Real surface eddy, for which the volume is defined by a criterion: the real eddy center is represented by a dashed gray line and the sampled vertical section in yellow. The eddy is not axisymmetric and its radius is a function of the cylindrical variables $\theta$ and $z$. This structure has been sampled by a yellow vertical ship track characterized by the distance $e$ from the real eddy center. b) Vertical section where the boundary is estimated by the same criterion: here the dashed grey line represents an approximation to the real eddy center. To be consistent with the previous notation, the radius of the vortex is denoted $L$. Since the eddy is not symmetric, we differentiate the radius associated with the positive and negative poles of the velocity field (even if the criterion is not based on velocity). c) The 3D shape of the eddy is reconstructed as an association of infinitesimal cylinders of radius averaged between $L^+$ and $L^-$ and of small height $dz$. The total volume can be calculated by summation. The center of each small cylinder is that of the 2D vertical section and thus remains in the plane of the ship section.

the 2D vertical slice. Then, compute the position of the apparent eddy center as the location where the orthogonal velocity $V_o$ is zero and the eddy radius $L(z)$ associated with the selected criterion. Finally, calculate the approximate volume as a sum of elementary cylinders.

This method defines the uncertainty due to resolution:

$$\frac{\delta\Omega}{\Omega} = \frac{\int_{-H-\Delta z}^{0} \pi(L(z)+\Delta x)^2 dz}{\int_{-H}^{0} \pi L^2(z) dz} - 1, \tag{14}$$

where $\Omega$ is the approximated volume, $\Delta x$ is the horizontal resolution, and $\Delta z$ is the vertical resolution (depending on the type of device). This formula is valid for a surface eddy. In the subsurface case, the integral must be replaced by $\int_{-\frac{H+\Delta z}{2}}^{\frac{H+\Delta z}{2}}$.

     By employing a comparable methodology and making use of certain geometrical considerations, we are able to extrapolate

the eddy volume using elliptically based tubes. Please refer to Appendix B for a detailed description of the methodology. This methodology enables the construction of two possible elliptically based tubes from a single ship section. On a ship section, the eddy center separates the core into two parts, which are then used to determine the volumes through the application of two ellipses, designated $(E_1)$ and $(E_2)$. The resulting volumes are determined by the left or right sides of the ship section,

respectively. As the vertical shape of eddies is not well understood, especially the shape of their thermohaline coherent core,
in the literature, we present the two ellipses as examples of what an eddy core can look like in 3D.

## 5    Results

### 5.1    Distinct waters in eddy cores

For each mesoscale eddy, thermohaline anomalies on the isopycnals have been computed using the methodology described in
Section 3.1. Examples of anomalies computed for some eddies are shown in Figures 4 and 5. We inform the reader that all
other vertical sections can be found in the supplementary material, Figures S1 to S16. Both salinity and temperature anomalies
are calculated for each eddy.

For the subsurface AE sampled in the Lofoten Basin ($N°$ 24 in Table 2), a significant thermohaline anomaly is visible in the
middle of the temperature and salinity panels between $-700$ m and $-1150$ m depth. The location of this anomaly coincides
with the maximum isopycnal anomaly, indicating that it corresponds to the eddy core. The trapped water is warmer and fresher
than the climatological average. Compared to the surrounding water, the trapped water appears warmer and saltier.

A distinct negative anomaly can be observed in the vertical sections of the subsurface AE sampled during EUREC4A-OA
($N°$ 2). This eddy transports water that is fresher and colder than the surrounding water. In the case of the surface AE sampled
during Physindien 2011, the warmer and saltier core is located at $x \approx 470$ km and is surrounded by colder and less salty
water that forms a rim around it. The subsurface cyclone sampled during M124 also shows anomalies in the region where the
isopycnals show the greatest anomaly. Water that is hotter and saltier than its surroundings is trapped in the eddy core. However,
the core is less well localized than in other examples, suggesting either that the eddy is losing water through instability and
filamentation, or that it is not well resolved in terms of horizontal resolution of vertical thermohaline properties.

In Figure 6 the thermohaline anomalies on isopycnals are collected for each eddy. The anomalies are computed with respect
to climatological averages, but especially for Figure 6 the maximum value between the eddy core and the surrounding is
computed using as boundary the outermost closed contour of $\Delta T$ and $\Delta S$ (see Figure 10 for examples). The maximum values
of the anomalies represent the difference between the properties of the potential trapped water and the surrounding water. An
eddy is considered to be TC when the maximum anomaly is reached at the eddy center (region where the velocity tends to
zero) and there is a marked difference in values between the trapped and surrounding waters.

According to the data, 18 out of 25 eddies have a significant thermohaline anomaly on isopycnals in their core, which means
higher than the climatological standard deviation in the considered region. Thus, 72 % are found with a significant anomaly
in their core and are observed to transport distinct water in their core. Even eddies sampled far from their origin show an
anomaly in their core (see Agulhas rings $N°15$, 16). Others have no significant difference in values between the enclosed and
surrounding waters. Note that, AE are not automatically associated with positive temperature anomalies and, conversely, CE
are not always associated with negative temperature anomalies.

A key point here is that eddies contain water characteristic of their region of formation. For example, Sandalyuk et al. (2020)
showed that the subsurface AE $N°24$ (Figure 4) was generated by baroclinic instability of the Norwegian Atlantic Slope

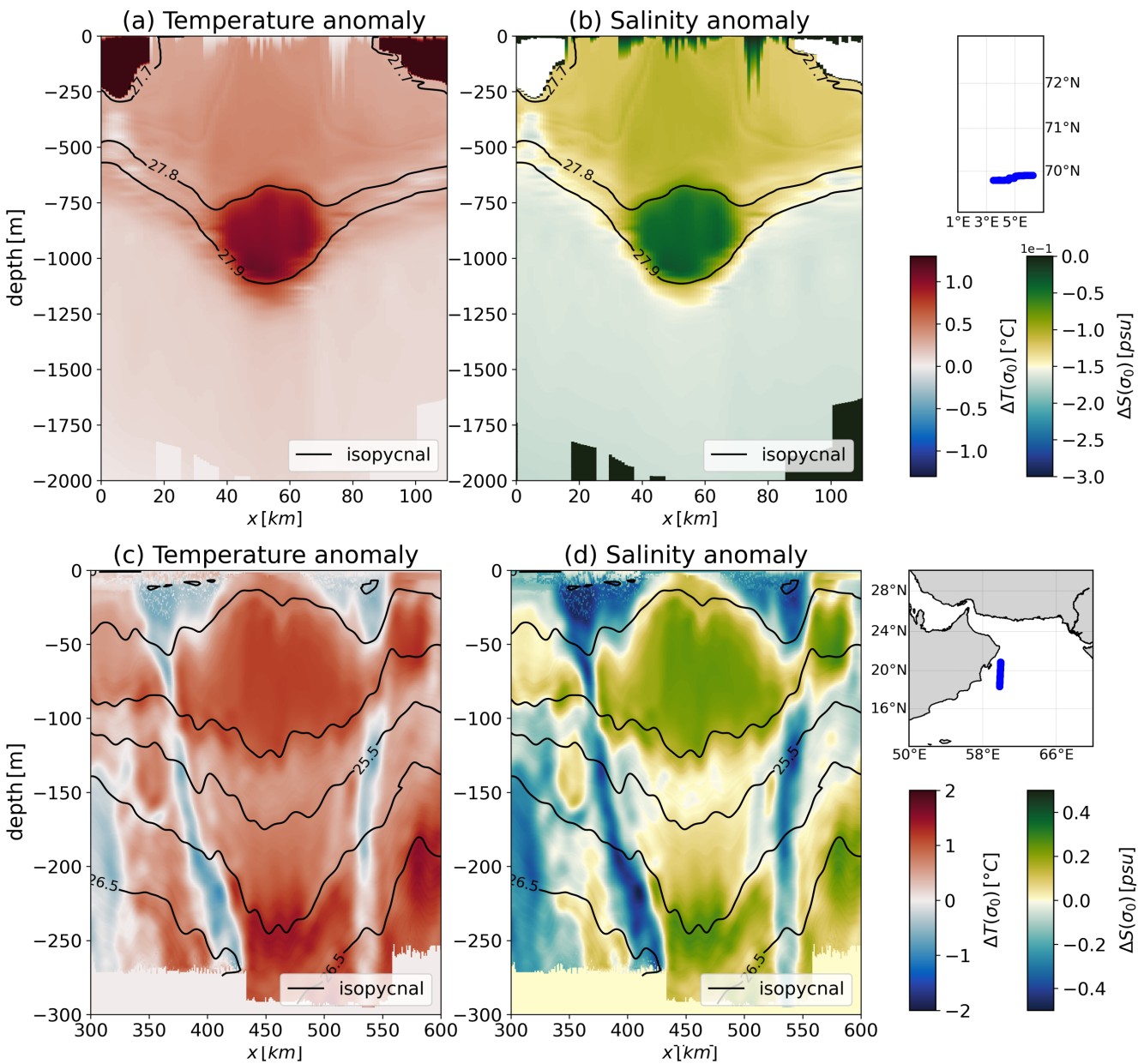

**Figure 4.** Thermohaline anomalies on isopycnals computed for mesoscale eddies: (a-b) the Lofoten Basin anticyclone ($N°24$) and (c-d) the Persian Gulf anticyclone dipole ($N°7$). For each eddy, three panels are shown: both temperature (a-c) and salinity (b-d) anomalies, and a small map showing the transect (in blue) along which the eddy was sampled. For panels showing anomalies, the abscissa axis is the horizontal scale in km and the ordinate axis is the depth in m. Isopycnals are shown in black. The white bands near the bottom indicate where the data ends.

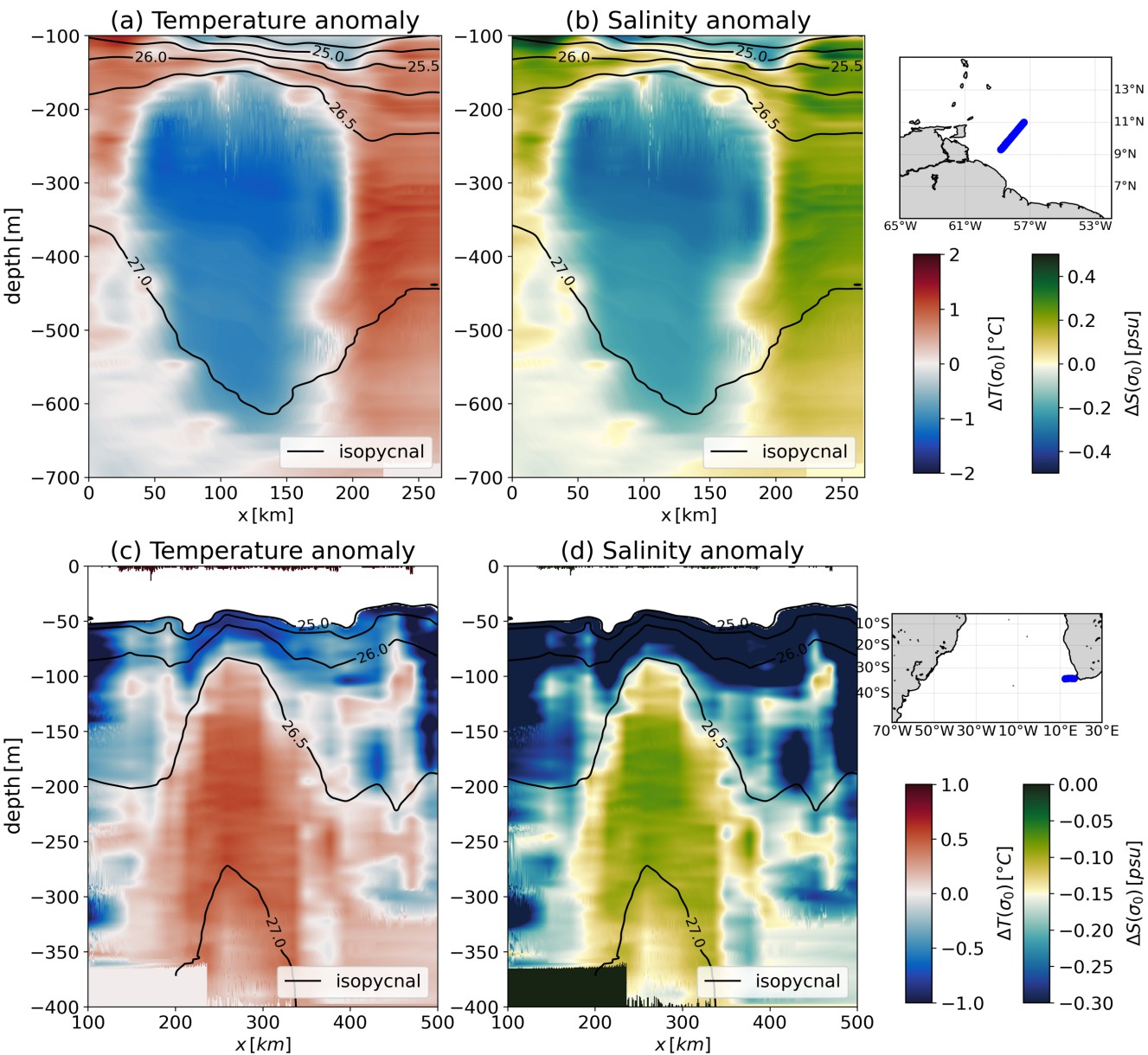

**Figure 5.** Thermohaline anomalies on isopycnals computed for mesoscale eddies: (a-b) the North Brazil Current anticyclone ($N°2$), and (c-d) the Southern Cape Basin cyclone ($N°9$). For each eddy, three panels are shown: both temperature (a-c) and salinity (b-d) anomalies, and a small map showing the transect (in blue) along which the eddy was sampled. For panels showing anomalies, the abscissa axis is the horizontal scale in km and the ordinate axis is the depth in m. Isopycnals are shown in black. The white bands near the bottom indicate where the data ends.

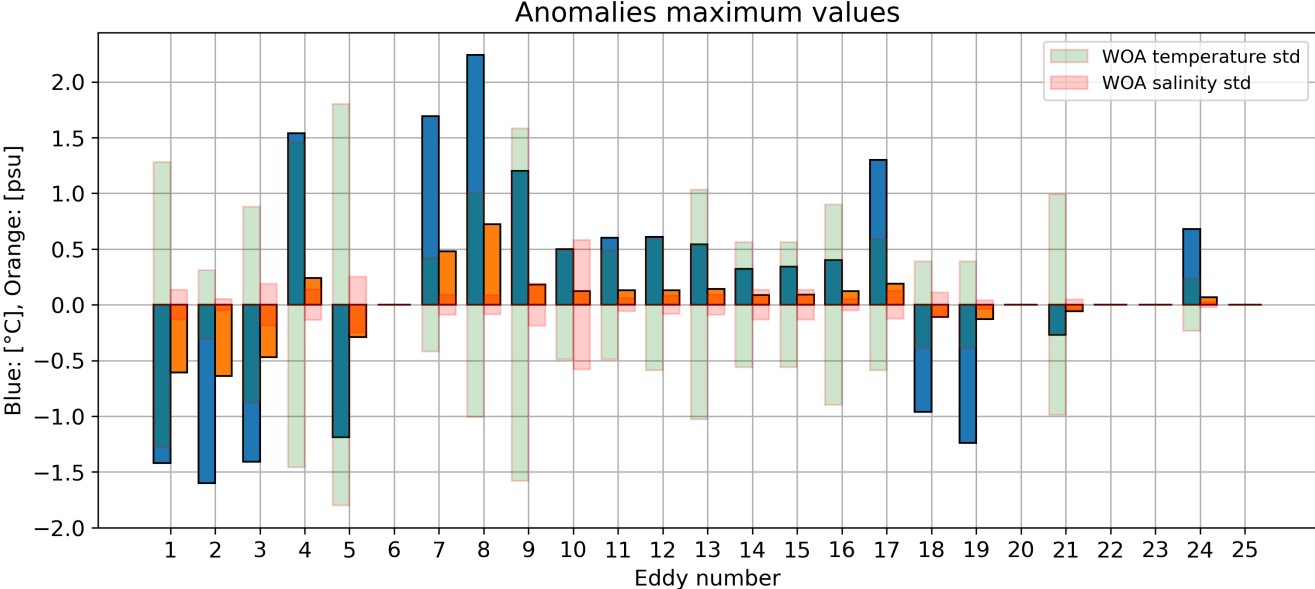

**Figure 6.** Maximum values for temperature (blue bars) and salinity (orange bars) anomalies on isopycnals (anomalies calculated with respect to the climatological mean). These values are obtained in the eddy cores and compared to the climatological standard deviation of temperature and salinity in each region computed by WOA 2023 at the depth where the maximum is reached (red bars). If there is no clear maximum in an eddy core, the enclosed water is not different from the surrounding water; no bar is shown: the eddy is then considered to be not TC. Note that the presence of the eddy center in a vertical section is not required to evaluate the MC.

Current which flows along the Norwegian Atlantic coast. Due to the $\beta-$effect, its westward propagation results in heat and salt transport to the central part of the Lofoten Basin. The trapped water coming from the Norwegian coast appeared warmer and saltier than the fresh and cold water of the Lofoten Basin resulting in positive anomalies in the eddy core. L'Hégaret et al. (2016) showed that the AE sampled in the Arabian Sea $N°7$ (Figure 4) transports Persian Gulf Water. In this region, this high-salinity water spreads into the Sea of Oman via the Strait of Hormuz under the influence of energetic mesoscale eddies. Eddy $N°7$ is one of them. Subirade et al. (2023) showed that the water transported by the subsurface AE $N°2$ (Figure 5) comes from the North Brazil Current retroflection region which appears colder and fresher than the surrounding waters. Finally, Laxenaire et al. (2020) showed that Algulhas rings transport water from the Mozambique Chanel to the South Atlantic Ocean. The water trapped by CE $N°9$ is thus hotter and saltier than the surrounding water.

### 5.2 Location of different water bodies

Figures 7 and 8 present a comparison of sampled eddies with eddies identified by the TOEddies algorithm using satellite altimetry. Our comparison is qualitative, as we are primarily interested in the surface or subsurface character of eddies. We will leave the quantitative aspects to a future study. These figures focus on 17 well-sampled eddies that provide important

information. Please note that eddies 3 and 8 are not included in the analysis as they are subsurface intensified eddies that lie below the main thermocline. For more detailed information on these eddies, please refer to the supplementary material.

Let us first examine panel (a) in Figure 7. This panel is about the subsurface eddy $N°2$, which is also shown in Figure 4. TOEddies detects this eddy as an anticyclone. The TC core of the vortex (location of the anomaly) is below $-150$ m and the velocity field tends to zero at this geopotential level. It is important to note that while the TOEddies algorithm successfully detects an AE, it does not correspond to a surface intensified eddy. This aligns with findings in analogous cases discussed in Laxenaire et al. (2018); Subirade et al. (2023). Therefore, knowledge of an eddy's vertical structure is crucial for assessing its characteristics and classification.

Panel (b) in Figure 7 presents another example of an eddy, the eddy $N°24$, which is also shown in Figure 4. As in the previous example, the ADT signature of the eddy corresponds to the actual sampled eddy. In this case, the ADCP velocity field at the surface is not zero. However, the TC core is located at approximately $-1000$ m depth. In fact, it is not possible to determine from satellite altimetry alone whether a given feature is a surface or subsurface intensified eddy. This is similarly the case for panels (c), (d), (e), (f) in Figure 7 and (a), and (b) in Figure 8. Some eddies give rise to a surface dynamic topography signal discernible in ADT maps. However, the distinct water that is trapped within them is situated at a considerably deeper level. For further details, please refer to the supplementary material, which illustrates thermohaline anomalies on isopycnals for each eddy discussed in this article.

Our data set indicates that the maximum thermohaline anomaly is often found at depth, rather than at the surface. This is also true for eddies that have been identified by satellite altimetry. By limiting the analysis to geostrophic velocity fields derived from satellite altimetry or other surface properties, the resulting eddy assessments lack the vertical properties of eddies. This is also the case for eddies that have been identified through satellite altimetry. Lagrangian studies suggest that the ability of eddies to trap a water mass is a consequence of closed flow trajectories (Beron-Vera et al., 2013; Haller et al., 2015). It should be noted, however, that such trajectories cannot be calculated from surface velocity fields alone. As has been previously discussed, a considerable number of eddies are found to be intensified in the subsurface. Moreover, satellite altimetry has a limited horizontal resolution in comparison to the dimensions and varying velocities of eddies. Consequently, integrating the geostrophic velocities derived from satellite altimetry introduces a bias in the diagnostic of eddy material coherence provided by Lagrangian estimates of water parcel trajectories. Indeed, numerous eddies that were previously classified as incoherent have been found to exhibit coherence when their full vertical extent is taken into account. Therefore, our study underscores the inherent limitations of relying on satellite altimetry or any surface field to ascertain eddy characteristics.

Consequently, the accuracy of tracer transport estimates is contingent upon the manner in which eddies are observed and characterized. It should be noted that the proportion of thermohaline subsurface intensified eddies indicated by our *in situ* dataset is 60.7%. Even if the number of surface intensified eddies is underestimated due to the fact that *in situ* velocity measurements often sample only the ocean below $-50$ m depth, this ratio serves to highlight the ubiquity of subsurface eddies. Furthermore, it highlights the inherent bias of studies based solely on satellite altimetry. This indicates a significant discrepancy between the surface geostrophic velocity derived from satellite altimetry and the velocity of the eddy core. This is exemplified

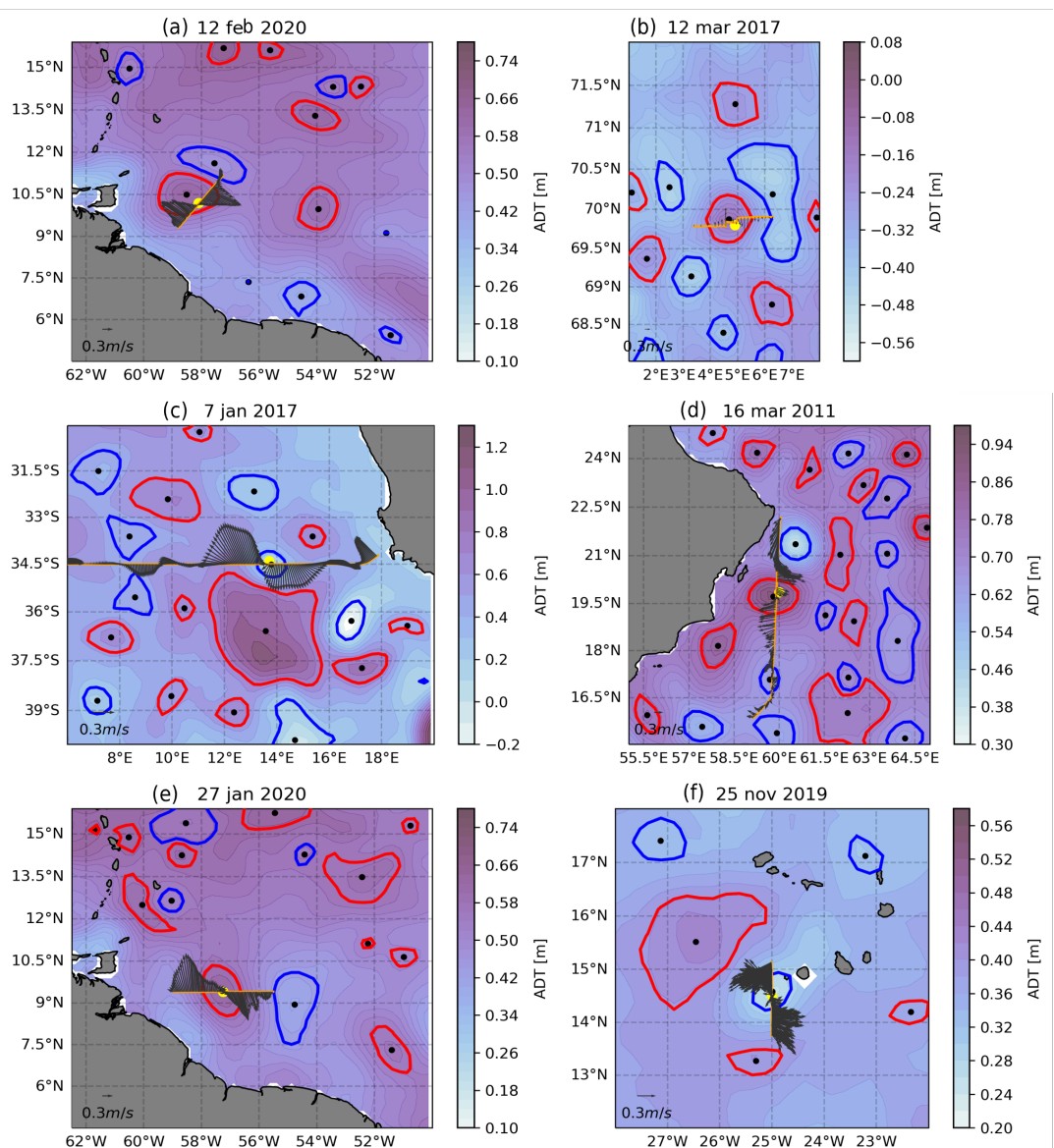

**Figure 7.** Comparison between satellite altimetry data and *in situ* data for some eddies. Each panel shows the same elements: ADT [m] as colored background, AE (red contours) and CE (blue contours) detected by the TOEddies algorithm, eddy centers (dark dots) also detected by the TOEddies algorithm, the ship track in orange, velocity vectors at a given depth in gray (the legend is given for each panel), and the eddy centers estimated at this depth level using the Nencioli et al. (2008) routine in yellow dots. Panel (a): AE $N°2$ and velocity field at $-300$ m depth. Panel (b): AE $N°24$ and velocity field at $-900$ m depth. Panel (c): CE $N°4$ and velocity field at $-50$ m depth. Panel (d): AE $N°1$ and velocity field at $-50$ m depth. Panel (e): AE $N°7$ and velocity field at $-50$ m depth. Panel (f): CE $N°23$ and velocity field at $-50$ m depth.

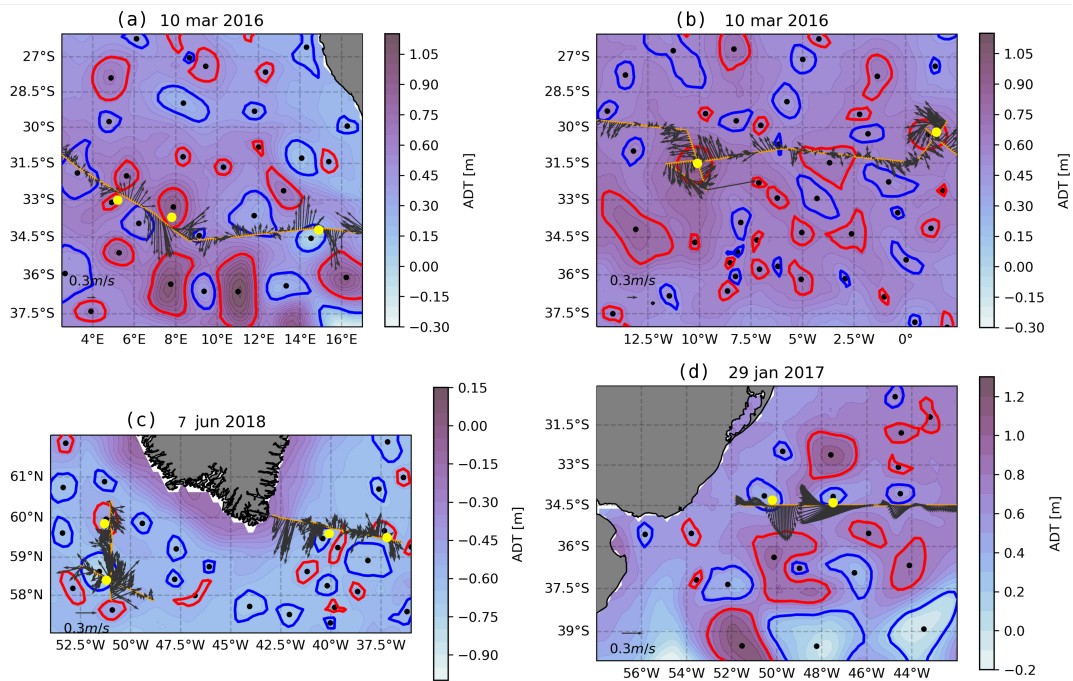

**Figure 8.** Comparison between satellite altimetry data and *in situ* data for some eddies. Each panel shows the same elements: ADT [m] as colored background, AE (red contours) and CE (blue contours) detected by the TOEddies algorithm, eddy centers (dark dots) also detected by the TOEddies algorithm, the ship track in orange, velocity vectors at a given depth in gray (the legend is given for each panel), and the eddy centers estimated at this depth level using the Nencioli et al. (2008) routine in yellow dots. Panel (a): AE $N°10, 11, 13$, CE $N°9$ and velocity field at $-100$ m depth. Panel (b): AE $N°17$ and velocity field at $-100$ m depth. Panel (c): CE $N°5, 6$ and velocity field at $-50$ m depth. Panel (d): AE $N°18, 22$ and CE $N°20, 21$ and velocity field at $-50$ m depth.

by eddies N° 1, 4, 7, 23, and 24 in Figure 9. Furthermore, in cases where the overlying water is well stratified, the subsurface
eddies may be entirely undetectable in altimetry fields. This is illustrated by AE N°2 in Figure 7.

In conclusion, a typical correlation can be observed between eddy velocity and thermohaline anomalies on isopycnals. However, a notable proportion of eddies identified by satellite altimetry are subsurface intensified, exhibiting a deep maximum of velocity and thermohaline anomalies. The presence of these eddies introduces a significant degree of uncertainty into the estimation of tracer transport based on satellite altimetry data alone.

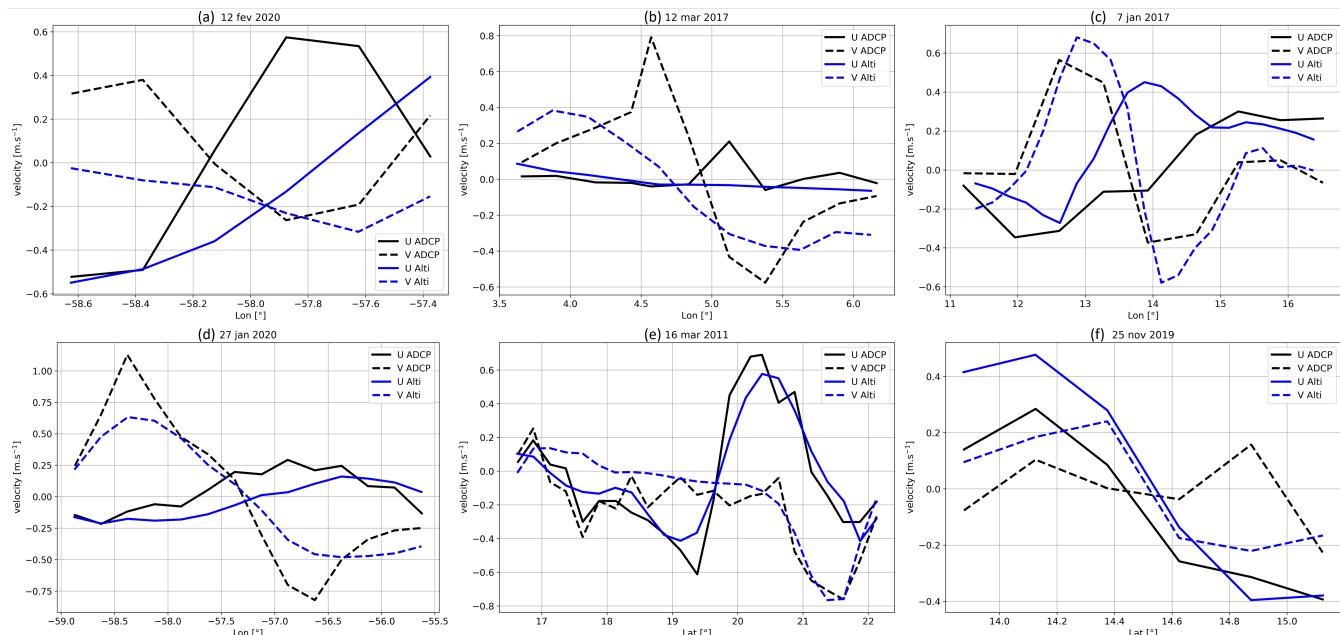

**Figure 9.** Velocity measured in situ using an ADCP at the location where the velocity is maximum. This comparison is shown in Figures 7 and 8 for some eddies. The symbols $U$ and $V$ are used to represent, respectively, the zonal and meridional velocity. The comparison was conducted by interpolating ADCP data and satellite data onto the same grid along each ship track. The panels are identical to those presented in Figures 7. Panel (a): AE $N°2$ and *in situ* velocity field at $-300$ m depth. Panel (b): AE $N°24$ and *in situ* velocity field at $-900$ m depth. Panel (c): CE $N°4$ and *in situ* velocity field at $-50$ m depth. Panel (d): AE $N°1$ and *in situ* velocity field at $-50$ m depth. Panel (e): AE $N°7$ and *in situ* velocity field at $-50$ m depth. Panel (f): CE $N°23$ and *in situ* velocity field at $-50$ m depth

## 5.3 Volume estimates

### 5.3.1 3D Eddy Boundary Characterisation

For TC eddies, our ultimate goal is to calculate their volume to quantify their contribution to tracer transport. As mentioned in the Methodology section, it is difficult to calculate the eddy volume with a single ship section; moreover, this calculation depends on the criteria used to delimit the core.

In this section, the eddy volume calculated in this way is analyzed along with 6 eddy core boundary criteria: Thermohaline anomalies on isopycnal surfaces (see equations (1) and (2)), relative vorticity (equation (5)), Brunt Väisälä frequency anomaly (equation (4)), norm of the 2D buoyancy gradient (equation (3)), $EPV$ anomaly (equation (9)), and the ratio $\Delta EPV_z/EPV_x$ (equation (10)). Depending on the data resolution and noise, some criteria may not be applicable.

Here three well-sampled AE ($N°1$, $7$ & $24$, denoted $C^+$ in table 3) have been selected for which the 6 criteria can be applied. Eddy $N°1$ (the surface AE sampled during EUREC4A-OA) and eddy $N°7$ (the surface AE sampled during Physindien 2011)

have the finest horizontal resolution, so the uncertainties are small. Eddy $N°24$ (the subsurface AE sampled in the Lofoten Basin) has a sharp boundary; although its sampling is not optimal, its structure raises interesting questions.

The methods presented are carefully followed. Figure 10 shows the vertical section of the ship overlaid with closed contours defined by the criteria for the 3 eddies considered. For the sake of clarity, the quantities used to draw the contours are calculated only in the vicinity of the core. In reality, due to the noise in the data, these criteria can also detect other features not related to the eddy core. In the background, the quantity $\Delta EPV_z/EPV_x$ is plotted. The eddy volume is insensitive to the threshold chosen for $\Delta EPV_z/EPV_x$ because its gradient is very pronounced at the eddy boundary. The difference in the eddy volume when choosing levels 10 or 30 is less than 3 %. However, this threshold must be greater than 10 for $EPV_x$ to be negligible before $\Delta EPV_z$.

As an example, in panel (a) this criterion highlights the deep core of the eddy between $-650$ m and $-1050$ m. Above this core, for $\sigma \in [27.7; 27.8]$ kg.m$^{-3}$, the quantity $\Delta EPV_z/EPV_x$ decreases slightly: this marks the upper boundary of the core. Below this core, where $\sigma > 27.88$ kg.m$^{-3}$, the quantity $\Delta EPV_z/EPV_x$ decreases rapidly to values below 5, forming the lower vortex boundary. The lateral eddy boundary is characterized by $EPV_x \approx \Delta EPV_z$, indicating that it is subject to symmetric instability.

This key finding is supported by the other five criteria. The region where $\Delta EPV_z/EPV_x > 30$ is consistent with the region where: thermohaline anomalies on isopycnals reach an extremum; the core is quite homogeneous according to the density gradients and is associated with a significant anomaly of potential vorticity. However, the relative vorticity seems to be less relevant for the detection of the upper and lower core boundaries. Since this criterion considers only the velocity field, it does not distinguish TC regions from others. As a result, the approximated volume appears much larger than that determined by the other criteria.

It is worth noting that the region where $\sigma < 27.7$ kg.m$^{-3}$ is also characterized by the $\Delta EPV_z/EPV_x > 30$ criterion, although the TC core appears to lie below it. In fact, since $EPV$ lies on buoyancy gradients, a non-TC region can be highlighted by buoyancy gradients created by isopycnal deviations. This shallower region is also consistent with the region where $\zeta_z < 0$.

Similar observations can be made for panels (b) and (c). As mentioned in section 3.4, the criterion based on $\Delta EPV_z/EPV_x$ is only efficient in regions where distinct water is trapped.

### 5.3.2   3D eddy reconstruction

In this section, methods for approximating eddy volumes are applied to the three eddies considered, but results are shown only for the AE of panel (a) in Figure 10. The eddy shapes are discussed before the numerical aspects are presented.

Figure 11 shows the 3D reconstructions assuming circularity of the eddy at each geopotential level. Since the position of the center does not vary with depth, the eddy is axisymmetric. The reconstructed volume associated with the thermal anomaly is the most connected of all shapes. The eddy shape using the relative vorticity criterion is almost cylindrical and its upper and lower boundaries cannot be clearly distinguished. On the contrary, any other criterion leads to an eddy radius that decreases near the upper and lower boundaries: the volume is closed. Using the criterion on the norm of the 2D density gradient gives a

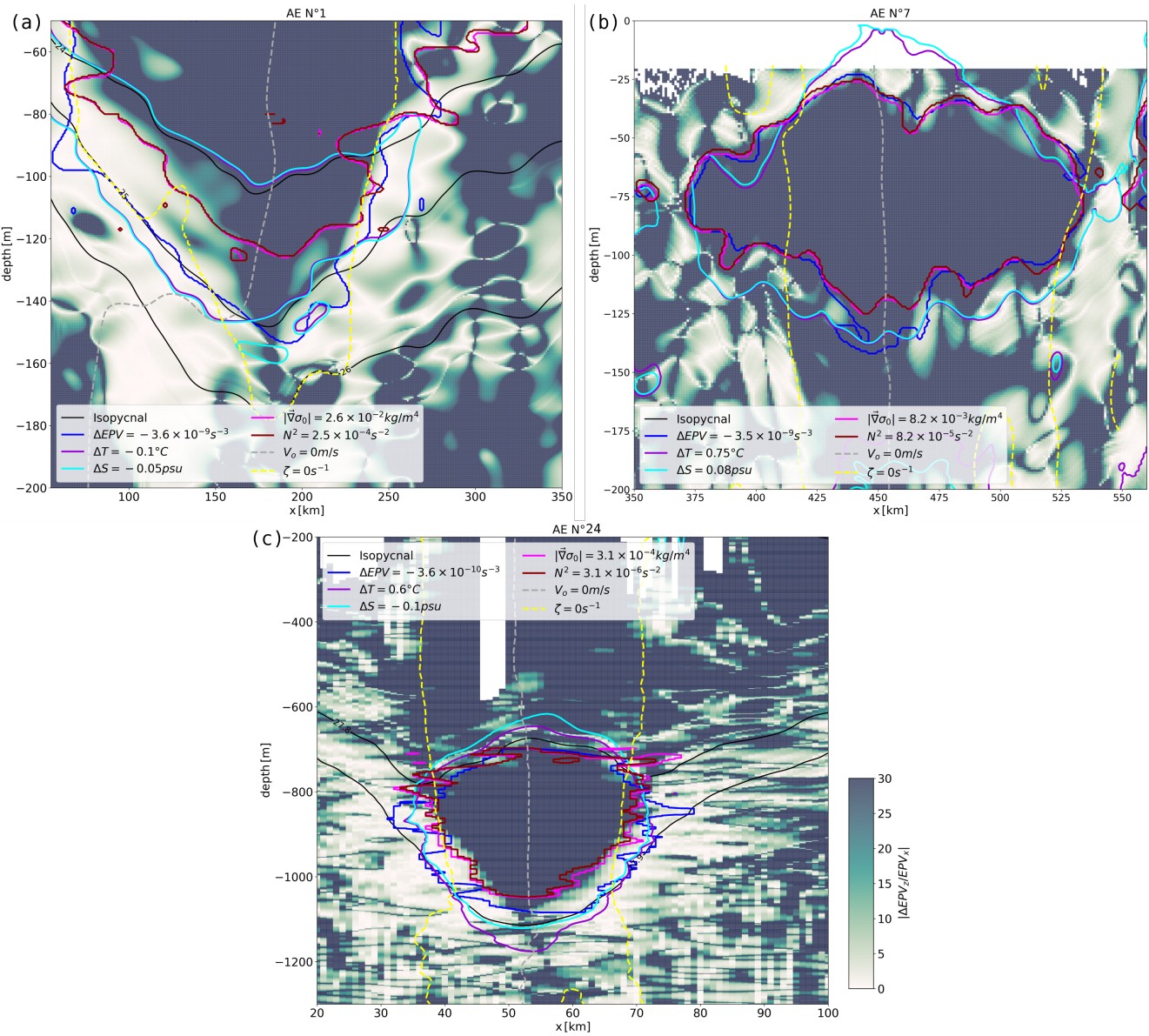

**Figure 10.** Outermost closed eddy contours computed using 5 criteria: thermal anomalies on isopycnal surfaces in purple, salinity anomalies on isopycnal surfaces in cyan, relative vorticity in dashed yellow, Brunt-Väisälä frequency in brown, density gradient norm in pink, $EPV$ anomaly in blue. The $\Delta EPV_z / EPV_x > 30$ criterion in the background is also able to capture the stable core of eddies 1 (panel (a)), 7 (panel (b)), and 24 (panel (c)). The color associated with this quantity has been saturated at level 30 to capture the region of weak frontality. The apparent eddy center is shown as a dashed gray line, the isopycnals as thin dark lines. The horizontal smoothing periods for panels (b) and (c) have been increased to 30 km so that the boundaries appear clearly.

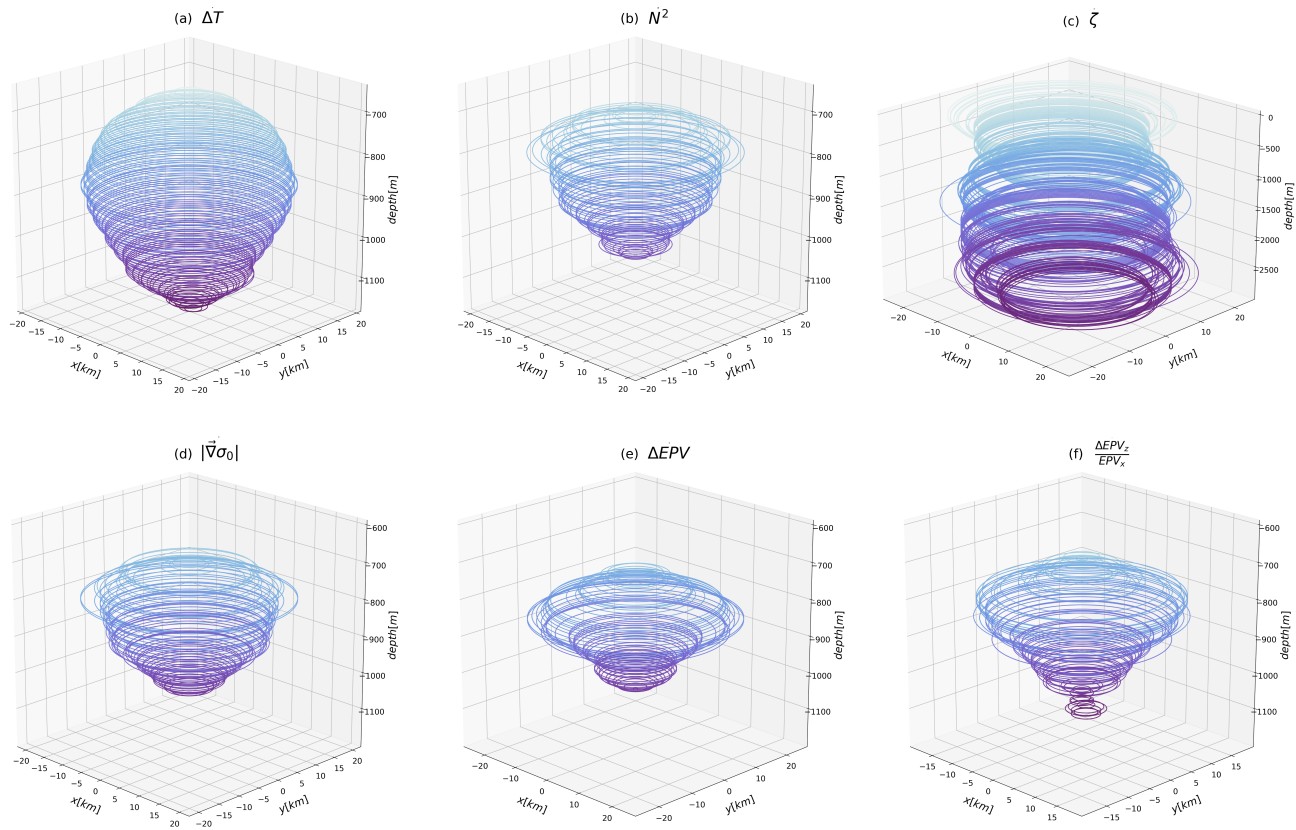

**Figure 11.** 3D reconstructions of AE $N°24$ assuming its circularity at each geopotential level. Each panel corresponds to one criterion. The criteria are detailed in Figure 10. (a): Thermal anomaly on isopycnals, (b): Brunt-Väisälä frequency, (c): relative vorticity, (d): norm of 2D density gradient, (e): Ertel potential vorticity anomaly, (f): $\Delta EPV_z/EPV_x$. Contours are plotted every five meters.

similar shape to the Brunt-Väisälä frequency criterion. Except for the relative vorticity criterion, the eddy core is top shaped.

The $\Delta EPV_z/EPV_x$ criterion results in a more conical eddy than the gradient based criteria.

Figure 12 shows the 3D reconstructions assuming the vortex core is elliptical at each geopotential level. For $N°1$ the eccentricity is set to $0.782$, for $N°7$ the value of $0.780$ is kept, and for $N°24$ the value of $0.792$ is kept. This Figure refers to the ellipses $(E1)$ mentioned earlier: the left side of the core was used to construct the volume. Again, the relative vorticity criterion leads to a cylindrical vortex shape. For all other criteria, the eddy base is thinner than for circular eddies (see Figure

11). This is consistent with Figure 10, where the eddy base radius is smaller on the left than on the right. As before, criteria based on the Brunt-Väisälä frequency or on the norm of the 2D density gradient give eddy shapes similar to those with the $\Delta EPV_z/EPV_x$ criterion.

Figure 13 shows the 3D reconstructions again assuming the ellipticity of the eddy core at each geopotential level, this time using the right side of the core (ellipses $E_2$) to construct volumes. In this case, the shapes are quite similar to those in Figure

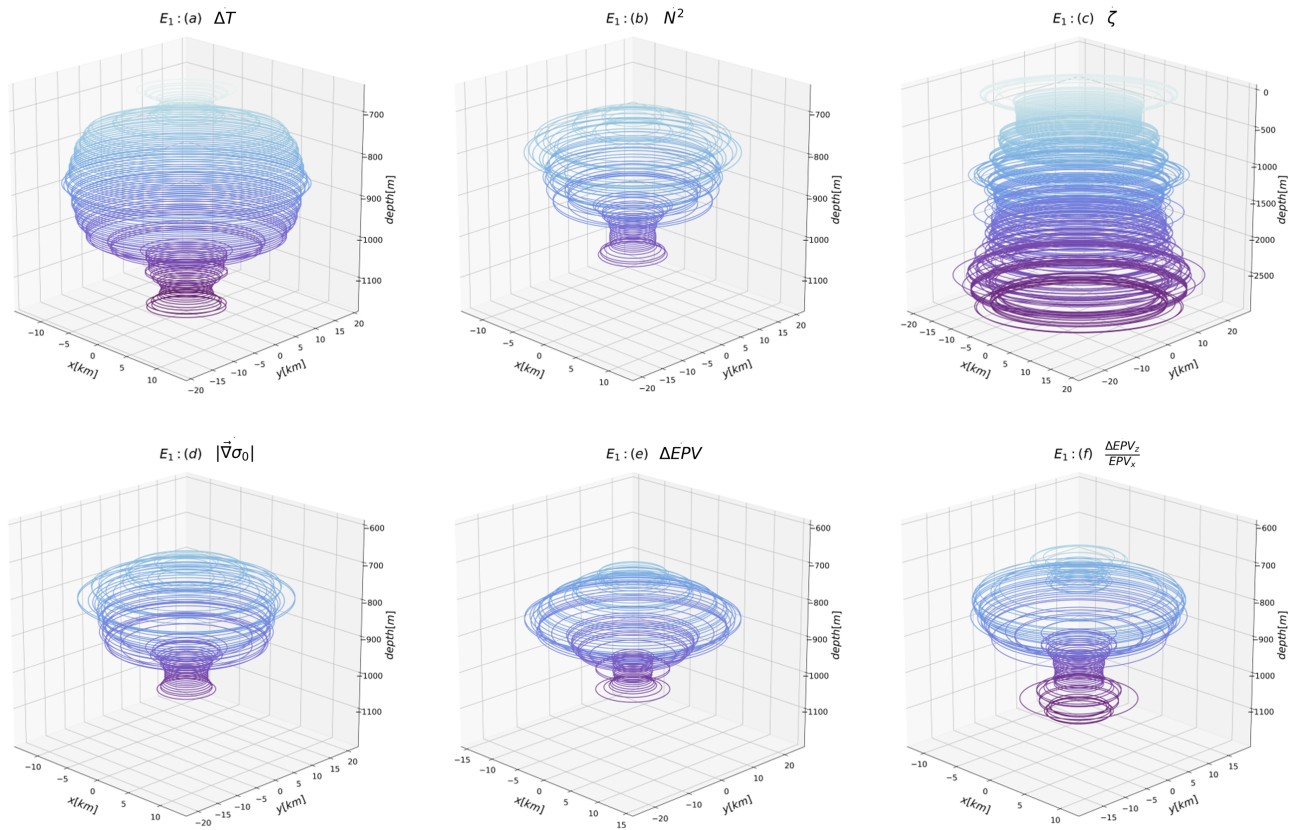

**Figure 12.** 3D reconstructions of AE $N°24$ assuming the ellipticity of the eddy at each geopotential level. Each panel corresponds to a criterion. The criteria are detailed in Figure 10. (a): Thermal anomaly on isopycnals, (b): Brunt-Väisälä frequency, (c): Relative vorticity, (d): 2D density gradient norm, (e): Ertel potential vorticity anomaly, (f): $\Delta EPV_z/EPV_x$. Contours are plotted every five meters.

11, but the eddy volumes are larger. The thermal anomaly criterion results in a very convex shape. The Brunt-Väisälä frequency criterion and the 2D density gradient norm give shapes similar to those of the circular eddy. Except for the relative vorticity criterion, the bottom of each eddy is thinner than the top, similar to Figure 11. We also recover the conical eddy using the criterion on $\Delta EPV_z/EPV_x$.

### 5.3.3    Eddy Volume Comparison

The volumes and uncertainties for the three eddies considered are now calculated and summarized in Figure 14. For each eddy, the volume has been normalized to the cylindrical volume $\Omega_0 = \pi L^2 H$, where $L$ and $H$ are given in Table 3 (note that $L$ is defined in Figure 1). The normalized volumes for circular vortices are obviously closer to 1 than for ellipses.

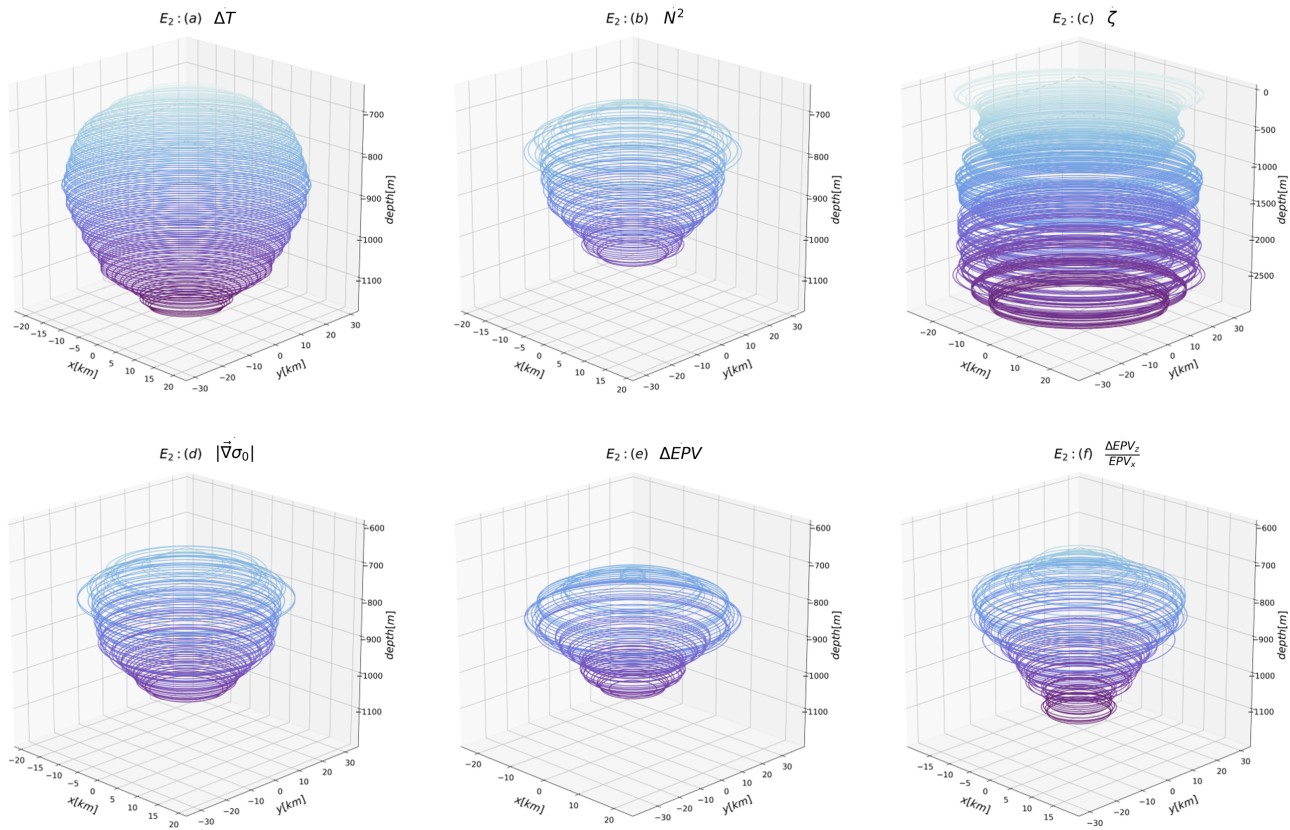

**Figure 13.** 3D reconstructions of AE $N°24$ assuming its ellipticity at each geopotential level. Each panel corresponds to one criterion. The criteria are detailed in Figure 10. (a): Thermal anomaly on isopycnals, (b): Brunt-Väisälä frequency, (c): Relative vorticity, (d): 2D density gradient norm, (e): Ertel potential vorticity anomaly, (f): $\Delta EPV_z/EPV_x$. Contours are plotted every five meters.

For any approximation method (circular or elliptical), the volume depends on the chosen criterion. For example, assuming the circularity of the eddy $N°24$, the volume is twice as small with the $\Delta EPV_z/EPV_x$ criterion as with the thermal anomaly criterion. Conversely, for a given criterion, the ellipses-based method yields larger volumes than the circular approximation. As expected, the relative vorticity criterion overestimates the entrapped volume. The criteria based on the Brunt-Väisälä frequency, the norm of the 2D density gradient, the $EPV$ anomaly, and the $\Delta EPV_z/EPV_x$ give closer values regardless of the method used.

In all cases, the approximation of the volume by a cylinder of constant radius ($\Omega_0$ in Figure 14) with *in situ* data leads to an overestimation of the trapped volume compared to the reconstruction using circles ("circ" in Figure 14). Conversely, for elliptical shapes, the tracer transport seems to be overestimated compared to the constant radius approximation.

Using the $\Delta EPV_z/EPV_x$ criterion as a reference, relative differences with other criteria have been calculated and are shown in Figure 15. As mentioned above, thermohaline anomalies on isopycnals lead to a larger volume estimate than with

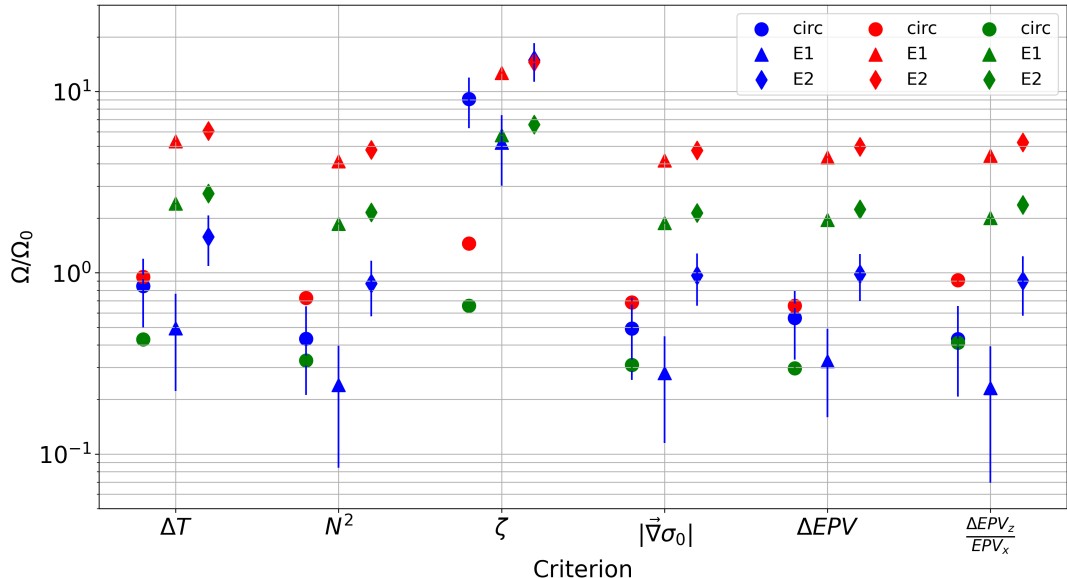

**Figure 14.** Normalized volume as a function of criterion used, for eddies $N°1$ (green markers), $N°7$ (red markers), $N°24$ (blue markers) using the two reconstruction methods. Normalized volumes are plotted by criterion and by method. Error bars have been added, but are only visible for AE $N°24$ because the horizontal resolution of AE $N°1$ and $N°7$ is finer than 3 % of the apparent eddy radius $L$. Since the volumes obtained with the relative vorticity criterion are much larger than those obtained with the other criteria, a logarithmic scale has been used.

the $\Delta EPV_z/EPV_x$ criterion (see Figure 10) and the relative difference between the volumes is large. For example, AE $N°24$
has twice the volume with thermohaline anomalies than with the $\Delta EPV_z/EPV_x$ criterion. The relative error between $EPV$ anomaly and $\Delta EPV_z/EPV_x$ is also noticeable, reaching more than 30 % for eddy $N°1$. Since the $EPV$ anomaly is calculated using the horizontal contribution $EPV_x$, and since this term increases near the boundary, the total volume increases even as $EPV_z$ decreases. Physically, the region where $EPV_x$ is large is more likely to experience frontal instabilities. Therefore, the water properties in this region can change due to mixing and the core can decay. As a consequence, the TC core is somewhat
overestimated by $\Delta EPV$.

Finally, the most remarkable result is that the volume obtained with the $N^2$ criterion is a good approximation of that obtained with $\Delta EPV_z/EPV_x$. In fact, the relative error between the two computed volumes does not exceed 20 %, regardless of the eddy and the method used. The criterion-based norm of the 2D density gradient also gives similar results to the latter two, which is consistent with their mathematical definitions. In fact, eddies modify the local stratification due to their trapped water;
thus creating a baroclinic contribution to the buoyancy field. Consequently, the calculation of $N^2$ reflects the eddy core. To illustrate this last point, Meunier et al. (2021) performed a decomposition of $EPV$ into three terms for an eddy sampled by gliders in the Gulf of Mexico; they showed that the eddy stretching (related to the vertical buoyancy gradient) was the dominant term. Our conclusions from Figure 15 are consistent with this result and our theoretical development.

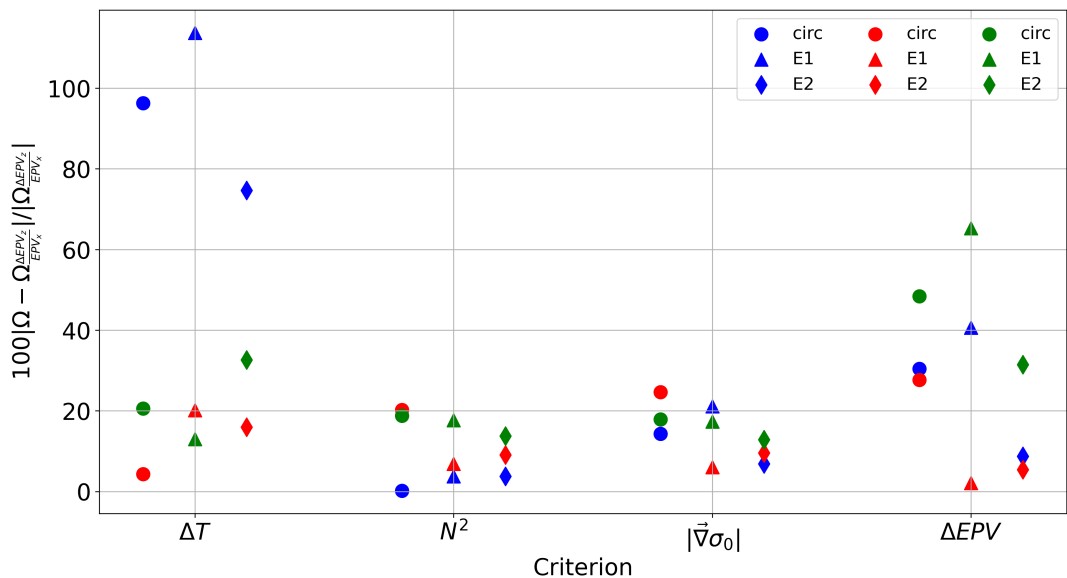

**Figure 15.** Relative gap between volume approximations using that of $\Delta EPV_z/EPV_x$ as a reference. As in Figure 14, results are plotted for eddies $N°1$ (green markers), $N°7$ (red markers), $N°24$ (blue markers)

## 6 Conclusions

This paper presents an evaluation of the thermohaline coherence of mesoscale eddies based on *in situ* data collected during several cruises, primarily in the Atlantic Ocean. Our findings indicate that TC eddies are not uncommon. Indeed, our analysis of the *in situ* data set has yielded a high rate of TC cores. A notable aspect of this study is that TC eddy cores are often situated beneath the surface or even the pycnocline, making them unidentifiable as such through satellite altimetry data alone. In such fields, the presence of subsurface eddies is either undetectable or, if discernible, the derived surface geostrophic velocity is not an appropriate velocity field for inferring the material coherence of the eddy. It is recommended that future studies exercise caution when using the terms "surface" or "subsurface" to describe an eddy, as the applicability of these adjectives is contingent upon the criteria employed.

For TC eddies, we present two methods to extrapolate eddy volume using a single ship section. The first method is based on the assumption of circularity at each geopotential level, which results in estimated volumes that are lower than those calculated using the second method, which assumes ellipticity of the eddy core. Moreover, volumes were calculated and compared using different criteria to define the boundaries of the eddies. Following theoretical considerations and data validation, it can be concluded that the outermost closed contour of the Brunt-Väisälä frequency anomaly at each depth provides an accurate approximation of the TC eddy core. This result corroborates the findings of previous studies (Meunier et al., 2021; Paillet, 1999; Paillet et al., 2002), further strengthening the body of research on eddy dynamics through the use of Argo profiling float

data. It is recommended that future studies exercise caution when attempting to describe the shape of eddies, as the outcome is contingent upon the criterion employed. It is important to note that eddies are not perfectly cylindrical or conical in shape.

Further studies are required to address thermohaline anomalies and Lagrangian criteria, enabling a comprehensive assessment of material coherence through temporal monitoring. The quantity of available *in situ* data is approaching its limits for this purpose, so additional data collection is necessary.

*Data availability.* In this study, we benefited from numerous data sets freely available and listed here.

The ADT produced by Ssalto/Duacs distributed by CMEMS, accessed on 19 January 2021:

https://resources.marine.copernicus.eu

The climatological standard deviation for temperature and salinity in the time period 1991-2020 are freely available on the WOA website (Reagan et al., 2024; Locarnini et al., 2024) :

https://www.ncei.noaa.gov/access/world-ocean-atlas-2023/

The concatenated RVs Atalante and Maria S Merian hydrographic and velocity data (L'Hegaret Pierre, 2020) are freely available on the SEANOE website :

https://www.seanoe.org/data/00809/92071/, accessed on 15 March 2021.

The hydrographic and velocity measurements taken during the M124 cruise (Karstensen and Wölfl, 2016; Karstensen et al., 2016;

Karstensen and Krahmann, 2016) of the RV Meteor are freely available on the PANGAEA web site:

https://doi.org/10.1594/PANGAEA.902947, https://doi.pangaea.de/10.1594/PANGAEA.863015, https://doi.pangaea.de/10.1594/PANGAEA.869740.

The hydrographic and velocity data collected during the M160 cruise (Dengler et al., 2022a, b, c) of the RV Meteor are freely available on the PANGAEA web site:

https://doi.org/10.1594/PANGAEA.943409, https://doi.org/10.1594/PANGAEA.943432, https://doi.org/10.1594/PANGAEA.943657.

The hydrographic and velocity data collected during the MSM60 cruise (Karstensen, 2020b, a; Karstensen et al., 2020) of the RV Meteor are freely available on the PANGAEA web site:

https://doi.pangaea.de/10.1594/PANGAEA.915879, https://doi.pangaea.de/10.1594/PANGAEA.915898 https://doi.pangaea.de/10.1594/PANGAEA.91

The hydrographic and velocity data collected during the MSM74 cruise (Karstensen and Krahmann, 2021; Karstensen and Czeschel, 2021) of the RV Meteor are freely available on the PANGAEA web site:

https://doi.pangaea.de/10.1594/PANGAEA.929000, https://doi.pangaea.de/10.1594/PANGAEA.928976

The hydrographic and velocity measurements along Physindien 2011 (L'Hégaret et al., 2016) are freely available on Ifremer website:

https://co-en.ifremer.fr/eulerianPlatform?contextId=8890&ptfCode=1901185&lang=en.

Finally, hydrographic and velocity data collected during the RV Kristine Bonnevie and RV Hakon Mosby KB2017606, HM2016611, KB2017618 cruise (Fer et al., 2019; Bosse et al., 2019) are freely available on the NMDC website:

https://doi.org/10.21335/NMDC-1093031037.

## Appendix A: Uncertainties

In order to compute uncertainties on *in situ* variables and quantities, we use the formula of Carton et al. (2002). For example, given the horizontal gradient of the temperature $\partial_x T$, since we use the finite difference method, the gradient and the error $\delta(\partial_x T)$ is written as follows

$$\delta(\partial_x T) = 2 \frac{\delta_H T}{\delta_H x}, \tag{A1}$$

where $\delta_H T$ and $\delta_H x$ refer to the uncertainty in temperature and horizontal resolution, respectively. Here $\delta_H$ refers to hydrological data: the horizontal resolution is that of the hydrological instruments. Similarly, $\delta_V$ refers to the uncertainty associated with the velocity data. For buoyancy, the linearized equation of state was used to determine the uncertainty:

$$\delta_H b = -\frac{g}{\sigma_0} \delta\sigma = -\frac{g}{\sigma_0}(-\alpha \delta_H T + \beta \delta_H S), \tag{A2}$$

where $g$ is the acceleration due to gravity, $\sigma_0$ is a reference value taken here as an average over each profile of a considered section, $\alpha = 2 \times 10^{-4} \text{ kg.m}^{-3}.\text{K}^{-1}$ and $\beta = 7.4 \times 10^{-4} \text{ kg}^2.\text{m}^{-3}.\text{g}^{-1}$ are classical averages to simplify the calculation. The lists of relative errors for the calculated quantities are given in Table A1.

## Appendix B: 3D reconstruction of eddies using elliptically based tubes

Using altimetry data and detection algorithms, Chen et al. (2019) showed that ellipses are the most common shape for ocean surface eddies. Perfectly elliptical eddies are rare, but ellipses remain the best fit to characterize the shape of almost the entirety of surface eddies. Indeed, isolated eddies tend to be circular, but in the global ocean, eddies are often deformed by the background flow or its beta drift, and thus undergo elongation. They calculated the best-fit ellipses for eddies over a 20-year period (1996-2016) and analyzed the eccentricity of the eddies that left an imprint on the ocean surface. They also studied the average orientation of the semi-major axis of these elliptical eddies with respect to the parallels in each ocean basin. As a result, they obtained the distribution of the mean eccentricity as a function of latitude, as well as the distribution of the mean semi-major axis orientation (see Figure 6 and 8 from Chen et al. (2019)). Although they worked on surface eddies, we assume that their results also apply to subsurface eddies. Here we show how to reconstruct an elliptical eddy using the latter two results and a ship track.

The approach is the same as in the previous part. At each geopotential level within the eddy core, an ellipse is constructed to find an elementary volume of height $dz$. By summing at each geopotential level, the total volume is obtained. Figure A1 illustrates the main geometric points and constructions used to find the semi-major and semi-major axes of the ellipse. For each geopotential level within the eddy core, the main steps can be described as follows:

1. Using the orthogonal velocity $V_o$, the eddy center $C$ on the ship section is calculated. With a given criterion, the eddy core boundary is determined and $P$ and $Q$, the extremities of the core on the ship section, are defined.

**Table A1.** Lists of uncertainties for the horizontal and vertical gradients of temperature, potential density and the relative vorticity.

| N° | $\delta(\partial_x T)$ [°C.m$^{-1}$] | $\delta(\partial_z T)$ [°C.m$^{-1}$] | $\delta(\zeta)$ [s$^{-1}$] | $\delta(\partial_z V_o)$ [s$^{-1}$] | $\delta(\partial_x \sigma_0)$ [kg.m$^{-4}$] | $\delta(\partial_z \sigma_0)$ [kg.m$^{-4}$] |
|----|------|------|------|------|------|------|
| 1 | 5.71E-06 | 4.00E-02 | 1.71E-05 | 7.50E-03 | 9.60E-06 | 6.72E-05 |
| 2 | 2.38E-06 | 4.00E-02 | 7.14E-06 | 7.50E-03 | 4.00E-06 | 6.72E-05 |
| 3 | 1.54E-06 | 4.00E-02 | 4.62E-06 | 7.50E-03 | 2.58E-06 | 6.72E-05 |
| 4 | 7.60E-07 | 2.00E-02 | 2.28E-06 | 6.00E-03 | 1.28E-06 | 3.36E-05 |
| 5 | 4.80E-07 | 2.00E-02 | 1.44E-06 | 6.00E-03 | 8.06E-07 | 3.36E-05 |
| 6 | 4.65E-07 | 2.00E-02 | 1.40E-06 | 6.00E-03 | 7.81E-07 | 3.36E-05 |
| 7 | 1.11E-05 | 2.00E-01 | 3.33E-05 | 7.50E-03 | 1.87E-05 | 3.36E-04 |
| 8 | 1.18E-05 | 2.00E-01 | 3.53E-05 | 7.50E-03 | 1.98E-05 | 3.36E-04 |
| 9 | 9.62E-07 | 2.00E-02 | 2.88E-06 | 1.87E-03 | 1.62E-06 | 3.36E-05 |
| 10 | 8.70E-07 | 2.00E-02 | 2.61E-06 | 1.87E-03 | 1.46E-06 | 3.36E-05 |
| 11 | 8.70E-07 | 2.00E-02 | 2.61E-06 | 1.87E-03 | 1.46E-06 | 3.36E-05 |
| 12 | 8.70E-07 | 2.00E-02 | 2.61E-06 | 1.87E-03 | 1.46E-06 | 3.36E-05 |
| 13 | 1.67E-06 | 2.00E-02 | 5.00E-06 | 1.87E-03 | 2.80E-06 | 3.36E-05 |
| 14 | 9.52E-07 | 2.00E-02 | 2.86E-06 | 1.87E-03 | 1.60E-06 | 3.36E-05 |
| 15 | 9.52E-07 | 2.00E-02 | 2.86E-06 | 1.87E-03 | 1.60E-06 | 3.36E-05 |
| 16 | 1.00E-06 | 2.00E-02 | 3.00E-06 | 1.87E-03 | 1.68E-06 | 3.36E-05 |
| 17 | 8.00E-07 | 2.00E-02 | 2.40E-06 | 7.50E-03 | 1.34E-06 | 3.36E-05 |
| 18 | 5.97E-07 | 2.00E-02 | 1.79E-06 | 7.50E-03 | 1.00E-06 | 3.36E-05 |
| 19 | 9.85E-07 | 2.00E-02 | 2.96E-06 | 7.50E-03 | 1.66E-06 | 3.36E-05 |
| 20 | 9.85E-07 | 2.00E-02 | 2.96E-06 | 7.50E-03 | 1.66E-06 | 3.36E-05 |
| 21 | 1.32E-06 | 2.00E-02 | 3.97E-06 | 7.50E-03 | 2.23E-06 | 3.36E-05 |
| 22 | 3.03E-06 | 2.00E-02 | 9.09E-06 | 7.50E-03 | 5.09E-06 | 3.36E-05 |
| 23 | 2.33E-06 | 2.00E-02 | 6.98E-06 | 7.50E-03 | 3.91E-06 | 3.36E-05 |
| 24 | 4.00E-06 | 2.00E-02 | 1.20E-05 | 7.50E-03 | 6.72E-06 | 3.36E-05 |
| 25 | 3.45E-06 | 2.00E-02 | 1.03E-05 | 7.50E-03 | 5.79E-06 | 3.36E-05 |

2. Using the Nencioli et al. (2008) routine for the considered geopotential level, the location of the real eddy center $N$ can be approximated. $N$ is then the center of the ellipse. $N$ is also taken as the center of the local $f-$plane Cartesian frame $(N, \boldsymbol{x}, \boldsymbol{y})$, where $\boldsymbol{x}$ is the zonal vector and $\boldsymbol{y}$ is the meridional vector. Starting from $N$, 1° north and 1° east are converted into horizontal and vertical length scales.

3. On this $f-$ plane, the line $(NC)$ can be drawn, and depending on its orientation with respect to the parallels, we set it as the semi-major axis or the semi-major axis, following the results of Chen et al. (2019). Since they obtained a global distribution of semi-major axis orientations for best-fit vortex ellipses, we can determine which $(NC)$ is more likely. Then $P'$ and $Q'$, two points on the ship's orbit, are computed such that $Q'C = CP$ and $QC = CP'$.

4. In a 2D Cartesian frame, 5 points are needed to compute the exact equation of an ellipse. Here, our ellipse is initially constrained by its center $N$, the orientation of the semimajor (or semiminor) axis $(NC)$, and the eccentricity imposed by the work of Chen et al. (2019). However, adding the two points $P'$ and $Q'$ will over-constrain the problem (considering its equations). Therefore, a choice must be made between $P'$ and $Q'$ to add a unique final constraint. As a consequence, two ellipses can be obtained: one passing through the point $P'$, arbitrarily called $(E_1)$, and one passing through the point $Q'$, arbitrarily called $(E_2)$. In the following steps, $P'$ will be used arbitrarily to explain the procedure.

5. In polar coordinates, if $(NC)$ is the orientation of the semi-major axis, the semi-major axis $b$ can be obtained by

$$b = |NP|\sqrt{1 - \varepsilon^2 \cos^2 \theta_1}, \tag{B1}$$

where $|NP| > 0$ is the Cartesian distance between $N$ and $P$, $\varepsilon$ is the imposed eccentricity, and $\theta_1 > 0$. If $(NC)$ is the orientation of the semi-minor axis, we replace $\theta_1$ with $\frac{\pi}{2} + \theta_1$. Then we can calculate the semi-major axis $a$

$$a = \frac{b}{\sqrt{1 - \varepsilon^2}}, \tag{B2}$$

6. Finally, the ellipse equation reads

$$\left(\frac{x \cos \alpha + y \sin \alpha}{a}\right)^2 + \left(\frac{-x \sin \alpha + y \cos \alpha}{b}\right)^2 = 1, \tag{B3}$$

where $\alpha$ is defined in the Figure A1, $x$ and $y$ are the two variables associated with the zonal and meridional axes respectively. The approximate volume is: $\Omega = \int_{-H}^{0} \pi a(z)b(z)dz$ for a surface vortex. For a subsurface vortex the boundary conditions have to be changed as in the previous part.

This method defines the uncertainty due to resolution as

$$\frac{\delta \Omega}{\Omega} = \frac{\int_{-H-\Delta z}^{0} \pi (a(z) + \Delta x)(b(z) + \Delta x)dz}{\int_{-H}^{0} \pi a(z)b(z)dz} - 1. \tag{B4}$$

This method preserves the non-axisymmetry of the eddy and takes into account the vertical structure. The center is that of the Nencioli et al. (2008) routine, which remains an approximation but gives a better estimate than the previous method. The elliptical shape is more common than the circular shape among vortices. Note, however, that this method requires that $N$ and $C$ are on the same semi-major (or minor) axis and that the eccentricity is known. Two ellipses can be determined by this method (there is no uniqueness). Furthermore, the real upper and lower limits of the core remain unknown, and our method extrapolates in this region. Indeed, in the ship section, the upper and lower limits are characterized by the fact that $P$ and $Q$ tend to $C$, so that $PQ$ tends to vanish. However, looking at equation (13), the semi-major axis will not remain zero when approaching these boundaries. To avoid this side effect, ellipses are found only at the geopotential level where $PQ \neq 0$. Therefore, the volume will be underestimated.

*Author contributions.* Yan Barabinot performed the main diagnostics and wrote the original draft. Complements and revisions have been carried out by Sabrina Speich and Xavier Carton

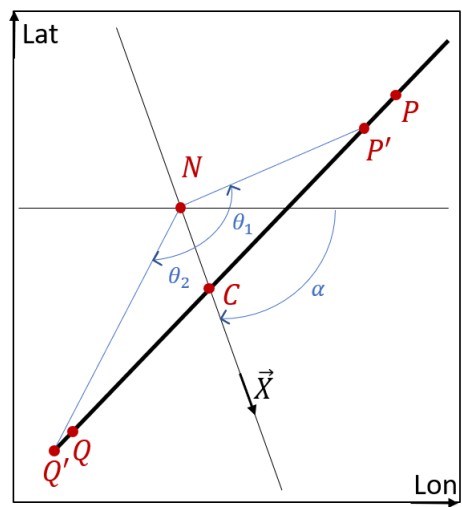

**Figure A1.** Main geometric constructions for solving ellipse equations.

*Competing interests.* The authors report no conflict of interest.

*Acknowledgements.* This research was supported by the European Union's Horizon2020 research and innovation program under grant agreement no. 817578 (TRIATLAS), the Centre National d'Etudes Spatiales through the TOEddies and EUREC4A-OA projects, the French National Program LEFE INSU, IFREMER, the French Vessel Research Fleet, the DATA TERRA French Research Infrastructures AERIS and ODATIS, IPSL, the Chaire Chanel Program of the ENS Geosciences Department, and the EUREC4A-OA JPI Ocean and Climate Program. We thank all the people who collected, processed and made public the data as well as all the institutions for which these people worked in particular the University of Bergen and GEOMAR Helmoltz Centre for Ocean Research Kiel. We also warmly thank every captain and crew of the RVs Atalante, Maria S. Merian, FS Meteor, RV Kristine Bonnevie and RV Hakon Mosby without whom this study could not have been carried out. Yan Barabinot is supported by a Ph.D. grant from the Ecole Normale Supérieure de Saclay. Xavier Carton acknowledges support from UBO and a CNES contract EUREC4A-OA.

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
