# Peer review of "Assessing the Thermohaline Coherence of Mesoscale Eddies as described from In Situ Data"

_EGUsphere, 2024_

## Referee Comment (RC2)

**Review of:** *Assessing the Material Coherence of Mesoscale Eddies as described from In Situ Data*, by Barabinot et al.

This manuscript uses in situ data from eight oceanographic cruises that cover seven different regions to assess the coherence of mesoscale ocean eddies. An impressive amount of work has been done, particularly insightful is the analysis on subsurface signatures of ocean eddies, and how these water masses of distinct thermohaline properties may propagate through the ocean without a surface signature. Below, I provide a list of major and particular concerns I have with the manuscript, as well as a list of minor typos/grammar. I don't think the manuscript, in its current form, is ready for acceptance.

**Major comment - Material coherence, and materially coherent eddies:**
The authors provide a broad overview of the literature for 'coherent eddies', with a nice introduction on the history of the field of understanding/quantifying eddy coherence. However, when it comes to material coherence, and materially coherent eddies, they've missed quite a number of relevant references from the Lagrangian coherent structures community.

Specifically, the discussion around how 'kinematic coherence' (KC) is reference frame dependent (and hence, not objective), is missing another key point, that eddies defined by Eulerian (frame-dependent) methods are not materially coherent. Quite a number of studies have shown that ocean eddies, identified from the Eulerian perspective (for e.g., Okubo-Weiss parameter, SSH anomaly), leak material across their identified boundaries relatively quickly (Andrade-Canto et al. 2020, Beron-Vera et al. 2013, Haller et al. 2013, Liu et al. 2019, Serra et al. 2017, Denes et al. 2022). This is a major drawback of using Eulerian approaches for quantifying material coherence, and mass transport by eddies.

I also found the definition of material coherence in the manuscript to be rather qualitative, that a 'vortex ceases to be coherent when it loses its trapped water mass'. How does one determine the point at which a vortex loses its trapped water mass? Recent Lagrangian analyses have considered the coherence of ocean eddies, specifically Abernathey et al. (2018), and Wang et al. (2015), found that only small 'coherent inner cores' of ocean eddies existed over long time-periods, while Denes et al. (2022) found ocean eddies may consists of coherent inner cores, and quasi/semi-coherent outer rings (and question the notion that ocean eddies have precise boundaries). This is briefly touched on by the authors on lines 70-71, but I think more is needed here.

A particular concept raised in 'material coherence' theory is that the coherence occurs over a finite window of time. As described in the manuscript, ocean eddies tend to have lifetimes of months up to years, however, with entrainment and flushing of water, an ocean eddy is not materially coherent over its entire lifetime. To assess the material coherence of an ocean eddy from a single snapshot in time seems rather suspect to me. As the authors note, it is 'impossible to perform temporal studies with in situ data', however material coherence has a time-evolution component that I think is quite important when assessing mesoscale ocean eddies.

I suggest the authors consider an alternate motivation for the paper. I can see the techniques described and analysis developed as nice ways to 'Reconstruct 3D ocean eddies from 2D in situ transects'.

**Particular comments:**

1. Methodology

I particularly liked the descriptive methodology of the paper, the authors explain and derive their methodology nicely. However, at times I was a little lost. I think the manuscript would benefit from an additional paragraph or two which may provide an overview of the methodology. For example, section 3 describes four identification methods, but no introductory sentence/paragraph which describes why these identification methods are used let alone useful. What alternative methods exist, and why are these four chosen?

2. 'Acronymisation'

I felt there were a few too many acronyms, but in cases some things could have been 'acronymised'. For example, at one point the authors use 'AC' and 'C', to describe (from what I understand) anticyclonic and cyclonic eddies (line 268, or 403, but not described anywhere in text). Especially when, in Table 3, 'C' is used to describe 'Complete' eddies. In contrast, to be in line with 'Kinematic coherence' and 'Material coherence' (KC and MC), I think the authors can use 'TC' for 'Thermohaline coherence', which would help with the flow of the discussion in section 5.2, where they describe 'thermohalinely coherent cores', for example.

3. Line 32, the term 'eddy boundary' needs to be placed in context.

Eddy boundaries are method dependent, and some literature argues that eddies do not have a sharp boundary. I think this sentence could be qualified further.

4. Line 64-66 on MC theory

Some material coherence theory is derived from 'minimal mixing' criteria, specifically, that a boundary of a materially coherent structure is undistorted/unfilamented over a finite-time window, such that diffusive mixing across the boundary is minimised. This sentence can be qualified further.

5. Line 67-68 on number of MC eddies vs. KC eddies.

I wasn't sure what the purpose of this sentence is. As the authors suggest, material coherence provides a rather rigorous definition for ocean eddies, and one should expect that more structures will appear from a less restrictive approach than a more restrictive approach. Is this considered a downside of MC approaches, or just a statement of fact?

6. Describing the trapped water as 'heterogeneous' (e.g. line 85, 306)

I found this a little confusing, as the trapped water mass is typically homogeneous within the eddy core, but distinct from the surrounding water mass, as it is 'characteristic of the region of formation of that eddy'. I would think of this water mass as 'distinct'.

7. Line 91 – providing a first answer to the question of the 3D material coherence of eddies

I'm not entirely sure what the question actually is. That 3D material coherence of eddies can be assessed using water masses and the PV approach? Certainly, 3D material coherence of eddies has been considered in the past, see Froyland et al. (2012).

8. Definition of the center of an eddy

The authors make contrasting statements '... is reached at the eddy center (region where the velocity tends to zero)' on line 100, and 'The center of the eddy is defined as the point where the mean tangential velocity is maximum', and '... cross the exact eddy center, defined as the location of the zero velocity' on lines 344-345. To me, these three statements are inconsistent.

Additionally, in the first instance the eddy center is a region, in the second it is either a point or a curve, and in the third it a point (possibly a region?).

9. Line 244 'These values are small compared to the first-order terms'

I would disagree, in the horizontal case they are of the same order of magnitude. They are smaller, but not negligibly small. Is this not important?

10. Lines 300-301 and lines 441-442 – Motivation

The motivation for the analysis described in both these sections would be great to see in the introduction. In both cases, I did not know this would form part of the analysis until I reached these sentences.

11. Line 361 – 'To reduce this uncertainty, …'

By computing volumes for a subset of eddies, you aren't exactly reducing uncertainty. Rather, you are only showing results for eddies where the uncertainty isn't too large. I would rephrase this.

12. Layout of Figures 4 and 5

Use of figure labelling would be really useful for both of these figures, using a), b) etc. Also, the change in orientation for the bottom panels of Figure 4 I found rather jarring, is there a reason to have these laid out vertically rather than horizontally like the top panel of Figure 4 and all of Figure 5? Lastly, I suggest a second diverging colorbar for the salinity anomaly (different to the thermal anomaly colorbar).

13. Lines 458-459 – 'Lagrangian studies suggest that the ability of eddies to trap a water mass is a consequence of closed trajectories'.

I'm not certain what you mean by this. What is a closed trajectory in this instance? A citation would be very useful.

14. Figure 7
    a. Plotted velocity vectors - I find these velocity vectors distract from the plots, especially in panel g), where I feel the figure is too busy to see what is really going on.
    b. Panel title (a) – 'fev', and panel title (i) – missing date (same on Figure 8)

15. Figure 9 colorbar and panel titles

I would change the colorbar to something that contrasts better with the isopycnal, EPV, and salinity curves (being black and blue), otherwise it is very hard to appreciate these curves. I also found the panel titles to be rather confusing and could do with an update.

16. Line 519-520 – 'The reconstructed volume associated with the thermal anomaly is the most convex of all shapes'.

I'm not sure what you mean by this, nor am I sure if I agree with the statement. In fact, some of the shapes plotted in Figure 10 aren't convex (panel (c) for example).

17. Figure 10 – panel titles

If you could add titles to each of the panels (not just in the figure caption), that would really help guide the reader. Also, given the circularity constraint, I think each of these plots should have equal x-y aspects, figure (c) looks out of shape due to the non-equal aspect ratio.

18. Lines 525 – 533 regarding Figures 11 and 12

The authors say 'This figure refers to the ellipses (E1) mentioned earlier' and later '(ellipses E2)', but the only mention I could find to E1 or E2 were in the caption of figure 9. Where do these numbers come from, and how were they computed? Secondly, what is the purpose of an elliptical reconstruction? That was not clear to me.

19. Line 569-570 – 'This paper presents an evaluation of the thermohaline coherence of mesoscale eddies based on in situ data collected during several cruises...'

While I agree with this statement, the title of the manuscript is on assessing 'material coherence'. While I don't think the manuscript has addressed the material coherence of ocean eddies, this statement in the conclusion is at odds with the title and abstract of the manuscript.

**Typos/grammar/missing info:**

Line 58 – there is an erroneous space between 'velocity fields' and the comma.

Line 128 "boundary current systems off Labrador" - Do you mean Labrador sea?

Line 196 – 'Localization' should be 'Localisation'.

Lines 105, 192, 438 – 'confront' could be 'compared'.

Line 195-197 – Could you provide a citation to the data described?

Line 233, Fig 4 caption, Fig 5 caption – you use 'gyre', do you mean eddy?

Line 280 – 'the EPV id the ocean at *rest*', what do you mean by 'rest'?

Equation 7 – $f_0$ is defined, but $f$ is not defined.

Line 321 – what are \overbar{b} and b' ?

Line 327 – 'adimensionalizing' should be 'nondimensionalising'.

Line 364 – 'analytically', I wasn't sure what you meant by this.

Line 366 – 'descrided' should be 'described'.

Line 564 – 'this' should be 'thus'.

Line 582 – 'This result corroborates the findings of previous studies...' citations are needed for this statement.

**References:**

F. Andrade-Canto, D. Karrasch, and F. J. Beron-Vera, Genesis, evolution, and apocalypse of loop current rings, Phys. Fluids 32, 116603 (2020).

F. J. Beron-Vera, Y. Wang, M. J. Olascoaga, G. J. Goni, and G. Haller, Objective detection of oceanic eddies and the Agulhas leakage, J. Phys. Oceanogr. 43, 1426 (2013).

G. Haller and F. J. Beron-Vera, Coherent Lagrangian vortices: The black holes of turbulence, J. Fluid Mech. 731, R4 (2013).

T. Liu, R. Abernathey, A. Sinha, and D. Chen, Quantifying Eulerian Eddy Leakiness in an Idealized Model, J. Geophys. Res. [Oceans] 124, 8869 (2019).

M. Serra and G. Haller, Forecasting long-lived Lagrangian vortices from their objective Eulerian footprints, J. Fluid Mechanics 813, 436 (2017).

M.C. Denes, G. Froyland, S.R. Keating, Persistence and material coherence of a mesoscale ocean eddy, Physical Review Fluids 7, 034501 (2022).

R. Abernathey and G. Haller, Transport by Lagrangian vortices in the Eastern Pacific, J. Phys. Oceanogr. 48, 667 (2018).

Y. Wang, M. J. Olascoaga, and F. J. Beron-Vera, Coherent water transport across the South Atlantic, Geophys. Res. Lett. 42, 4072 (2015).

G. Froyland, C. Horenkamp, V. Rossi, N. Santitissadeekorn, and A. Sen Gupta. Three-dimensional characterization and tracking of an Agulhas Ring. *Ocean Modelling*, V52-53:69-75 (2012)

---

## Author Response (AR1)

Referee 1

We would like to thank the referee for his/her comments. Please find our point-to-point response below and modifications associated with the revised version in purple.

The authors of this manuscript should conduct a more comprehensive review of the existing literature on material frame-indifferent (i.e., objective) coherent vortex detection. The manuscript contains several inaccurate or unsubstantiated statements, including the following:

The intention is not to offer a critique of the material frame-indifferent coherent (MC) vortex theory as we concur with its mathematical and physical formulation. The objective of our study is to examine the impact of material coherence on the thermohaline structure of eddies. It is accurate to note that our analysis does not explicitly demonstrate a temporal correspondence between the MC theory and the presence of anomalies in the core of eddies. However, the presence of anomalies is indicative of the existence of water that does not exhibit the same thermohaline properties as the surrounding water. As postulated by the MC theory, the material boundaries maintain the thermohaline characteristics of the transported water, from the eddy region of generation. Consequently, the trapped water exhibits disparities from the surrounding water as a consequence of the eddy motion in the ocean. Anomalies on isopycnals represent a primary indicator of heterogeneous water masses within eddy cores, enabling the computation of heat and salt content in these structures through the use of in situ data (Aguedjou et al., 2021; Yang et al., 2015; Dong et al., 2017; Dong et al., 2014; Chen et al., 2012).

We have changed the introduction from lines 93 to 104 to clarify our purpose.

The authors of this manuscript should conduct a more comprehensive review of the extant literature on material frame-indifferent (i.e., objective) coherent vortex detection. The manuscript contains several inaccurate or unsubstantiated statements, including the following:

1) 'Flierl (1981) showed that when the tangential velocity of the vortex is higher than its translational velocity, fluid particles are trapped in the vortex core.' This claim is more an expression of belief rather than a rigorous conclusion, as velocity is dependent on the observer.

We are grateful to the reviewer for bringing this point to our attention and for prompting us to address it. Nevertheless, we consider the assertion to be somewhat misleading in that it fails to acknowledge the role of mathematical Lagrangian arguments in the article by Flierl (1981). To clarify, the robust mathematical Lagrangian approach presented in the article by Flierl (1981) is fundamental to our framework. In particular, equations (3.6) and (3.7) are of significant relevance.

2) 'In particular, MC theory ignores the fact that water masses at the edge of eddies can change their properties due to various types of instabilities.' This statement is incorrect because, if a vortex is characterized as materially coherent, no fluid can traverse its

boundary. More precisely, no material surface can be intersected by fluid flow since it is flow-invariant, regardless of its coherence.

From a theoretical standpoint, we concur with this assertion when considering the macroscopic and cinematic perspective. However, it should be noted that the actual boundaries of ocean eddies are not perfectly impermeable, and thus cannot be represented by line-sized walls. Rather, they are turbulent zones of a certain width where small-scale instabilities occur, such as centrifugal-symmetric instabilities, and where layering can appear, involving salt fingers and vertical recirculation. This is corroborated by the findings of Barabinot et al. (2024), Molodtsov et al. (2020), Bebieva et al. (2016), Hua et al. (2013) among others. A comparison between the boundary defined by the MC theory and the region where this small-scale turbulence occurs would undoubtedly prove insightful and contribute to the existing body of knowledge in this field.

To this end, we have included a paragraph from lines 74 to 82  to our manuscript to clarify this point.

3) 'Furthermore, MC theory is based only on fluid flow and does not consider the potential permeability of the eddy boundary due to diffusion processes or lateral intrusion (Joyce, 1977, 1984; Ruddick et al., 2010).' This is inaccurate since the boundaries of geodesically-detected coherent material vortices serve as minimizers of diffusion (refer to the Annual Review of Fluid Mechanics paper by Haller).

We appreciate the reviewer's insightful comment. We agree that the boundaries of geodesically-detected coherent material vortices serve as minimizers of diffusion, as highlighted in Haller's work in the Annual Review of Fluid Mechanics. However, it is important to note that "minimizers of diffusion" does not equate to "no diffusion." Additionally, we must consider the impact of small-scale instabilities, which can still play a significant role in the permeability of the eddy boundary. These factors indicate that, while diffusion is minimized, it is not entirely eliminated, and small-scale processes can influence the overall dynamics of the eddy boundary.

In the revised version of our manuscript we have added precisions and references from lines 69 to 72.

4) 'We revisited coherence definitions and checked data accuracy.'  This manuscript does not encompass an examination of coherent material vortex framing. Indeed, the manuscript lacks any explicit articulation or statement concerning this subject matter.

We appreciate the reviewer's comment. To clarify, we have reviewed previous definitions of coherence and explained their relevance to our study in lines 18 to 83 of the manuscript (also included in the revised version). While our primary focus was not on framing coherent material vortices explicitly, we believe that our discussion on the definitions of coherence provides a necessary foundation for understanding the context and significance of our study.

More detail on this is provided in the revised version of the manuscript between lines 95 and 101.

5) 'Comparing the horizontal positions of these core anomalies with eddy surface signatures revealed that surface data alone are insufficient for characterizing the eddy material coherence.' This is plausible, but to assess it, time-dependent flow data must be analyzed using an objective method.

We appreciate the reviewer's comment and agree that analyzing time-dependent flow data using an objective method would provide a comprehensive assessment. However, we believe that the snapshots provided by *in situ* data are sufficient to analyze the position of the anomalies with respect to the sea surface. These snapshots allow us to capture the spatial relationship between core anomalies and the eddy surface signatures effectively, even without continuous time-dependent data. We will improve the wording on this point in the revised version of the manuscript.

In the revised version, we have clarified our arguments from lines 105 to 116. We have also rephrased the entire part 5.2 to clarify our results. .

Beyond these imprecise statements, the TC criterion remains nebulous and can only be regarded as qualitative in nature; temporal flow data are imperative to establish coherence. All criteria (gradients; potential vorticity — an observer-dependent quantity; pythagorean arguments) are applied to instantaneous snapshots of observed mass fields. A temporal history is requisite to ascertain if the 'gradient of a property of some kind' is conserved under advection by the flow. It is evident that transect data are unsuitable for this type of analysis.

In our article, we attempted to define a new concept that can be applied to *in situ* oceanographic data. The TC criterion has a physical sense and enables both to identify trapped water mass inside eddy cores and to compute heat and salt transport by these structures.

We acknowledge the reviewer's concern regarding the qualitative nature of the TC criterion and the necessity for temporal flow data to establish coherence. It is indeed a valid point that temporal assessment is crucial for a comprehensive understanding of whether the "gradient of a property of some kind" is conserved under advection by the flow. We agree that gradients, potential vorticity, and other criteria applied to instantaneous snapshots are limited in capturing the dynamic evolution of water masses.

However, we would like to address the reviewer's suggestion that *in situ* data obtained by hydrological transect are unsuitable for this type of analysis. While temporal history is ideal, it is not always feasible. Observations of ocean subsurface thermohaline and velocity properties are very scarce due to logistical and resource constraints. Our methodology aims to provide a practical approach to identifying heterogeneous water masses within eddy cores using available *in situ* data.

By applying the TC criterion to in situ observations along ship transects, we can still gain valuable insights into the structure and composition of eddies. Even without temporal data, the identification of distinct water masses within an eddy can help in understanding the spatial variability and potential transport processes. This approach can be seen as a first step, providing a foundation for future studies that may incorporate more comprehensive temporal assessments.

In conclusion, while we recognize the limitations of our dataset and the importance of temporal analysis, we believe that our methodology offers a meaningful contribution to the study of eddies using available data. It provides a practical tool for identifying and characterizing water masses within eddy cores, which can be further refined with more detailed temporal observations in future research.

In conclusion, I am unable to endorse the publication of this manuscript. I acknowledge the significant effort expended by the author in analyzing in-situ oceanographic campaign data, which holds intrinsic value. However, I urge the author to consider presenting their analysis within an alternative context, as the current application towards assessing material coherence is not appropriate for the available data.

We thank the reviewer for his/her thoughtful comments and for acknowledging the significant effort expended in analyzing the in-situ oceanographic campaign data, which indeed holds intrinsic value.

While we understand the reviewer's concerns regarding the application of our analysis towards assessing material coherence with the available data, we would like to address this point further. The notion of anomaly, as utilized in our study, is intrinsically linked to the concept of material coherence.

Anomalies in oceanographic data often refer to deviations from a mean state, indicating the presence of distinct water masses or features such as eddies. These anomalies are markers of coherent structures within the ocean that can have significant impacts on heat, salt, and carbon, oxygen and nutrient transport. Even though our data might not allow for a complete temporal assessment, it still provides valuable snapshots that reveal these coherent structures and their properties.

By identifying these anomalies, we can infer the presence and characteristics of coherent water masses. This approach, although limited by the lack of temporal data, is still valid and valuable for understanding the spatial distribution and potential impacts of these structures. It serves as a crucial first step, paving the way for future studies that can incorporate more extensive temporal datasets.

We appreciate the reviewer's suggestion to present our analysis within an alternative context. However, we believe that the current context of assessing material coherence through the detection of anomalies in in-situ observations is appropriate and provides significant insights. It aligns with the practical constraints of field oceanographic research, where temporal data may not always be available, and yet meaningful analysis can still be conducted.

In conclusion, while we acknowledge the limitations of our study, we maintain that our methodology and findings contribute valuable knowledge to the field. We are open to further refining our approach and incorporating additional data in future research to enhance the assessment of material coherence.

In the revised version of the manuscript, we will provide a more detailed conclusion in order to clarify this point.

In the revised version, we advise the referee to see the revised Introduction, Section 5.2 and the Conclusion. We believe that the goal and our development are now clear.

We would like to thank the referee for his/her valuable comments, which helped to improve the manuscript.

**References**

Aguedjou, H. M. A., Chaigneau, A., Dadou, I., Morel, Y., Pegliasco, C., Da‑Allada, C. Y., & Baloïtcha, E. (2021). What can we learn from observed temperature and salinity isopycnal anomalies at eddy generation sites? Application in the tropical Atlantic Ocean. *Journal of Geophysical Research: Oceans*, *126*(11), e2021JC017630.

Chen, G., Gan, J., Xie, Q., Chu, X., Wang, D., & Hou, Y. (2012). Eddy heat and salt transports in the South China Sea and their seasonal modulations. *Journal of Geophysical Research: Oceans*, *117*(C5).

Dong, C., McWilliams, J. C., Liu, Y., & Chen, D. (2014). Global heat and salt transports by eddy movement. *Nature communications*, *5*(1), 3294.

Dong, D., Brandt, P., Chang, P., Schütte, F., Yang, X., Yan, J., & Zeng, J. (2017). Mesoscale eddies in the northwestern Pacific Ocean: Three‑dimensional eddy structures and heat/salt transports. *Journal of Geophysical Research: Oceans*, *122*(12), 9795-9813.

Yang, G., Yu, W., Yuan, Y., Zhao, X., Wang, F., Chen, G., ... & Duan, Y. (2015). Characteristics, vertical structures, and heat/salt transports of mesoscale eddies in the southeastern tropical I ndian O cean. *Journal of Geophysical Research: Oceans*, *120*(10), 6733-6750.

Molodtsov, S., Anis, A., Amon, R. M. W., & Perez‑Brunius, P. (2020). Turbulent mixing in a loop current eddy from glider‑based microstructure observations. *Geophysical Research Letters*, *47*(14), e2020GL088033.

Bebieva, Y., & Timmermans, M. L. (2016). An examination of double‑diffusive processes in a mesoscale eddy in the Arctic Ocean. *Journal of Geophysical Research: Oceans*, *121*(1), 457-475.

Hua, B. L., Ménesguen, C., Le Gentil, S., Schopp, R., Marsset, B., & Aiki, H. (2013). Layering and turbulence surrounding an anticyclonic oceanic vortex: In situ observations and quasi-geostrophic numerical simulations. *Journal of Fluid Mechanics*, *731*, 418-442.

Referee 2

We would like to thank the referee for his/her comments. Please find our point-to-point response below and modifications associated with the revised version in purple.

The authors provide a broad overview of the literature for 'coherent eddies', with a nice introduction on the history of the field of understanding/quantifying eddy coherence. However, when it comes to material coherence, and materially coherent eddies, they've missed quite a number of relevant references from the Lagrangian coherent structures community.

We want to clarify that it was never our intention to overlook or minimize the importance of relevant references published by the Lagrangian community (nor those from the Eulerian community). Our approach in the manuscript was to cite a limited selection of what we consider to be the most relevant and comprehensive works, from which a broader understanding of the literature can be developed. We aimed to provide a focused reference list while acknowledging that these sources represent only a part of the extensive body of work in this field. We thank the referee for providing more relevant references. We will include the suggested references in the revised manuscript.

In the revised version of the manuscript, references have been added. The suggested changes are in lines 57-60 and 74-82.

Specifically, the discussion around how 'kinematic coherence' (KC) is reference frame dependent (and hence, not objective), is missing another key point, that eddies defined by Eulerian (frame-dependent) methods are not materially coherent. Quite a number of studies have shown that ocean eddies, identified from the Eulerian perspective (for e.g., Okubo-Weiss parameter, SSH anomaly), leak material across their identified boundaries relatively quickly (Andrade-Canto et al. 2020, Beron-Vera et al. 2013, Haller et al. 2013, Liu et al. 2019, Serra et al. 2017, Denes et al. 2022). This is a major drawback of using Eulerian approaches for quantifying material coherence, and mass transport by eddies..

Our intention is not to criticise the material frame-indifferent coherent (MC) vortex theory or the concept of "kinematic coherence" in eddies. We fully agree with the mathematical and physical formulation underlying the Lagrangian framework. The primary objective of our study is to explore the impact of material coherence on the thermohaline structure of eddies, which is a direct result of the "coherent" mass transport by oceanic eddies. We will, however, include additional details on the limitations of the Eulerian approach. It is also important to highlight that, while the Lagrangian approach is theoretically robust, it has been misapplied in some published studies. This is particularly the case when ocean observations are used that do not correspond to the required velocity fields, such as when geostrophic surface velocities derived from satellite altimetry maps are used incorrectly in the context of this theory. We can expand also on this point in the revised version of the manuscript.

In the revised version of the article, some references have been added to line 59. We have also reworded the entire section 5.2 to better emphasize the importance of using the surface geostrophic velocity with caution.

I also found the definition of material coherence in the manuscript to be rather qualitative, that a 'vortex ceases to be coherent when it loses its trapped water mass'. How does one determine the point at which a vortex loses its trapped water mass? Recent Lagrangian analyses have considered the coherence of ocean eddies, specifically Abernathey et al. (2018), and Wang et al.(2015), found that only small 'coherent inner cores' of ocean eddies existed over long time periods, while Denes et al. (2022) found ocean eddies may consists of coherent inner cores, and quasi/semi-coherent outer rings (and question the notion that ocean eddies have precise boundaries). This is briefly touched on by the authors on lines 70-71, but I think more is needed here.

Indeed, Lagrangian coherent eddies are theoretical fluid structures that maintain their identity over time, characterized by a consistent material coherence in the flow. They are identified based on their ability to transport mass without "significant mixing" with surrounding fluid, preserving their internal material properties. This coherence is, as the reviewer states, best determined using Lagrangian methods that track the trajectories of fluid particles, ensuring that the eddies' boundaries remain impermeable and that the particles within them move cohesively over a given time period.

The impermeability of eddy boundaries in the Lagrangian sense is mathematically defined by the concept that the boundary is a material surface that does not allow fluid exchange with the surrounding flow. This is quantified using the concept of Lagrangian Coherent Structures (LCS), which are identified using methods such as finite-time Lyapunov exponents (FTLE) or geodesic theory. These methods ensure that particles within the coherent eddy remain within the boundary over time, and particles outside do not cross into it.

A mathematically rigorous way to define impermeability is through the geodesic theory of coherent vortices, which involves identifying elliptic LCS. An elliptic LCS is a closed curve in the flow that acts as a transport barrier, meaning no particles can cross it over the time period considered. The key references for the definition of such "Impermeability" are:

Haller, G., & Beron-Vera, F. J. (2013). "Coherent Lagrangian vortices: The black holes of turbulence." Journal of Fluid Mechanics, 731, R1-R16. This paper introduces the idea of "Lagrangian Coherent Vortices" (LCVs) as impermeable structures in a turbulent flow, using the concept of geodesic theory to define and identify them.

Haller, G. (2015). "Lagrangian Coherent Structures." Annual Review of Fluid Mechanics, 47(1), 137-162. This review further elaborates on the mathematical foundations of impermeable boundaries in coherent structures, providing a broad overview of the techniques used to identify such structures.

However, real ocean mesoscale eddies are less well-defined coherent structures compared to their theoretical Lagrangian representations. The dynamic nature of the ocean—characterized by fast rotation, stratification, continuous interaction with the atmosphere in the upper layers, and interactions with the ocean floor and steep topography in the bottom layers—introduces various instabilities. These instabilities, further influenced by neighboring eddies (since ocean eddies are rarely isolated), can lead to partial permeability of eddy boundaries or even complete disruption and dissipation of the eddy. In particular, these processes are inherently related to the 3D nature of oceanic vortices. The

3D coherence of oceanic vortices is therefore not fully grasped by MC theory applied to surface velocities only.

The reviewer correctly points out that observed ocean eddies generally consist of a coherent inner core (which is not as small as might be assumed; see Barabinot et al., 2024, and the relatively stable core evolution observed in satellite-derived Absolute Dynamic Topography maps: Laxenaire et al., 2019; 2020) surrounded by an outer region that actively exchanges fluids with the surrounding environment. We will incorporate these references, along with those suggested by the reviewer, and integrate the reviewer's valuable insights into the revised manuscript.

In the revised version of the manuscript, we have added the reference to Armi et al. (1989) to emphasize the importance of small-scale processes on the lifetime of mesoscale eddies. We have also incorporated the suggested references and clarifications and refer the reviewer to lines 74-82.

A particular concept raised in 'material coherence' theory is that the coherence occurs over a finite window of time. As described in the manuscript, ocean eddies tend to have lifetimes of months up to years, however, with entrainment and flushing of water, an ocean eddy is not materially coherent over its entire lifetime. To assess the material coherence of an ocean eddy from a single snapshot in time seems rather suspect to me. As the authors note, it is 'impossible to perform temporal studies with in situ data' , however material coherence has a time-evolution component that I think is quite important when assessing mesoscale ocean eddies.

I suggest the authors consider an alternate motivation for the paper. I can see the techniques described and analysis developed as nice ways to 'Reconstruct 3D ocean eddies from 2D in situ transects'.

We thank the reviewer for his suggestion. However, we do not agree with his arguments. Indeed, it is accurate to note that our analysis does not explicitly demonstrate a temporal correspondence between the MC theory and the presence of anomalies in the core of eddies. However, the presence of thermohaline and potential vorticity anomalies is indicative of the existence of water that does not exhibit the same thermohaline properties as the surrounding water.

In this study we want to emphasize how the Lagrangian coherence definition can be coupled to the uniqueness of ocean water masses. In fact, the latter represent distinct "fingerprints" within the ocean, characterized by specific combinations of temperature and salinity that are not randomly distributed but rather result from precise regional conditions. Each water mass originates in a specific region where unique air-sea interactions imprint it with a characteristic temperature-salinity (T-S) signature. Once formed, these water masses are remarkably consistent in their properties as they are advected within the ocean, below the mixing layer, allowing them to be identified and tracked over great distances.

As postulated by the MC theory, the material boundaries maintain the thermohaline characteristics of the transported water, from the eddy region of generation. Consequently, the trapped water exhibits disparities from the surrounding water as a consequence of the eddy motion in the ocean. Anomalies on isopycnals represent a primary indicator of

heterogeneous water masses within eddy cores, enabling the computation of heat and salt content in these structures through the use of in situ data (Aguedjou et al., 2021; Yang et al., 2015; Dong et al., 2017; Dong et al., 2014; Chen et al., 2012). For instance, Armi et al. (1989) show that meddy remains essentially coherent for 2 years before collapsing very rapidly by thernohaline intrusions. This is one of the few instances where we have 4 radials in a single eddy, over several years.

In the revised manuscript, we will more clearly elucidate this evident relationship, which justifies the main objective of the study as succinctly summarized by our proposed title, "Assessing the Material Coherence of Mesoscale Eddies using In Situ Data."

In the revised version of the article, we have changed the title, introduction, and conclusion to better clarify our goals and approach. Thermohaline anomalies, as defined in the article, are evidence of the material coherence of eddies and can be seen as a way to analyze the water transported by eddies without relying solely on temporal data.

**Particular comments:**

1. Methodology I particularly liked the descriptive methodology of the paper, the authors explain and derive their methodology nicely. However, at times I was a little lost. I think the manuscript would benefit from an additional paragraph or two which may provide an overview of the methodology. For example, section 3 describes four identification methods, but no introductory sentence/paragraph which describes why these identification methods are used let alone useful. What alternative methods exist, and why are these four chosen?

Thank you for your valuable suggestion. We will incorporate an additional paragraph in the revised version as proposed. In Section 3, we described four methods that have been widely applied to in situ data analysis. Specifically, the use of T/S anomalies, gradients, and potential vorticity (PV) has been extensively employed to develop diagnostics for eddies sampled during in situ experiments (Aguedjou et al., 2021; Paillet et al., 2002; Paillet et al., 1999; Bosse et al., 2019; Carton et al., 2002). These methods have proven effective in enhancing our understanding of the dynamic properties of oceanic eddies.

We have modified section 3 according to the referee's comment.

2. 'Acronymisation' I felt there were a few too many acronyms, but in cases some things could have been 'acronymised'. For example, at one point the authors use 'AC' and 'C', to describe (from what I understand) anticyclonic and cyclonic eddies (line 268, or 403, but not described anywhere in text). Especially when, in Table 3, 'C' is used to describe 'Complete' eddies. In contrast, to be in line with 'Kinematic coherence' and 'Material coherence' (KC and MC), I think the authors can use 'TC' for 'Thermohaline coherence', which would help with the flow of the discussion in section 5.2, where they describe 'thermohalinely coherent cores', for example.

We thank the reviewer for his/her insightful remarks. We will clarify the points in the revised version of the article according to your suggestions.

Throughout the manuscript, we have included the notation "TC" to make it easier to follow the article.

3. Line 32, the term 'eddy boundary' needs to be placed in context. Eddy boundaries are method dependent, and some literature argues that eddies do not have a sharp boundary. I think this sentence could be qualified further.

We thank the reviewer for his/her remark. We agree that the definition of eddy boundaries is method dependent. We will provide a more detailed explanation of this in the revised manuscript.

In the revised version of the manuscript, we have replaced the notion of "boundary" with the "radius of maximum velocity" in line 42.

4. Line 64-66 on MC theory Some material coherence theory is derived from 'minimal mixing' criteria, specifically, that a boundary of a materially coherent structure is undistorted/unfilamented over a finite-time window, such that diffusive mixing across the boundary is minimised. This sentence can be qualified further.

We thank the reviewer for his/her insightful remark. We agree that the sentence on material coherence theory could benefit from further clarification. In the revised manuscript, we will provide a more detailed explanation of how material coherence theory is derived from the 'minimal mixing' criteria, specifically focusing on how the undistorted and unfilamented boundaries of a materially coherent structure minimize diffusive mixing over a finite-time window.

In the revised version of the manuscript, we have clarified this point from lines 74 to 82.

5. Line 67-68 on number of MC eddies vs. KC eddies. I wasn't sure what the purpose of this sentence is. As the authors suggest, material coherence provides a rather rigorous definition for ocean eddies, and one should expect that more structures will appear from a less restrictive approach than a more restrictive approach. Is this considered a downside of MC approaches, or just a statement of fact?

We thank the reviewer for his/her thoughtful remark. We will revise the tone of this sentence to ensure clarity. The sentence is intended as a statement of fact, highlighting that the more rigorous definition of ocean eddies provided by material coherence may result in fewer identified structures compared to less restrictive approaches. This distinction is significant, as it directly impacts the assessment of heat and salt transport by mesoscale eddies.

In the revised version of the manuscript, we have modified the sentence in line 85.

6. Describing the trapped water as 'heterogeneous' (e.g. line 85, 306) I found this a little confusing, as the trapped water mass is typically homogeneous within the eddy core, but distinct from the surrounding water mass, as it is 'characteristic of the region of formation of that eddy'. I would think of this water mass as 'distinct' .

We thank the reviewer for his/her suggestion. We will change the word "heterogeneous" by "distinct".

In the revised version of the manuscript, we have changed the word "heterogeneous" with "distinct" throughout the manuscript.

7. Line 91 – providing a first answer to the question of the 3D material coherence of eddies I'm not entirely sure what the question actually is. That 3D material coherence of eddies can be assessed using water masses and the PV approach? Certainly, 3D material coherence of eddies has been considered in the past, see Froyland et al. (2012).

We thank the reviewer for his/her comment. We will rephrase the question in the manuscript to ensure clarity. The intent is to explore how 3D material coherence of eddies can be assessed using water masses and the PV approach. While we acknowledge that 3D material coherence has been addressed in previous studies, such as Froyland et al. (2012), our focus is on providing a new perspective through these specific methodologies.

In the revised version of the manuscript, we have modified the paragraph from lines 105 to 116.

8. Definition of the center of an eddy The authors make contrasting statements '… is reached at the eddy center (region where the velocity tends to zero)' on line 100, and 'The center of the eddy is defined as the point where the mean tangential velocity is maximum' , and '… cross the exact eddy center, defined as the location of the zero velocity' on lines 344-345. To me, these three statements are inconsistent. Additionally, in the first instance the eddy center is a region, in the second it is either a point or a curve, and in the third it a point (possibly a region?).

We thank the reviewer for his/her detailed remark. We understand the concern regarding the apparent inconsistencies in the definitions of the eddy center. To clarify, on a 2D vertical section, the eddy center is identified as the region where the in situ velocity orthogonal to the ship tends to zero, indicating that this center lies within the plane of the section. In 3D, however, the eddy center is determined at each depth level on a 2D horizontal plane using the Nencioli routine on in situ data (Nencioli et al., 2008). This routine involves constructing a grid with a specified resolution around the ship's track, and for each grid point, the mean radial and tangential velocities are calculated, assuming that point is the eddy center. The grid point with the minimum radial velocity or maximum tangential velocity is then considered the eddy center at that depth level. We will revise the manuscript to ensure these definitions are consistently and clearly presented. For further details, we encourage the reviewer to refer to Nencioli et al. (2008).

In the revised version of the manuscript, we have changed the lines 114 and 189.

9. Line 244 'These values are small compared to the first-order terms' I would disagree, in the horizontal case they are of the same order of magnitude. They are smaller, but not negligibly small. Is this not important?

We thank the reviewer for his/her thoughtful remark We agree that while the values are smaller, they are not negligibly small, particularly in the horizontal case where they are of the

same order of magnitude. To address this, we also tested a second-order centered scheme () which was not detailed in the article. This approach resulted in less than a 5% change in the outcomes. We will clarify this point in the revised manuscript to acknowledge the importance of these terms.

In the revised version of the manuscript, we have modified the beginning of Section 3.2 from lines 260 to 262.

10. Lines 300-301 and lines 441-442 – Motivation The motivation for the analysis described in both these sections would be great to see in the introduction. In both cases, I did not know this would form part of the analysis until I reached these sentences.

We thank the reviewer for his/her remark. The revised manuscript will be modified accordingly.

In the revised version of the manuscript, we have clarified our objectives in the introduction from lines 117 to 129.

11. Line 361 – 'To reduce this uncertainty, …' By computing volumes for a subset of eddies, you aren't exactly reducing uncertainty. Rather, you are only showing results for eddies where the uncertainty isn't too large. I would rephrase this.

We thank the reviewer for his/her remark. This sentence will be rephrased accordingly.

In the revised version of the manuscript, this sentence has been rephrased in line 376.

12. Layout of Figures 4 and 5 Use of figure labelling would be really useful for both of these figures, using a), b) etc. Also, the change in orientation for the bottom panels of Figure 4 I found rather jarring, is there a reason to have these laid out vertically rather than horizontally like the top panel of Figure 4 and all of Figure 5? Lastly, I suggest a second diverging colorbar for the salinity anomaly (different to the thermal anomaly colorbar).

We thank the reviewer for his/her valuable remark. We will revise Figures 4 and 5 to improve clarity. In the revised version of the manuscript, we will include figure labeling (a, b, etc.) to make the figures easier to reference. Additionally, we will adjust the orientation of the bottom panels in Figure 4 to match the horizontal layout of the top panel and Figure 5 for consistency. We will also introduce a separate diverging colorbar for the salinity anomaly to distinguish it from the thermal anomaly colorbar.

In the revised version of the manuscript, Figures 4 and 5 have been changed per the reviewer's suggestion. Figure captions have been added and the salinity color bar has been changed to contrast with the temperature color bar.

13. Lines 458-459 – 'Lagrangian studies suggest that the ability of eddies to trap a water mass is a consequence of closed trajectories'. I'm not certain what you mean by this. What is a closed trajectory in this instance? A citation would be very useful.

We thank the reviewer for his/her remark. To clarify, by "closed trajectories," we are referring to the circular or looping paths that fluid particles follow within an eddy, which allow the eddy to trap water masses effectively. We will add a citation to relevant Lagrangian studies in the revised version of the manuscript to support this statement and provide further context.

In the revised version of the manuscript, we have added some references in line 482 to corroborate our argument.

14. Figure 7a. Plotted velocity vectors - I find these velocity vectors distract from the plots, especially in panel g), where I feel the figure is too busy to see what is really going on. b. Panel title (a) – 'fev', and panel title (i) – missing date (same on Figure 8)

We appreciate the reviewer's insightful comments. In response, we will adjust Figure 7 by either reducing the density of the velocity vectors or providing an alternative visualization to reduce clutter, particularly in panel g). Additionally, we will correct the panel title in Figure 7a from 'fev' to the appropriate label and ensure the date is added to panel i) in both Figure 7 and Figure 8 in the revised manuscript.

In the revised version of the manuscript, Figure 7 has been split in two in order to increase the size of each panel and improve the readability. It was important to keep the difference between the geostrophic surface properties (ADT and eddy centers) and the properties derived from ADCP measurements (centers obtained by the routine of Nencioli et al (2008)).

15. Figure 9 colorbar and panel titles I would change the colorbar to something that contrasts better with the isopycnal, EPV, and salinity curves (being black and blue), otherwise it is very hard to appreciate these curves. I also found the panel titles to be rather confusing and could do with an update.

We thank the reviewer for his/her valuable feedback. We will revise the colorbar in Figure 9 to provide better contrast with the isopycnal, EPV, and salinity curves, enhancing their visibility. Additionally, we will update the panel titles to improve clarity and reduce any potential confusion. These changes will be reflected in the revised version of the manuscript.

In the revised version of the manuscript, figure 9 has been modified following the referee's advice. Figure 9 appears clearer now.

16. Line 519-520 – 'The reconstructed volume associated with the thermal anomaly is the most convex of all shapes'. I'm not sure what you mean by this, nor am I sure if I agree with the statement. In fact, some of the shapes plotted in Figure 10 aren't convex (panel (c) for example).

We thank the reviewer for his/her remark. We agree that the term "convex" is inappropriate here. The appropriate word is "connected". This sentence will be modified in the revised version of the manuscript.

In the revised version of the manuscript, the word "convex" has been changed with "connected" in line 543.

17. Figure 10 – panel titles If you could add titles to each of the panels (not just in the figure caption), that would really help guide the reader. Also, given the circularity constraint, I think

each of these plots should have equal x-y aspects, figure (c) looks out of shape due to the non-equal aspect ratio.

We thank the reviewer for his/her remark. We will modify accordingly the figures in the revised version of the manuscript.

In the revised version of the manuscript, figure 10 has been modified according to the referee's comment.

18. Lines 525 – 533 regarding Figures 11 and 12 The authors say 'This figure refers to the ellipses (E1) mentioned earlier' and later '(ellipses E2)', but the only mention I could find to E1 or E2 were in the caption of figure 9. Where do these numbers come from, and how were they computed? Secondly, what is the purpose of an elliptical reconstruction? That was not clear to me.

We thank the reviewer for pointing out this issue. The references to (E1) and (E2) were intended to refer to the ellipses described in Appendix B. We apologize for the confusion and will correct these mentions in the revised manuscript to clearly associate them with the appropriate sections.

Regarding the purpose of the elliptical reconstruction, as outlined in Chen et al. (2019), ellipses often provide the best fit for the shape of eddies observed in altimetric maps. Furthermore, an eddy exhibiting barotropic instability with a second mode, as discussed by Marez et al. (2020), typically assumes an elliptical form. This makes an elliptical reconstruction a suitable approach for accurately characterizing the eddy structure in our study. We will clarify this rationale in the revised manuscript.

In the revised version of the manuscript, we have added a paragraph in lines 412 to solve this issue.

19. Line 569-570 – 'This paper presents an evaluation of the thermohaline coherence of mesoscale eddies based on in situ data collected during several cruises…' While I agree with this statement, the title of the manuscript is on assessing 'material coherence'. While I don't think the manuscript has addressed the material coherence of ocean eddies, this statement in the conclusion is at odds with the title and abstract of the manuscript.

We thank the reviewer for this insightful comment. We acknowledge the inconsistency between the manuscript's title and the focus on thermohaline coherence in the conclusion. To resolve this, we will revise the title and abstract to more accurately reflect the focus on thermohaline coherence. Alternatively, we will adjust the discussion to better address aspects of material coherence, ensuring alignment with the title. The revised manuscript will present a consistent narrative that accurately represents the study's objectives and findings.

In the revised version of the manuscript, the conclusion has been rewritten to better address the issues we raised in the introduction. The conclusion should now be clearer.

**Typos/grammar/missing info:**

Line 58 – there is an erroneous space between 'velocity fields' and the comma.

Line 128 "boundary current systems off Labrador" - Do you mean Labrador sea?

Line 196 – 'Localization' should be 'Localisation'.

Lines 105, 192, 438 – 'confront' could be 'compared'.

Line 195-197 – Could you provide a citation to the data described?

Line 233, Fig 4 caption, Fig 5 caption – you use 'gyre', do you mean eddy?

Line 280 – 'the EPV id the ocean at rest', what do you mean by 'rest'?

Equation 7 – $f\_0$ is defined, but $f$ is not defined.

Line 321 – what are \overbar{b} and b' ?

Line 327 – 'adimensionalizing' should be 'nondimensionalising'.

Line 364 – 'analytically', I wasn't sure what you meant by this.

Line 366 – 'descrided' should be 'described'.

Line 564 – 'this' should be 'thus'.

Line 582 – 'This result corroborates the findings of previous studies…' citations are needed for this statement.

We sincerely thank the reviewer for his/her thorough review and valuable comments. We will carefully address all the minor suggestions, ensuring that the revised manuscript is fully corrected and improved accordingly.

Once again, we would like to thank the reviewer for his/her valuable review. In the revised version of the manuscript, we have carefully addressed all the major and minor suggestions and hope our work is now more rigorous.

**References**

Aguedjou, H. M. A., Chaigneau, A., Dadou, I., Morel, Y., Pegliasco, C., Da‑Allada, C. Y., & Baloïtcha, E. (2021). What can we learn from observed temperature and salinity isopycnal anomalies at eddy generation sites? Application in the tropical Atlantic Ocean. *Journal of Geophysical Research: Oceans*, *126*(11), e2021JC017630.

Chen, G., Gan, J., Xie, Q., Chu, X., Wang, D., & Hou, Y. (2012). Eddy heat and salt transports in the South China Sea and their seasonal modulations. *Journal of Geophysical Research: Oceans*, *117*(C5).

Dong, C., McWilliams, J. C., Liu, Y., & Chen, D. (2014). Global heat and salt transports by eddy movement. *Nature communications*, *5*(1), 3294.

Dong, D., Brandt, P., Chang, P., Schütte, F., Yang, X., Yan, J., & Zeng, J. (2017). Mesoscale eddies in the northwestern Pacific Ocean: Three‑dimensional eddy structures and heat/salt transports. *Journal of Geophysical Research: Oceans*, *122*(12), 9795-9813.

Yang, G., Yu, W., Yuan, Y., Zhao, X., Wang, F., Chen, G., ... & Duan, Y. (2015). Characteristics, vertical structures, and heat/salt transports of mesoscale eddies in the southeastern tropical I ndian O cean. *Journal of Geophysical Research: Oceans*, *120*(10), 6733-6750.

Molodtsov, S., Anis, A., Amon, R. M. W., & Perez‑Brunius, P. (2020). Turbulent mixing in a loop current eddy from glider‑based microstructure observations. *Geophysical Research Letters*, *47*(14), e2020GL088033.

Bebieva, Y., & Timmermans, M. L. (2016). An examination of double‑diffusive processes in a mesoscale eddy in the Arctic Ocean. *Journal of Geophysical Research: Oceans*, *121*(1), 457-475.

Hua, B. L., Ménesguen, C., Le Gentil, S., Schopp, R., Marsset, B., & Aiki, H. (2013). Layering and turbulence surrounding an anticyclonic oceanic vortex: In situ observations and quasi-geostrophic numerical simulations. *Journal of Fluid Mechanics*, *731*, 418-442.

Paillet, J., Le Cann, B., Carton, X., Morel, Y., & Serpette, A. (2002). Dynamics and evolution of a northern meddy. *Journal of Physical Oceanography*, 32(1), 55-79.

Carton, X., Chérubin, L., Paillet, J., Morel, Y., Serpette, A., & Le Cann, B. (2002). Meddy coupling with a deep cyclone in the Gulf of Cadiz. *Journal of Marine Systems*, 32(1-3), 13-42.

Paillet, J. (1999). Central water vortices of the eastern North Atlantic. *Journal of physical oceanography,* 29(10), 2487-2503.

de Marez, C., Meunier, T., Morvan, M., L'hégaret, P., & Carton, X. (2020). Study of the stability of a large realistic cyclonic eddy. *Ocean Modelling*, 146, 101540.

Bosse, A., Fer, I., Lilly, J. M., & Søiland, H. (2019). Dynamical controls on the longevity of a non-linear vortex: The case of the Lofoten Basin Eddy. *Scientific reports*, 9(1), 13448.

Armi, L., Hebert, D., Oakey, N., Price, J. F., Richardson, P. L., Rossby, H. T., & Ruddick, B. (1989). Two years in the life of a Mediterranean salt lens. *Journal of Physical Oceanography,* 19(3), 354-370.

Referee 3

We thank the reviewer for his comments. Please find our point-by-point response below, with changes associated with the revised version in purple.

This manuscript details the analysis of in situ float data along with CTD sections to asses how eddies transport material. This work represents a valuable contribution to the study of eddy dynamics and can be strengthened by the addition of specific derivations to go along w the text. My suggestion is that the authors consider my comments and maybe add some proofs that the methods of defining material coherence are similar or different. If this is addressed it is my opinion that this paper be published.

My recommendation is to back up the arguments made in to the introduction. The authors compare multiple methods to estimate eddy transport and do so only describing similarities and differences. This section needs to be analytical and needs to show mathematically what is described in the text. At the present it is not sufficient to just say two methods are the same or different with out showing it.

As to the analysis and results, I find them truly interesting and am excited to have them become part of the literature. The strengthening of the introduction in my opinion would make this a very valuable paper.

We thank the reviewer for his/her positive comment and his/her suggestion.

However, we are not sure to understand what is stipulated in the comment especially regarding the addition of mathematical proofs. We are not sure whether this relates to the confrontation between Eulerian and Lagrangian visions or to the comparison of criteria applied to a vertical section.

The mathematical confrontation between Eulerian and Lagrangian vision was treated in great details by the papers of Haller et al. cited in the introduction. Moreover, several Eulerian and Lagrangian criteria are used to detect the material coherent core of eddies making the mathematical development long to perform.

In addition, criteria are tested through their ability to keep particles in the eddy cores and are better assessed using numerical simulations. However, we remain open to include an analytical idealized case in the introduction of the revised version that highlights the difference between an Eulerian and a Lagrangian criterion.

Regarding the comparison of Eulerian criteria applied to vertical ship sections, Barabinot et al. (2024) proposes a comparison of these criteria which we supplement in part 3 of our manuscript.

For now, we have not changed the introduction as suggested by the reviewer for several reasons.

The mathematical and numerical confrontation between Lagrangian and Eulerian criteria has already been established in several studies or simulations, and we believed that we could not fully illustrate this issue by introducing only a few formulas (Andrade-Canto et al. 2020,

Beron-Vera et al. 2013, Haller et al. 2013, Liu et al. 2019, Serra et al. 2017, Denes et al. 2022).

In our opinion, introducing the mathematical development in a four-page introduction would lead to an unclear development and could confuse the readers. Moreover, the confrontation between Lagrangian and Eulerian visions remains complex, as there are several criteria (Haller et al. (2015)), and we believe that we cannot cover all the literature with our introduction.

We intended to explain the main differences between Eulerian and Lagrangian perspectives by reviewing the main physical concept of the literature, which we believe is sufficient to understand our objectives. The fact that Lagrangian criteria take into account trajectories and thus the history of the flow instead of streamlines, as used for Eulerian criteria, is already described in the article and consists in the main difference between these visions.

For clarity's sake, we regret to reject the reviewer's suggestion for now. However, we remain open to other suggestions for improving our introduction.

---

## Referee Report (RR1)

Referee's Report on "**Assessing the Material Coherence of Mesoscale Eddies as described from In Situ Data**" by Barabinot et al. submitted to Ocean Science explored the material coherence of mesoscale eddies using in situ data.

From my point of view, this manuscript is not well organized. The key points/findings are not clearly shown in the abstract and conclusion sections. This introduction is very long, but the scientific questions are not well introduced/addressed, and I do not think the authors need some many short paragraphs in the introduction section. The authors also use two sections (Section 3 and section 4) to introduce the methods, it may be better to incorporate them into one section. The results shown here are not very convincing to me. There are no discussion section in this manuscript. Without discussions, the reader would not know the limitations and implications of the manuscript. Last but not the least, there are many typos and nonstandard writing. I suggest the authors carefully double check the writing, calculation and statements.

Main concerns:

1. I do not think the authors need to get the climatology temperature and salinity profiles from two different datasets (ARGO and WOA2023). How

are the temperature and salinity anomalies calculated? Are the seasonal cycle removed?

2. In the vertical, eddies will tilt with depth. In horizontal, most eddies are not circular or elliptical in shape. The 3D reconstructions of eddies shown in figures 10-12 are not convincing.

3. The authors argue that surface mesoscale eddies detected from satellite altimetry data do not match with subsurface eddies from in situ data. There are many factors may cause this, such as the TOEddies algorithm, the methods and data used to extract the subsurface temperature/salinity anomalies here. The authors should double this before they draw any conclusions.

**Monir comments**

1. Section 3.2

$1.10^{-6}°C.m^{-1}$ , $0.6.10^{-4}°C.m^{-1}$ , $7.6.10^{-6}°C.m^{-1}$ and $2.5.10^{-2}°C.m^{-1}$ are incorrect.

2. Each panel should be labeled as in figures 4-5. The top right colorbar for the salinity seems incorrect to me.

---

## Author Response (AR2)

Referee 4

We would like to thank the referee for his/her comments. Please find our point-to-point response below.

**Referee's Report on "Assessing the Material Coherence of Mesoscale Eddies as described from In Situ Data" by Barabinot et al. submitted to Ocean Science explored the material coherence of mesoscale eddies using in situ data.**

*From my point of view, this manuscript is not well organized. The key points/findings are not clearly shown in the abstract and conclusion sections. This introduction is very long, but the scientific questions are not well introduced/addressed, and I do not think the authors need some many short paragraphs in the introduction section. The authors also use two sections (Section 3 and section 4) to introduce the methods, it may be better to incorporate them into one section. The results shown here are not very convincing to me. There are no discussion section in this manuscript. Without discussions, the reader would not know the limitations and implications of the manuscript. Last but not the least, there are many typos and nonstandard writing. I suggest the authors carefully double check the writing, calculation and statements.*

We would like to thank the reviewer for his/her thorough review. We are grateful for the effort invested in evaluating our work and for highlighting important points for consideration. While we value these insights, we respectfully disagree with some of the major concerns raised, as addressed in detail in our responses below.

*Main concerns:*
*1. I do not think the authors need to get the climatology temperature and salinity profiles from two different datasets (ARGO and WOA2023). How are the temperature and salinity anomalies calculated? Are the seasonal cycle removed?*

We thank the reviewer for their comment and appreciate the opportunity to clarify this point. The methodology for computing thermohaline anomalies on isopycnals is detailed in Section 3.1. For a given eddy, we aimed to construct a local climatological average. To achieve this, we applied the methodology described in Laxenaire et al. (2019), which utilizes ARGO float profiles and has been shown to be effective for studying eddy dynamics.

This approach calculates a climatological average centered on the position of the eddy during the specific month it was sampled. The climatological averages of temperature and salinity on geopotential levels were derived from ARGO float profiles collected over a 20-year period within a small area surrounding the sampled eddy. These averages are assumed to represent the baseline state of the ocean at the specific location and time of sampling, allowing for the construction of precise, localized, and temporally consistent climatological averages.

Once anomalies were computed, we imposed a threshold on the anomaly values using WOA2023, which provides robust standard deviation estimates. The use of WOA2023 was motivated by its tabulated standard deviations, which were essential for validating our

climatology derived from Laxenaire et al. (2019). WOA2023 served as a reference for this validation, and its  standard deviations values were used to compare against the computed anomalies.

To address the reviewer's final question, we did not remove the seasonal cycle. The climatological average of temperature and salinity on geopotential levels was calculated using ARGO float profiles collected over 20 years within a small area surrounding the sampled eddy. Importantly, this average incorporates profiles measured during the same month the eddy was sampled. Therefore, the seasonal cycle is inherently accounted for in this methodology and does not need to be explicitly removed.

*2. In the vertical, eddies will tilt with depth. In horizontal, most eddies are not circular or elliptical in shape. The 3D reconstructions of eddies shown in figures 10-12 are not convincing.*

We thank the reviewer for his/her comment and appreciate the opportunity to clarify this point. As demonstrated in the supplementary materials, the centers of the eddies exhibit minimal tilting with depth, at least in 2D vertical sections. The tilting of eddy centers is typically limited to a few kilometers, which is negligible compared to their overall radius.

To support this observation, we have plotted the velocity fields of six representative eddies below. In each case, a vertical purple line has been added to highlight the vertical alignment of the eddy center (marked in dashed blue line). This vertical reference is built such that the shallower position of the eddy center belongs to this straight line. For each eddy, we provide the maximum deviation of the eddy center with respect to this vertical reference and express it as a percentage of the eddy maximum radius. Results are presented below. It is worth noting that the maximum deviation is an integer as data have been interpolated on a horizontal grid of 1 km resolution. Results show that the deviation of eddy centers from the vertical does not exceed 10% of the eddy maximum radius.

It is also worth noting that our methodology described in Figure 3 and in part 4.2 does take into account the eddy tilting. Here is our paragraph in the article: *"In summary, the approach consists of three steps. First,  a criterion (the outermost closed contour of a given size) is chosen to delimit the materially coherent eddy core from its surroundings on the 2D vertical slice. Then, compute the position of the apparent eddy center as the location where the orthogonal velocity $V_o$ is zero and the eddy radius $L(z)$ associated with the selected criterion. Finally, calculate the approximate volume as a sum of elementary cylinders."* As eddy centers are almost perfectly vertical in their thermohaline coherent core (see results below), the tilting appears negligible in Figure 10, 11 and 12. However, as mentioned, the tilting is well taken into account when reconstructing eddies.

Additionally, based on geophysical fluid dynamics and observational evidence discussed throughout Section 4, eddies are generally axially symmetric on average (Chaigneau et al., 2009; Chelton et al., 2011). They often exhibit mode-1 and mode-2 deformations. Furthermore, as written in the manuscript and corroborated by Chen et al. (2019), an elliptical shape best represents the geometry of eddies.

Subsurface AE sampled during EUREC4A-OA experiment: The maximum velocity radius is provided in Table 3 and equals 71 km. Above -400 m, the deviation of the center is inferior to 2 km which represents 2.8% of the eddy maximum radius. Below -400 m, the slope of the deviation is on average 0.04 km/m. Therefore, the maximum deviation of the center is reached at -700 m and its value is 12 km. Considering the full 2D vertical section, the maximum deviation corresponds to 16.9% considering the entire structure. However, what is important is the coherent core. Reducing our analysis to the thermohaline coherent core, the maximum deviation is reached at -630 m and its value is 7 km which represents 9.8% of the eddy maximum radius.

[Figure]

2 AEs sampled during M124 experiment: For these 3 eddies, as the resolution of ADCP is 32m on the vertical (see in Table 2.), eddies appear more barotropic. In these 3 cases, the maximum deviations of the center are respectively 1 and 2 km. These deviations are almost invisible and are reached around -60 m depth. The eddy maximum radii are respectively 58 and 55 km. The maximum deviations of center are respectively 1.7% and 3.6% of eddies maximum radius.

[Figure]

Subsurface AE sampled during KB2016 experiment: In that case, the maximum deviation of the center is 2 km. The maximum velocity radius is 15 km. The maximum deviation of the center represents 13.3% considering the entire section. In the thermohaline coherent core

(here between isopycnals 27.8 kg/m^3 and 27.9 kg/m^3), the center is perfectly vertical (see the blue dashed line). Deviations of the center only occur when crossing isopycnals 27.8 kg/m^3 and 27.9 kg/m^3.

[Figure]

AE sampled during Physindien 2011: Here, the maximum deviation of the center is 6 km. The maximum velocity radius is 95 km. The maximum deviation of the center thus represents 6.3 % of the eddy maximum radius. Moreover, as shown in the following figure, the main deviation occurs at the surface. Below -50 m depth, the eddy center remains quite vertical and, taking this depth level as a new reference, the maximum deviation of the center is therefore 1 km and represents 1.1% of the eddy maximum velocity radius.

[Figure]

*3. The authors argue that surface mesoscale eddies detected from satellite altimetry data do not match with subsurface eddies from in situ data. There are many factors that may cause this, such as the TOEddies algorithm, the methods and data used to extract the subsurface temperature/salinity anomalies here. The authors should double this before they draw any conclusions.*

We thank the reviewer for their comment and appreciate the opportunity to clarify this point. Satellite altimetry measures the dynamic height of the ocean, a variable representing the integrated vertical thermohaline structure. Specifically, the absolute dynamic topography (ADT) on which the TOEddies algorithm is based is derived from satellite altimetry. ADT reflects the integral properties of the water column, which are influenced by the local vertical thermohaline structure (water masses). Intense mesoscale eddies imprint their signature on this property, whether they move at the ocean surface or below it.

When the upper ocean stratification is relatively weak (as is often the case at mid- and high-latitudes), deep subsurface eddies—such as Mediterranean Outflow Eddies (Meddies), which are typically centered at depths of 600–1000 m (Ienna et al., 2022; Ciani et al., 2017)—can still influence ADT. However, the surface geostrophic velocities derived from this ADT footprint do not necessarily represent the core velocities of the eddies creating the ADT anomaly. For example, the eddy core may begin at 200 m depth or deeper, as discussed by Laxenaire et al. (2019, 2020). Meddies are a good illustration of this: while their presence impacts the ADT, the ADT signal is also affected by the thermohaline stratification of the water column above them. It is also worth noting that ADT can be seen as a streamfunction and is thus not conserved contrary to the Potential Vorticity (PV) which is a Lagrangian invariant. This is the reason why the streamfunction is often not enough to locate an eddy.

In our article, the main conclusion is that the material core of eddies, identified by thermohaline anomalies on isopycnals, often lies below the ocean surface and thus cannot be fully captured by traditional altimetry. Specifically, applying Lagrangian methods to satellite data is insufficient to detect subsurface trapped waters. Regarding the velocity field, however, altimetry matches in situ measurements reasonably well at mid-latitudes (see Figure 9 in our manuscript).

We acknowledge the reviewer's suggestion to double-check our methodology and would like to respectfully emphasize the following points:

- The **TOEddies algorithm** has been demonstrated to effectively detect mesoscale eddies in comparison to traditional products, as shown in numerous studies (Laxenaire et al., 2018, 2019, 2020; Manta et al., 2021; Chen et al., 2022; Subirade et al., 2023). The TOEddies Atlas is now available as open source:
  *Laxenaire, R., Guez, L., Chaigneau, A., Isic, M., Ioannou, A., and Speich, S.: TOEddies Global Mesoscale Eddy Atlas Colocated with Argo Float Profiles, https://doi.org/10.17882/102877 , 2024.*

  and it is described (and compared with other atlases) in this new published manuscript:
  *Ioannou, A.; Guez, L.; Laxenaire, R.; Speich, S. Global Assessment of Mesoscale Eddies with TOEddies: Comparison Between Multiple Datasets and Colocation with*

*In Situ Measurements. Remote Sens. **2024**, 16, 4336.*
*https://doi.org/10.3390/rs16224336*

- The use of **thermohaline anomalies** has also proven highly effective for analyzing eddy dynamics and trapping, given that diffusion is negligible at the mesoscale (Robinson et al., 1983; Paillet et al., 2002; Carton et al., 2001, 2010).

We hope this clarifies the robustness of our approach and the validity of our conclusions.

*Monir comments*
*1. Section 3.2 1.10-6°C.m-1 , 0.6.10-4°C.m-1 , 7.6.10-6°C.m-1 and 2.5.10-2°C.m-1 are incorrect.*

We thank the reviewer for his/her comment. However, the values mentioned are not present in the current version of our article. Section 3.2 does not include these values. It is possible that the reviewer may have referred to a previous version of the manuscript.

*2. Each panel should be labeled as in figures 4-5. The top right colorbar for the salinity seems incorrect to me.*

We thank the reviewer for his/her comment. We respectfully request the reviewer to clarify their remark, as we are unsure which figure is being referred to. In the current version of the manuscript, all figure panels are labeled with letters.

Refs:

Carton, X. (2001). Hydrodynamical modeling of oceanic vortices. *Surveys in Geophysics*, *22*(3), 179-263.

Carton, X. A. V. I. E. R., Daniault, N., Alves, J. O. S. E., Cherubin, L. A. U. R. E. N. T., & Ambar, I. (2010). Meddy dynamics and interaction with neighboring eddies southwest of Portugal: Observations and modeling. *Journal of Geophysical Research: Oceans*, *115*(C6).

Chaigneau, A., Eldin, G., & Dewitte, B. (2009). Eddy activity in the four major upwelling systems from satellite altimetry (1992–2007). *Progress in Oceanography*, *83*(1-4), 117-123.

Chelton, D. B., Schlax, M. G., & Samelson, R. M. (2011). Global observations of nonlinear mesoscale eddies. *Progress in oceanography*, *91*(2), 167-216.

Chen, G., Han, G., & Yang, X. (2019). On the intrinsic shape of oceanic eddies derived from satellite altimetry. *Remote Sensing of Environment*, *228*, 75-89.

Chen, Y., Speich, S., & Laxenaire, R. (2022). Formation and transport of the South Atlantic subtropical mode water in Eddy‑Permitting observations. *Journal of Geophysical Research: Oceans*, *127*(1), e2021JC017767.

Ciani, D., Carton, X., Aguiar, A. B., Peliz, A., Bashmachnikov, I., Ienna, F., ... & Santoleri, R. (2017). Surface signature of Mediterranean water eddies in a long-term high-resolution simulation. *Deep Sea Research Part I: Oceanographic Research Papers*, *130*, 12-29.

Ienna, F., Bashmachnikov, I., & Dias, J. (2022). Meddies and their sea surface expressions: Observations and theory. *Journal of Physical Oceanography*, *52*(11), 2643-2656.

Ioannou, A.; Guez, L.; Laxenaire, R.; Speich, S. Global Assessment of Mesoscale Eddies with TOEddies: Comparison Between Multiple Datasets and Colocation with In Situ Measurements. Remote Sens. **2024**, 16, 4336. https://doi.org/10.3390/rs16224336

Laxenaire, R., Speich, S., Blanke, B., Chaigneau, A., Pegliasco, C., & Stegner, A. (2018). Anticyclonic eddies connecting the western boundaries of Indian and Atlantic Oceans. *Journal of Geophysical Research: Oceans*, *123*(11), 7651-7677.

Laxenaire, R., Speich, S., & Stegner, A. (2019). Evolution of the thermohaline structure of one Agulhas ring reconstructed from satellite altimetry and Argo floats. *Journal of Geophysical Research: Oceans*, *124*(12), 8969-9003.

Laxenaire, R., Speich, S., & Stegner, A. (2020). Agulhas ring heat content and transport in the South Atlantic estimated by combining satellite altimetry and Argo profiling floats data. *Journal of Geophysical Research: Oceans*, *125*(9), e2019JC015511.

Laxenaire, R., Guez, L., Chaigneau, A., Isic, M., Ioannou, A., and Speich, S.: TOEddies Global Mesoscale Eddy Atlas Colocated with Argo Float Profiles, https://doi.org/10.17882/102877 , 2024.

Manta, G., Speich, S., Karstensen, J., Hummels, R., Kersalé, M., Laxenaire, R., ... & Meinen, C. S. (2021). The South Atlantic meridional overturning circulation and mesoscale eddies in the first GO‑SHIP section at 34.5° S. *Journal of Geophysical Research: Oceans*, *126*(2), e2020JC016962.

Paillet, J., Le Cann, B., Carton, X., Morel, Y., & Serpette, A. (2002). Dynamics and evolution of a northern meddy. *Journal of Physical Oceanography*, *32*(1), 55-79.

Subirade, C., L'hégaret, P., Speich, S., Laxenaire, R., Karstensen, J., & Carton, X. (2023). Combining an Eddy Detection Algorithm with In-Situ Measurements to Study North Brazil Current Rings. *Remote Sensing*, *15*(7), 1897.

We would like to thank the referee for his/her comments. Please find our point-to-point response below.

*Mesoscale ocean eddies are known to play a significant role in transporting heat, momentum, water mass, and biota large distances across ocean basins, thereby impacting productivity, biogeochemical properties, circulation, and climate. While there have been many studies that have identified and tracked mesoscale eddies as persistent signatures in surface fields such as sea surface height, temperature, or Okubo Weiss parameter, or as materially coherent regions of fluid derived from upper ocean velocity estimates, relatively few studies have studied the subsurface signature of eddies. This is important on two counts: not all eddies have a detectable surface expression, and estimates of transport by ocean eddies depends sensitively on the subsurface structure of the eddy.*

*This paper attempts to characterise the material coherence of mesoscale ocean eddies using in situ data gathered from a large number of oceanographic cruises in different ocean basins. This is a challenging task because very rarely do we have repeat observations of subsurface water mass properties in the same ocean feature over significant periods of time. Two notable exceptions (that I am aware of) are the classic study of Armi et al. (1989) -- cited in this article -- which tracked a Mediterranean salt lens (or "meddy") over a period of two years, and a more recent study by Rykova and Oke (2022) that analysed two Tasman Sea warm-core eddies using Argo float profiles co-located within the eddies.*

*There are many different -- sometimes contradictory -- definitions of eddy "coherence" in the oceanographic literature. The authors categorise previous analyses of mesoscale ocean eddies as identifying features that exhibit either "kinematic coherence" or "material coherence". The former features exhibit persistent signatures in a fixed (Eulerian) frame that can be tracked over long times/distance, while the latter are identified by either closed material boundaries that minimize mixing with the surrounding fluid or regions of the fluid that remain coherent and retain mass as they are advected by the flow. A similar categorisation is used by Denes et al. (2022), who distinguish between the "persistence" of eddy signals over long time periods (i.e. kinematic coherence) and the material coherence of these features, which can be over much shorter timescales. [Full disclosure: I am one of the co-authors of this paper.] As pointed out by the authors of the present article, these concepts are not mutually exclusive, but they are also not equivalent.*

*Here, the authors introduce a new approach which they call "thermohaline coherence", which seeks to describe the tendency of eddies to trap water mass with a distinctive signature of temperature and salinity and transport it, without modification, far from its region of formation. This idea is not without precedence: Robinson (1983) and Chelton et al. (2011) also defined eddies by identifying coherent anomalies of oceanographic variables. Unlike one of the reviewers, I don't see this approach as conceptually different from or inconsistent with material coherence --- the one implies the other. Rather, I view this as an alternative approach to identify material coherence of ocean eddies without a time history of observations. This is crucial. As described above, repeat subsurface observations of ocean eddy properties are vanishingly rare. Instead, the authors identify material coherence by looking for anomalous water mass properties in individual oceanographic sections. The*

*approach might be likened to a detective identifying a thief based on his possession of stolen goods, even though the detective did not follow the thief home from the scene of the crime.*

*I found this to be a thoughtful, well-designed, and well-executed study of a large oceanographic data set. Some of the methods and results will be of great interest to both theorists and observers, and I am happy to recommend publication with (very) minor edits. My comments below are mostly cosmetic. I look forward to seeing this when it is published.*

*S. Keating, Sydney, Australia.*

We would like to thank the reviewer for his thorough review and constructive feedback.We sincerely appreciate the effort and time the reviewer has invested in evaluating our work. We have carefully implemented all the suggested corrections and hope that the revised manuscript is now clearer and more reader-friendly.

*L24: "Coherence" missing second quotation marks.*

We thank the reviewer for bringing this to our attention. A second quotation mark has been added in the revised version of the manuscript.

*L43: "persistence of water MASS properties"*

We thank the reviewer for bringing this to our attention. The sentence has been corrected in the revised version of the manuscript.

*L52: add hyphen in "frame-dependent". Add "and" before "imposed a vortex coherence criterion"*

We thank the reviewer for his correction. The sentence has been corrected in the revised version of the manuscript.

*L55: Vortices continually lose and entrain water mass, so there is no single "point" at which the vortex loses trapped water. Denes et al. (2022) define a median residence lifetime over which half the original fluid is lost, but there are likely other ways to define a coherence lifetime.*

We thank the reviewer for this very interesting comment. While we cannot fully agree or disagree with this statement, we would like to provide additional context.

As far as we know, certain eddies do exhibit a distinct "point" where they lose their coherence rapidly, often within a few days, due to interactions with topography or through splitting and merging events. For instance, NBC rings interacting with the Lesser Antilles (Andrade-Canto et al., 2022; Subirade et al., 2023) and Gulf Stream rings interacting with the continental slope (Richardson et al., 1983) are notable examples. In these cases, eddies remain coherent but experience a sudden collapse, often driven by friction with topography.

To illustrate this, we reference Figure 3 from Andrade-Canto et al. (2022), which depicts an NBC ring losing its coherence between March 11th and March 20th, 2004. This example highlights a clear "point" of coherence loss in the dynamics of NBC rings.

[Figure]

**Figure 3.** Genesis (yellow), evolution (red), and apocalypse (orange) of a coherent Lagrangian North Brazil Current Ring (NBCR), geodesically detected from altimetry-derived velocity data using the sea-surface heigh (SSH) eddy trajectory of Figure 2 as a reference. Selected isobaths (in km) are shown in gray.

We agree with the reviewer's comment regarding the continual entrainment of water masses by eddies, particularly along their boundaries. This is well illustrated by the anticyclonic eddy sampled during the Physindien 2011 experiment, for which thermohaline anomalies are plotted in Figure 4, panels c) and d). These panels show colder and fresher waters advected by the flow around the TC cores.

However, we respectfully do not fully agree with the assertion that vortices continually lose water mass. As long as the vortex is coherent, its loss rate is very small. For instance, Laxenaire et al. (2019) demonstrated that the thermohaline anomalies in the core of Agulhas rings can remain constant for at least up to a year. Similarly, the meddy studied by Armi et al. (1989) retained its trapped water for at least a year, as evidenced by the persistence of thermohaline anomalies.

If water is lost or exchanged, this typically occurs in the outer boundary region rather than within the eddy core itself. For this reason, in our previous work (Barabinot et al., 2024), we defined mesoscale eddy boundaries as the region where the trapped water transitions to and interacts with surrounding waters.

*L56: Please be more specific about how you define "material coherence".*

We thank the reviewer for their comment. Defining "material coherence" is indeed a challenging task, as multiple criteria have been proposed in the literature (e.g., Beron-Vera et al., 2013; Haller et al., 2016; Denes et al., 2022; Froyland et al., 2013, 2015; Bettencourt

et al., 2012; El Aouni et al., 2020). These criteria are typically based on closed flow trajectories.

To address the reviewer's concern, we have added a sentence at line 55 to clarify our statement : *"This vortex ceases to be coherent when it loses its trapped water mass, that is when trajectories are no longer closed"*.

*L67 and elsewhere. Parenthetical citations should not have the year in brackets. Use \citep or similar.*

We thank the reviewer for bringing it to our attention. Citations have been corrected in the revised version.

*L68: "do not consider... diffusion processes". Actually, several MC methods explicit include diffusion. See Refs 27, 28 and 35 in Denes et al. (2022).*

We thank the reviewer for bringing this to our attention. We have revised the wording accordingly and added the suggested references at line 68 to support our statement in the updated manuscript. Here is the new formulation: *"Furthermore, MC theory is based on advection processes only and often does not consider the potential permeability of the eddy boundary due to diffusion processes or lateral intrusion (Joyce et al. 1977, 1984, Ruddick et al. 2010). Nevertheless, some criteria including diffusion can be found in the literature (Froyland et al. 2010, 2013, 2015)."*

*L70: "meddy" is not defined.*

We thank the reviewer for bringing this to our attention. The term "meddy" has been defined in line 70 in the revised version of the manuscript.

*L79: Add comma before "thus challenging..."*

We thank the reviewer for bringing this to our attention. A comma has been added on line 79 in the revised version of the manuscript.

*L106: remove the word "vertically" from inside the brackets, i.e. "O(10m) vertically".*

We thank the reviewer for bringing this to our attention. The word "vertically" has been removed from inside the brackets on line 79 in the revised version of the manuscript.

*L110: "relies of the fact that thermohaline properties of the eddy are maintained throughout its lifetime". This is more of a hypothesis than a fact, albeit a reasonable one. As noted earlier, instabilities and mixing can lead to modifications of water mass properties near the eddy edge. So one might expect the T-S signature of trapped water masses to modulate as there is exchange and interaction with surrounding waters. The timescale over which this occurs is a good question, not addressed here*.

We thank the reviewer for this very interesting comment. The advent of Argo floats has indeed made it easier to measure thermohaline anomalies in eddy cores across different regions, and several examples of persistent thermohaline anomalies maintained throughout eddy lifecycles can be found. Laxenaire et al. (2019, 2020) provide excellent examples of Agulhas rings where thermohaline anomalies remained stable and constant for over a year.

Similarly, Armi et al. (1989) and Paillet et al. (2002), both cited in our manuscript, describe meddies with persistent thermohaline anomalies on isopycnals. Additionally, Aguedjou et al. (2021) offers a comprehensive census of thermohaline anomalies in the eddy cores of the North Tropical Atlantic.

As noted, instabilities primarily affect the eddy boundary, and in the absence of significant mesoscale barotropic or baroclinic instabilities, the eddy core—and its thermohaline properties—remains largely unperturbed. Thermohaline anomalies dissipate when the eddy ceases to be materially coherent.

To strengthen our argument, we have added two sentences to the revised manuscript at line 110. We have revised the manuscript to include the following new sentence:

*"Indeed, with the advent of Argo floats, measuring thermohaline anomalies in eddy cores across different regions has become easier, and several examples of thermohaline anomalies maintained throughout the eddy lifecycle can be found (Aguedjou et al., 2021; Laxenaire et al., 2019, 2020; Paillet et al., 2002; Armi et al., 1989)."*

*L113: Remove extra period.*

We thank the reviewer for bringing this to our attention. We have removed  the extra period in the revised version of the manuscript.

*L118: Remove "To the best of our knowledge"*

We thank the reviewer for his suggestion. We have removed the expression in the revised version of the manuscript.

*L138: Fix name of EUREC4A-OA campaign.*

We thank the reviewer for bringing this to our attention. We have fixed the name of EUREC4A-OA in the revised version of the manuscript.

*L149: "TABLE 1 summarises..."*

We thank the reviewer for bringing this to our attention. We have corrected the sentence in the revised version of the manuscript.

*L150: "LOWERED and ship-mounted"*

We thank the reviewer for bringing this to our attention. We have corrected the sentence in the revised version.

*Table 1: XBT, xCTD and VM are listed in the table but not defined or used elsewhere in the article. Suggest you remove.*

We thank the reviewer for their suggestion. However, we believe that mentioning the instruments is important, as they can be valuable references for future studies. The use of multiple instruments during cruises has facilitated the measurement of vertical sections with

improved horizontal resolution. To address this point, we have defined the instruments in the title of Table 1 for clarity.

*L207: "In the literature" --- provide refs.*

We thank the reviewer for bringing this to our attention. In the revised manuscript, we have removed the sentence at line 207, as we felt the statement was too strong.

*L215: Use in-line citation for Chaigneau et al. 2009.*

We thank the reviewer for bringing this to our attention. We have used in-line citation in the revised version of the manuscript.

*L257: "By recurrence" -- not sure what you mean here.*

We thank the reviewer for bringing this to our attention. In the revised version of the manuscript, we have replaced this expression with the simpler term "therefore" at line 257.

*L260: don't use bold text for x and z.*

We thank the reviewer for bringing this to our attention. The expression has been corrected in the revised version of the manuscript.

*L271 and elsewhere. Be consistent with the abbreviations used for anticyclonic and cyclonic eddies. I prefer AE and CE.*

We thank the reviewer for bringing this to our attention. We have adopted AE and CE as notations for eddies in the revised version of the manuscript.

*L279: "in the literature" -- provide refs.*

We thank the reviewer for bringing this to our attention. In the revised version of the manuscript, this sentence has been removed as several references have already been provided between lines 278 and 285.

*L282: "in studies" --- provide refs.*

We thank the reviewer for bringing this to our attention. We have added three references in the revised version of the manuscript.

*L287: "vertical component of the linear momentum EQUATION". Or simply "the vertical momentum equation"*

We thank the reviewer for his suggestion. The wording has been corrected in the revised version of the manuscript.

*L293: use italic for EPV for consistency.*

We thank the reviewer for bringing this to our attention. We have corrected the expression in the revised version of the manuscript.

*L341: not sure what lower case r means.*

We thank the reviewer for bringing this to our attention. R refers to the eddy radius. We have modified the corresponding sentence in the revised version of the manuscript.

*L351: "some altimetric studies" -- provide refs.*

We thank the reviewer for bringing this to our attention. We have added three references in the revised version of the manuscript.

*L378: the word "and" should not be italicised.*

We thank the reviewer for bringing this to our attention. We have corrected the expression in the revised version of the manuscript.

*L416: The two ellipses E1 and E1 provide inner and outer bounds on the best fitted ellipse. So why not take an average of the volumes calculated using each?*

We thank the reviewer for their comment. Since eddies do not appear symmetrical on 2D vertical sections, the reconstructed 3D shapes of ellipses E1 and E2 differ significantly. The 3D structure of eddies remains an area of ongoing investigation within the scientific community, and our goal was to provide several examples of what this 3D shape might look like.

To address this point, we have added a sentence at line 423 in the revised manuscript to provide additional detail. Here is the sentence:

*"As the vertical shape of eddies is not well understood, especially the shape of their thermohaline coherent core, in the literature, we present the two ellipses as examples of what an eddy core can look like in 3D."*

*L430: "Physindien" is capitalised in section 2.1 and called something different in Tables 1-3.*

We thank the reviewer for bringing this to our attention. The wording has been revised throughout the updated version of the manuscript..

*L460: Add space after period before "Our comparison"*

We thank the reviewer for bringing this to our attention. A space has been added on line 467 in the revised version of the manuscript.

*Figure 9 caption. Capitalise "velocity". Add space after period before "The symbols..."*

We thank the reviewer for bringing this to our attention. Typos have been corrected in the revised version of the manuscript.

*L512: Physindien again different from earlier in the manuscript.*

We thank the reviewer for bringing this to our attention. The wording has been revised throughout the updated version of the manuscript.

*Figure 10 caption. I think the green lines are actually "cyan".*

We thank the reviewer for bringing this to our attention. The caption has been corrected in the revised version of the manuscript.

*Equation A1 and L643: mixing lower case and upper case deltas here. Please be consistent.*

We thank the reviewer for bringing this to our attention. Notations in equation (A1) have been corrected in the revised version of the manuscript.

*L678: The units for alpha and beta are not consistent. I think alpha should be units kg m^{-3} / K and beta should be kg m^{-3} / g kg^{-1}, assuming that salinity is measured in g/kg.*

We thank the reviewer for his correction. The units were indeed incorrect and now they have been corrected in the revised version of the manuscript.

*L668: "f-PLANE"*

We thank the reviewer for his correction. The expression has been corrected in the revised version of the manuscript.

*L674: Not entirely sure, but I think there is an error here. I think the points P' and Q' need to be defined such that Q'C = C P and Q C = C P'.*

We thank the reviewer for pointing out this oversight. It was an honest mistake, and we fully agree with the correction.

Refs:

D. Chelton, P. Gaube, M. Schlax, J. Early, and R. Samelson, The influence of nonlinear mesoscale eddies on near-surface oceanic chlorophyll, Science 334, 328 (2011).

A. R. Robinson, ed., Eddies in Marine Science (Springer, Berlin, 1983).

Rykova, T., & Oke, P. R. (2022). Stacking of EAC eddies observed from Argo. Journal of Geophysical Research: Oceans, 127, e2022JC018679."

Refs:

Aguedjou, H. M. A., Chaigneau, A., Dadou, I., Morel, Y., Pegliasco, C., Da-Allada, C. Y., & Baloïtcha, E. (2021). What can we learn from observed temperature and salinity isopycnal anomalies at eddy generation sites? Application in the tropical Atlantic Ocean. *Journal of Geophysical Research: Oceans*, *126*(11), e2021JC017630.

Andrade‑Canto, F., & Beron‑Vera, F. J. (2022). Do eddies connect the tropical Atlantic Ocean and the Gulf of Mexico?. *Geophysical research letters*, *49*(20), e2022GL099637.

Barabinot, Y., Speich, S., & Carton, X. (2024). Defining mesoscale eddies boundaries from in‑situ data and a theoretical framework. *Journal of Geophysical Research: Oceans*, *129*(2), e2023JC020422.

Beron-Vera, F. J., Wang, Y., Olascoaga, M. J., Goni, G. J., & Haller, G. (2013). Objective detection of oceanic eddies and the Agulhas leakage. *Journal of Physical Oceanography*, *43*(7), 1426-1438.

Bettencourt, J. H., López, C., & Hernández-García, E. (2013). Characterization of coherent structures in three-dimensional turbulent flows using the finite-size Lyapunov exponent. *Journal of Physics A: Mathematical and Theoretical*, *46*(25), 254022.

El Aouni, A., Daoudi, K., Yahia, H., Maji, S. K., & Minaoui, K. (2020). Defining Lagrangian coherent vortices from their trajectories. *Physics of Fluids*, *32*(1).

Froyland, G. (2013). An analytic framework for identifying finite-time coherent sets in time-dependent dynamical systems. *Physica D: Nonlinear Phenomena*, *250*, 1-19.

Froyland, G., & Padberg-Gehle, K. (2015). A rough-and-ready cluster-based approach for extracting finite-time coherent sets from sparse and incomplete trajectory data. *Chaos: An Interdisciplinary Journal of Nonlinear Science*, *25*(8).

Haller, G., Hadjighasem, A., Farazmand, M., & Huhn, F. (2016). Defining coherent vortices objectively from the vorticity. *Journal of Fluid Mechanics*, *795*, 136-173.

Laxenaire, R., Speich, S., & Stegner, A. (2019). Evolution of the thermohaline structure of one Agulhas ring reconstructed from satellite altimetry and Argo floats. *Journal of Geophysical Research: Oceans*, *124*(12), 8969-9003.

Laxenaire, R., Speich, S., & Stegner, A. (2020). Agulhas ring heat content and transport in the South Atlantic estimated by combining satellite altimetry and Argo profiling floats data. *Journal of Geophysical Research: Oceans*, *125*(9), e2019JC015511.

Paillet, J., Le Cann, B., Carton, X., Morel, Y., & Serpette, A. (2002). Dynamics and evolution of a northern meddy. *Journal of Physical Oceanography*, *32*(1), 55-79.

Richardson, P. L. (1983). Gulf stream rings. In *Eddies in marine science* (pp. 19-45). Berlin, Heidelberg: Springer Berlin Heidelberg.

Subirade, C., L'hégaret, P., Speich, S., Laxenaire, R., Karstensen, J., & Carton, X. (2023). Combining an Eddy Detection Algorithm with In-Situ Measurements to Study North Brazil Current Rings. *Remote Sensing*, *15*(7), 1897.